# On the convergence of policy gradient methods to Nash equilibria in general stochastic games

**Angeliki Giannou**
University of Wisconsin-Madison
giannou@wisc.edu

**Kyriakos Lotidis**
Stanford University
klotidis@stanford.edu

**Panayotis Mertikopoulos**
Univ. Grenoble Alpes, CNRS, Inria, Grenoble INP, LIG, 38000 Grenoble, France
Criteo AI Lab
panayotis.mertikopoulos@imag.fr

**Emmanouil V. Vlatakis-Gkaragkounis**
University of California, Berkeley
emvlatakis@berkeley.edu

## Abstract

Learning in stochastic games is a notoriously difficult problem because, in addition to each other's strategic decisions, the players must also contend with the fact that the game itself evolves over time, possibly in a very complicated manner. Because of this, the convergence properties of popular learning algorithms – like policy gradient and its variants – are poorly understood, except in specific classes of games (such as potential or two-player, zero-sum games). In view of this, we examine the long-run behavior of policy gradient methods with respect to Nash equilibrium policies that are second-order stationary (SOS) in a sense similar to the type of sufficiency conditions used in optimization. Our first result is that SOS policies are locally attracting with high probability, and we show that policy gradient trajectories with gradient estimates provided by the REINFORCE algorithm achieve an $\mathcal{O}(1/\sqrt{n})$ distance-squared convergence rate if the method's step-size is chosen appropriately. Subsequently, specializing to the class of *deterministic* Nash policies, we show that this rate can be improved dramatically and, in fact, policy gradient methods converge within a *finite* number of iterations in that case.

## 1 Introduction

Ever since they were introduced by Shapley [50] in the 1950's, stochastic games have been one of the staples of non-cooperative game theory, with a range of pioneering applications to multi-agent reinforcement learning [51], unmanned vehicles [49], general game-playing [37, 52, 57], etc. Informally, a stochastic game unfolds in discrete time as follows: At each point in time, the players are at a given state which determines the rules of the game for that stage. The actions of the players in this state determine not only their instantaneous payoffs (as defined by the stage game), but also the transition probabilities towards the next state of the process. In this way, each player has to balance two distinct – and often competing – objectives: optimizing the payoffs of *today* versus picking a possibly suboptimal action which could yield significant benefits *tomorrow* (i.e., by influencing the transitions of the process towards a more favorable state for the player).

36th Conference on Neural Information Processing Systems (NeurIPS 2022).

Since all players in the game are involved in a similar dilemma, the decision-making problem for each player is a very complicated affair. In particular, in addition to their changing strategic decisions, the players of the game must also contend with the fact that the stage game itself evolves over time. Because of this, even the existence of a Nash equilibrium policy – viz. a stationary Markovian policy that is stable to unilateral deviations [16] – is far more difficult to prove compared to standard, stateless normal form games; for a comprehensive survey, cf. [41, 53] and references therein.

The question we seek to address in this paper is whether an ensemble of boundedly rational players can reach an equilibrium policy in a stochastic game. Specifically, if players do not have sufficient information – or the computational resources required – to solve a high-dimensional Bellman equation [15, 54], it is not at all clear if they would somehow end up playing a Nash policy in the long run. After all, the complexity of most games increases exponentially with the number of players, so the identification of a game's equilibria quickly becomes prohibitively difficult [27].

**Our contributions in the context of related work.** This issue has sparked a vigorous literature with important ramifications for the range of applications mentioned above. Nevertheless, these efforts must grapple with a series of strong lower bounds for computing even weaker solution concepts like coarse correlated equilibria in turn-based stochastic games [12, 27]. On that account, a recent line of work has focused on establishing convergence in *specific* subclasses of stochastic games, such as *min-max* [7, 11, 32, 47, 48, 58] and common interest *potential* games [13, 31, 61]. However, despite these encouraging results, the general case remains particularly elusive.

Our paper takes a complementary approach to the above and seeks to study the convergence landscape of a class of *equilibrium policies* – not *games*. For concreteness, we focus on the general class of policy gradient methods as pioneered by [28, 29, 55, 59], and we examine the methods' convergence properties in general random stopping games – as opposed to ergodic stochastic games with an infinite horizon [32, 42]. Concretely, this means that the sequence of play evolves episode-by-episode: within each episode, the players commit a policy and play the game, and from one episode to the next, they use an iterative gradient step to update their policy and continue playing.

Our main contributions in this general context may be summarized as follows:

1. We introduce a flexible algorithmic template for the analysis of policy gradient methods which accounts for different information and update frameworks – from perfect policy gradients to value-based estimates obtained on a per-episode basis, e.g., via the REINFORCE algorithm [4, 55, 59].

2. Within this framework, we show that Nash policies that satisfy a certain strategic stability condition are locally attracting with arbitrarily high probability. Moreover, to estimate the method's rate of convergence, we focus on Nash policies that satisfy a second-order sufficiency condition similar to the type of sufficiency conditions used in optimization, and we show that such policies enjoy an $\mathcal{O}(1/\sqrt{n})$ squared distance convergence rate.

3. Finally, we also consider the method's convergence to *deterministic* Nash policies – a special case of SOS policies – and we show that, generically, the above rate can be improved dramatically. In particular, by a simple tweak to the method's projection step, the induced sequence of play converges to equilibrium in a *finite* number of iterations, despite all the noise and uncertainty.

It is also worth noting that our analysis focuses squarely on the actual, episode-by-episode trajectory of play, not any "best-iterate" or time-averaged variant thereof. In regards to the latter class of guarantees, the recent work of Jin et al. [26] proposed an algorithm (called V-learning) which updates the policy $\pi_n$ of the $n$-th episode based on the observed rewards so far. Thanks to the algorithm's regret guarantees, Jin et al. [26] showed that (*a*) in min-max games, the time-averaged policy $\bar{\pi}_n = (1/n) \sum_{k=1}^{n} \pi_k$ converges to equilibrium at a rate of $\mathcal{O}(1/\sqrt{n})$; whereas (*b*) in *general* stochastic games, the empirical frequency of play converges to the game's set of coarse correlated equilibria (a substantial relaxation of the notion of Nash equilibrium) at a rate of $\mathcal{O}(1/\sqrt{n})$.

By contrast, as we mentioned above, our paper focuses on the *actual* sequence of play, i.e., the policy $\pi_n$ employed at each episode of the game. Moreover, the rates that we obtain all concern the convergence of the players' policies to a *Nash* equilibrium – not a correlated equilibrium or other relaxation thereof. In this regard, the best-iterate / ergodic convergence rates are incomparable to our own as they concern a weaker type of convergence (time-averaged instead of the actual sequence), and to a weaker solution concept (correlated equilibria instead of Nash equilibria). This aspect of our

results is especially relevant for multi-agent reinforcement learning scenarios where agents learn "on the fly", and it has important ramifications for many of the practical applications of stochastic games.

From a technical standpoint, our analysis is based on mapping the problem of multi-agent policy learning to the problem of equilibrium learning in a class of continuous games characterized by the fact that first-order stationary points are necessarily Nash (itself a consequence of the so-called "gradient dominance" property of stochastic games). By means of this reframing, we are able to leverage a series of recent techniques for establishing local convergence in (non-monotone) continuous games and variational inequalities [3, 8, 24, 25, 33, 45], which ultimately also yield convergence in our setting. As a result, even though the unbounded variance of the REINFORCE estimator is a source of considerable complications, the resulting link between stochastic and continuous games is of particular technical interest because it opens up a wide array of stochastic approximation tools and techniques that can be used for the analysis of multi-agent learning in stochastic games.

## 2    Preliminaries

**2.1. Setup of the game.** Throughout this work we consider $N$-player generic stochastic games where players repeatedly select actions in a shared Markov decision process (MDP) with the goal of maximizing their individual value functions. Formally, we study the tabular version with random stopping of general stochastic games, which is specified by a tuple $\mathcal{G} = (\mathcal{S}, \mathcal{N}, \{\mathcal{A}_i, R_i\}_{i \in \mathcal{N}}, P, \zeta, \rho)$ with the following primitives:

- A finite set of *agents* $i \in \mathcal{N} = \{1, 2, \ldots, N\}$ and a finite set of *states* $\mathcal{S} = \{1, \ldots, S\}$.
- For each $i \in \mathcal{N}$, a finite space of *actions* (or *pure strategies*) $\mathcal{A}_i$ indexed by $\alpha_i = 1, \ldots, A_i = |\mathcal{A}_i|$. We will write $\mathcal{A} = \prod_{i \in \mathcal{N}} \mathcal{A}_i$ and $\mathcal{A}_{-i} = \prod_{j \neq i} \mathcal{A}_j$ for the action space of all agents and that of all agents other than $i$ respectively. In a similar vein, we will also write $\alpha = (\alpha_i, \alpha_{-i})$ when we want to highlight the action $\alpha_i$ of player $i$ against the action profile $\alpha_{-i}$ of $i$'s opponents.
- For each $i \in \mathcal{N}$, we will write $R_i \colon \mathcal{S} \times \mathcal{A} \to [-1, 1]$ for the *reward function* of agent $i \in \mathcal{N}$, i.e., $R_i(s, \alpha_i, \alpha_{-i})$ will denote the value of the reward of agent $i$ when the game is at state $s \in \mathcal{S}$, the focal agent $i \in \mathcal{N}$ plays $\alpha_i \in \mathcal{A}_i$, and all other agents take actions $\alpha_{-i} \in \mathcal{A}_{-i}$.
- The game transits from one state to another according to a Markov transition process, so that $P(s' \mid s, \alpha)$ denotes the probability of transitioning from $s$ to $s'$ when $\alpha \in \mathcal{A}$ is the action profile chosen by the agents.
- Given an action profile $\alpha$ at state $s$, the process terminates with probability $\zeta_{s,\alpha} > 0$, i.e., $\zeta_{s,\alpha} = 1 - \sum_{s' \in \mathcal{S}} P(s' \mid s, \alpha)$; for convenience, we will write $\zeta := \min_{s,\alpha}\{\zeta_{s,\alpha}\}$.
- $\rho \in \Delta(\mathcal{S})$ is the distribution for the initial state of the game.

**Episodic Setting.**    We consider an episodic setting, where in each episode a realization of the game is completed. At every time step $t \geq 0$ of each episode, all agents observe the common state $s_t \in \mathcal{S}$, select actions $\alpha_t$ and receive rewards $\{R_i(s_t, \alpha_t)\}_{i \in \mathcal{N}}$. Then, with probability $\zeta_{s_t,\alpha_t}$ the game terminates, and with probability $1 - \zeta_{s_t,\alpha_t}$, it moves to the state $s_{t+1}$, which is drawn according to $P(\cdot|s_t, \alpha_t)$. Denoting the realized reward of player $i$ at time $t$ as $r_{i,t} := R_i(s_t, \alpha_t)$, we will write $\tau = (s_t, \alpha_t, r_t)_{t \leq T(\tau)}$ to denote the trajectory of the episode, where $r_t := (r_{i,t})_{i \in \mathcal{N}}$, and $T(\tau)$ the time the episode terminates.

**Policies and value functions.**    We consider *stationary Markovian* policies, i.e., policies that do not depend on the time-step and the history, given the current state of the game. More specifically, for each agent $i \in \mathcal{N}$, a *policy* $\pi_i \colon \mathcal{S} \to \Delta(\mathcal{A}_i)$ specifies a probability distribution over the actions of agent $i$ in state $s \in \mathcal{S}$, i.e., $\alpha_i \sim \pi_i(\cdot|s)$ denotes the (random) action drawn by agent $i$ at state $s \in \mathcal{S}$ according to $\pi_i$, viewed here as an element of $\Pi_i := \Delta(\mathcal{A}_i)^{\mathcal{S}}$. In addition, we will also write $\pi = (\pi_i)_{i \in \mathcal{N}} \in \Pi := \prod_i \Pi_i$ and $\pi_{-i} = (\pi_j)_{j \neq i} \in \Pi_{-i} := \prod_{j \neq i} \Pi_j$ for the policy profile of all agents and all agents other than $i$, respectively.

The expected reward of agent $i \in \mathcal{N}$ if agents follow policy $\pi$, starting from initial state $s \in \mathcal{S}$, defines the *value function* of agent $i$, denoted as $V_{i,s}(\pi)$, and is equal to

$$V_{i,s}(\pi) := \mathbb{E}_{\tau \sim \text{MDP}}\left[\sum_{t=0}^{T(\tau)} R_i(s_t, \alpha_t)\Big| s_0 = s\right] \tag{1}$$

where $\tau \sim \text{MDP}$ denotes the randomness induced by the policy profile $\pi$, and the state-transition probabilities of the MDP. Overloading the notation, we set $V_{i,\rho}(\pi) := \mathbb{E}_{s \sim \rho}[V_{i,s}(\pi)]$. Although value

functions are, in general, non-convex, they share similar smoothness properties with the payoff functions of normal form games, namely bounded and Lipschitz gradients. For precise statements, we defer to the paper's supplement.

**Visitation distribution and the mismatch coefficient.** For a policy profile $\pi \in \Pi$ and an arbitrary initial state distribution $s_0 \sim \rho$, we define the discounted state visitation measure/distribution as

$$\tilde{d}_\rho^\pi(s) = \mathbb{E}_{\tau \sim \mathrm{MDP}}\left[\sum_{t=0}^{T(\tau)} \mathbb{1}\{s_t = s\}\Big| s_0 \sim \rho\right], \quad d_\rho^\pi(s) := \tilde{d}_\rho^\pi(s)/Z_\rho^\pi$$

In the appendix, we prove formally that the above definition is well-posed for the random stopping episodic framework described above, i.e., $\tilde{d}_\rho^\pi(s) < \infty$, so $Z_\rho^\pi := \sum_{s \in \mathcal{S}} \tilde{d}_\rho^\pi(s)$ is well-defined. In our proofs, we will leverage a standard property of visitation distributions, namely the equivalence of the expected value of state-action function and the expected cumulative value over a random trajectory. More precisely, we have:

**Lemma 1.** [Conversion Lemma] *For an arbitrary state-action function $f : \mathcal{S} \times \mathcal{A} \to \mathbb{R}$, a policy profile $\pi$ and an initial state distribution $s_0 \sim \rho$, we have*

$$\mathbb{E}_{\tau \sim \mathrm{MDP}}\left[\sum_{t=0}^{T(\tau)} f(s_t, \alpha_t)\right] = Z_\rho^\pi \mathbb{E}_{s \sim d_\rho^\pi} \mathbb{E}_{\alpha \sim \pi(\cdot|s)}[f(s, \alpha)] \tag{2}$$

Finally, to quantify the difficulty of hard-to-reach states via a policy gradient method, we will follow the standard approach of [9, 14, 38, 39, 61] and use an appropriately-defined distribution "mismatch coefficient", generalizing the single-agent counterpart of Agarwal et al. [1]. More precisely, for a stochastic game $\mathcal{G}$, we define the *mismatch coefficient* as $\mathcal{C}_\mathcal{G} := \max_{\pi, \pi' \in \Pi}\{\|\tilde{d}_\rho^\pi/\tilde{d}_\rho^{\pi'}\|_\infty\}$ or, more simply, as $\mathcal{C}_\mathcal{G} := \max_{\pi, \in \Pi}\{\frac{1}{\zeta}\|d_\rho^\pi/\rho\|_\infty\}$. Similar to prior work in this direction [1, 5, 11], we will assume $\mathcal{C}_\mathcal{G}$ is finite, which, equivalently, means that $d_\rho^\pi(s) > 0$ for any policy $\pi$ and state $s$.

**2.2. Solution concepts.** The most widely used solution concept in game theory is that of a Nash equilibrium i.e., a strategy profile $\pi^* \in \Pi$ that discourages unilateral deviations. However, in stochastic games, the definition of a Nash policy is much more involved because of the existence of multiple states and steps, cf. [16, 50, 53, 56] and references therein. Formally, we have:

**Definition 1** (Nash policies). A policy $\pi^* = (\pi_i^*)_{i \in \mathcal{N}} \in \Pi$ is said to be a *Nash policy* for a given distribution of initial states $\rho \in \Delta(\mathcal{S})$ if, for every player $i \in \mathcal{N}$, we have

$$V_{i,\rho}(\pi_i^*; \pi_{-i}^*) \geq V_{i,\rho}(\pi_i; \pi_{-i}^*) \quad \text{for all } i \in \mathcal{N} \text{ and all } \pi_i \in \Delta(\mathcal{A}_i)^\mathcal{S}. \tag{NE}$$

In contrast to general non-convex continuous games, stochastic games satsify a version of the well-known Polyak-Łojasiewicz condition [44] but with linear gradient growth, also known as a *gradient dominance property* (GDP) [1, 5]. For the multi-agent case, Zhang et al. [61] and Daskalakis et al. [11] showed that a similar property holds even in the episodic setting:

**Lemma 2** (Gradient dominance property). *For any policy profile $\pi = (\pi_i)_{i \in \mathcal{N}} \in \Pi$, we have that*

$$V_{i,\rho}(\pi_i'; \pi_{-i}) - V_{i,\rho}(\pi_i; \pi_{-i}) \leq \mathcal{C}_\mathcal{G} \max_{\bar{\pi}_i \in \Pi_i}\langle \nabla_i V_{i,\rho}(\pi), \bar{\pi}_i - \pi_i\rangle \tag{GDP}$$

*for any unilateral deviation $\pi_i' \in \Pi_i$ of player $i \in \mathcal{N}$.*

*Remark.* In the above and throughout our paper, we will write $\nabla_i$ to denote the gradient of the quantity in question with respect to $\pi_i$, i.e., when $\pi_{-i}$ is kept fixed and only $\pi_i$ is varied. For concision, we will write $v_i(\pi) = \nabla_i V_{i,\rho}(\pi)$ for the individual gradient of player $i$'s value function, and $v(\pi) = (v_i(\pi))_{i \in \mathcal{N}}$ for the ensemble thereof. ♦

Thanks to (GDP), it is straightforward to check that first-order stationary (FOS) points of $V$ are Nash. Formally, as in [11, 31, 61], we have the following characterization:

**Lemma 3** (First-order stationary policies are Nash). *A policy $\pi^* = (\pi_i^*)_{i \in \mathcal{N}} \in \Pi$ is Nash if and only if it satisfies the first-order stationary condition*

$$\langle v(\pi^*), \pi - \pi^*\rangle \leq 0 \quad \text{for all } \pi \in \Pi. \tag{FOS}$$

Leonardos et al. [31] and Zhang et al. [61] proved a relaxation of the above lemma to the effect that policies that satisfy (FOS) up to $\varepsilon$ (i.e., in lieu of 0 in the RHS) are $\mathcal{O}(\varepsilon)$-Nash. Going in the other direction, we will consider the following series of refinements of Nash policies which are particularly important from a learning standpoint [30, 53]:

**Definition 2.** Let $\pi^* = (\pi_i^*)_{i \in \mathcal{N}} \in \Pi$ be a Nash policy. We then say that:

- $\pi^*$ is *stable* if $\langle v(\pi), \pi - \pi^* \rangle < 0$ for all $\pi \neq \pi^*$ sufficiently close to $\pi^*$.

- $\pi^*$ is *second-order stationary* if it satisfies the sufficiency condition
$$(\pi - \pi^*)^\top \operatorname{Jac}_v(\pi^*)(\pi - \pi^*) < 0 \quad \text{for al } \pi \in \Pi \setminus \{\pi^*\}, \tag{SOS}$$
where $\operatorname{Jac}_v(\pi^*) = (\nabla_j v_i(\pi^*))_{i,j \in \mathcal{N}} = (\nabla_j \nabla_i V_i(\pi^*))_{i,j \in \mathcal{N}}$ denotes the Jacobian of $v$ at $\pi^*$.

- $\pi^*$ is *deterministic* if it induces a deterministic selection rule $\pi_i^* \colon \mathcal{S} \to \mathcal{A}_i$ for all $i \in \mathcal{N}$.

- $\pi^*$ is *strict* if it is deterministic and (FOS) holds as a strict inequality whenever $\pi \neq \pi^*$.

*Remark* 1. In the above and what follows, "sufficiently close" means that there exists a neighborhood $\mathcal{U}$ of $\pi^*$ in $\Pi$ such that the stated inequality holds for all $\pi \in \mathcal{U}$. Unless mentioned otherwise, we will measure distances on $\Pi$ relative to the Euclidean norm, but this choice does not impact our results.

Intuitively, the condition for equilibrium stability is a game-theoretic analogue of first-order KKT sufficiency condition, while the condition for second-order stationarity is the second-order version thereof. In this regard, the distinction between first-order stationary, stable and second-order stationary points is formally analogous to the distinction between critical points, minimizers, and second-order minimum points in optimization. As for deterministic policies, we should mention that, generically, deterministic policies are also strict, so we will use the two terms interchangeably.[1]

Importantly, as we show in **??**, these refinements admit the following characterizations:

**Proposition 1.** *Let $\pi^* = (\pi_i^*)_{i \in \mathcal{N}} \in \Pi$ be a Nash policy. Then:*

a) *If $\pi^*$ is second-order stationary, there exists some $\mu > 0$ such that*
$$\langle v(\pi), \pi - \pi^* \rangle \leq -\mu \|\pi - \pi^*\|^2 \quad \text{for all } \pi \text{ sufficiently close to } \pi^*. \tag{3a}$$

b) *If $\pi^*$ is strict, there exists some $\mu > 0$ such that*
$$\langle v(\pi), \pi - \pi^* \rangle \leq -\mu \|\pi - \pi^*\| \quad \text{for all } \pi \text{ sufficiently close to } \pi^*. \tag{3b}$$

In view of all the above, we get the following string of implications for equilibria in generic games:
$$\text{strict/deterministic} \implies \text{SOS} \implies \text{stable} \implies \text{FOS} = \text{Nash} \tag{4}$$

For posterity, we should clarify here that, due to the highly complicated structure of the game's value functions, it is not trivial to construct a concrete example where (3a) holds but (3b) does not. Examples of strict Nash policies abound in the literature [30, 53], but we are not otherwise aware of an argument that could be used to close the gap between (3a) and (3b). In view of this, our analysis will treat both cases concurrently (with the obvious anticipation that more refined solution concepts should enjoy stronger convergence guarantees).

## 3 Policy gradient methods

We now proceed to describe our general model for episodic learning in stochastic games. To that end, we will consider a framework where agents follow a specific policy $\pi_n$ within each episode, and update it from one episode to the next with the objective of increasing their individual rewards. Formally, our approach will adhere to the following inter-episode sequence of events:

1. At the beginning of each episode $n = 1, 2, \ldots$, every agent $i \in \mathcal{N}$ chooses a policy $\pi_{i,n} \in \Pi_i$.
2. Within the $n$-th episode, each player executes their chosen policy $\pi_{i,n}$, inducing in this way an intra-episode trajectory of play $\tau_n = (s_t^{(n)}, \alpha_t^{(n)}, r_t^{(n)})_{t \leq T(\tau_n)}$.
3. Once the episode terminates, agents update their policies and the process repeats.

In terms of feedback, we will treat several models, depending on what type of information is available to the agents during play. More precisely, we will focus on the generic policy gradient (PG) template
$$\pi_{n+1} = \operatorname{proj}_\Pi(\pi_n + \gamma_n \hat{v}_n) \tag{PG}$$
where:

---

[1]The notion of genericity is stated here in the sense of Baire, i.e., the stated property holds for all but a "meager" set of games (i.e., a countable union of nowhere dense sets in the space of all games).

| **Algorithm 1:** REINFORCE | **Algorithm 2:** $\varepsilon$-GREEDY POLICY GRADIENT |
|---|---|
| 1: **Input:** $\hat{\pi} \in \Pi, \tau = (s_t, \alpha_t, r_t)_{t \leq T(\tau)} \in \mathcal{T}$ | 1: **Input:** $\pi_1, \{\gamma_n\}_{n \in \mathbb{N}}, \{\varepsilon_n\}_{n \in \mathbb{N}}$ |
| 2: **for** $i = 1, \ldots, N$ **do** | 2: **for** $n = 1, 2, \ldots$ **do** |
| 3: $\quad R_i(\tau) \leftarrow \sum_{t=0}^{T(\tau)} r_{i,t}$ | 3: $\quad \hat{\pi}_n \leftarrow (1 - \varepsilon_n)\pi_n + \frac{\varepsilon_n}{|\mathcal{A}|}$ |
| 4: $\quad \Lambda_i(\tau) \leftarrow \sum_{t=0}^{T(\tau)} \nabla_i(\log \hat{\pi}_i(\alpha_{i,t}|s_t))$ | 4: $\quad$ Sample $\tau_n \sim \text{MDP}(\hat{\pi}_n|s_0)$ |
| 5: $\quad \hat{v}_i \leftarrow R_i(\tau) \cdot \Lambda_i(\tau)$ | 5: $\quad \hat{v}_n \leftarrow \text{REINFORCE}(\hat{\pi}_n, \tau_n)$ |
| 6: **end for** | 6: $\quad \pi_{n+1} \leftarrow \text{proj}_\Pi(\pi_n + \gamma_n \hat{v}_n)$ |
| 7: **return** $\{\hat{v}_i\}_{i \in \mathcal{N}}$ | 7: **end for** |

1. $\pi_n = (\pi_{i,n})_{i \in \mathcal{N}} \in \Pi$ denotes the player's policy profile at each episode $n = 1, 2, \ldots$
2. $\hat{v}_n = (\hat{v}_{i,n})_{i \in \mathcal{N}} \in \prod_i \mathbb{R}^{\mathcal{A}_i \times \mathcal{S}}$ is an estimate for the agents' inidividual policy gradients.
3. $\text{proj}_\Pi \colon \prod_i \mathbb{R}^{\mathcal{A}_i \times \mathcal{S}} \to \Pi$ denotes the Euclidean projection to the agents' policy space $\Pi$.
4. $\gamma_n > 0$ is the method's step-size, for which we will assume throughout that $\sum_n \gamma_n = \infty$; typically, (PG) is run with a step-size of the form $\gamma_n = \gamma/(n + m)^p$ for some $\gamma > 0$, $m \geq 0$ and $p \geq 0$.

Regarding the gradient signal $\hat{v}_n$, we will decompose it as

$$\hat{v}_n = v(\pi_n) + U_n + b_n \tag{5}$$

where

$$U_n = \hat{v}_n - \mathbb{E}[\hat{v}_n | \mathcal{F}_n] \quad \text{and} \quad b_n = \mathbb{E}[\hat{v}_n | \mathcal{F}_n] - v(\pi_n). \tag{6}$$

In the above, we treat $\pi_n$, $n = 1, 2, \ldots$, as a stochastic process on some complete probability space $(\Omega, \mathcal{F}, \mathbb{P})$, and we write $\mathcal{F}_n \coloneqq \mathcal{F}(\pi_1, \ldots, \pi_n) \subseteq \mathcal{F}$ for the history (adapted filtration) of $\pi_n$ up to – and including – stage $n$. By definition, $\mathbb{E}[U_n | \mathcal{F}_n] = 0$ and $b_n$ is $\mathcal{F}_n$-measurable, so $U_n$ can be intepreted as a random, zero-mean error relative to $v(\pi_n)$, whereas $b_n$ captures all systematic (non-zero-mean) errors. To make this precise, we will further assume that $b_n$ and $U_n$ are bounded as

$$\mathbb{E}[\|b_n\| \,|\, \mathcal{F}_n] \leq B_n \qquad \text{and} \qquad \mathbb{E}[\|U_n\|^2 \,|\, \mathcal{F}_n] \leq \sigma_n^2 \tag{7}$$

where the sequences $B_n$ and $\sigma_n$, $n = 1, 2, \ldots$, are to be construed as deterministic upper bounds on the bias, fluctuations, and magnitude of the gradient signal $\hat{v}_n$.

Depending on these bounds, a gradient signal with $B_n = 0$ will be called *unbiased*, and an unbiased signal with $\sigma_n = 0$ will be called *perfect*. More generally, we will assume that the above statistics are bounded as

$$B_n = \mathcal{O}(1/n^{\ell_b}) \qquad \text{and} \qquad \sigma_n = \mathcal{O}(n^{\ell_\sigma}) \tag{8}$$

for some $\ell_b, \ell_\sigma > 0$ which depend on the specific model under consideration. For concreteness, we describe below three basic models that adhere to the above template for $\hat{v}_n$ in order of decreasing information requirements:

**Model 1** (Full gradient information). The first model we will consider assumes that agents observe their *full policy gradients*, i.e.,

$$\hat{v}_n = v(\pi_n) \tag{9}$$

implying in particular that $U_n = b_n = 0$. This model is fully deterministic across episodes (though intra-episode play remains stochastic). In particular, it tacitly assumes that agents know the game (and can observe their opponents' policies) sufficiently well so as to calculate the full gradients of their individual value functions $V_{i,\rho}$, cf. [2, 31, 61] and references therein. $\blacklozenge$

**Model 2** (Learning with stochastic gradients). A relaxation of the above model which is particularly relevant for applications to deep reinforcement learning concerns the case where the player have access to stochastic policy gradients [60], i.e., unbiased gradient estimates of the form

$$\hat{v}_n = v(\pi_n) + U_n \tag{10}$$

with $\mathbb{E}[U_n | \mathcal{F}_n] = 0$ (so we can formally take $\ell_b = \infty$ and $\ell_\sigma = 0$ in Eq. (8) above). $\blacklozenge$

**Model 3** (Value-based learning). The last model we will consider concerns the case where agents only have access to their instantaneous rewards and need to reconstruct their individual gradients based on this information. A widely used method to achieve this is via the Reinforce subroutine, which we describe in pseudocode form in Algorithm 1. In words, when employing Reinforce, each agent $i \in i$ commits to a sampling policy $\hat{\pi}_i \in \Pi_i$ and executes it in an episode of the stochastic game in play. Then, at the end of the episode, players gather the total reward $R_i(\tau) \leftarrow \sum_{t=0}^{T(\tau)} r_{i,t}$ associated to the intra-episode trajectory of play $\tau$, and they estimate their policy gradients via the so-called "log-trick" [59] as

$$\hat{v}_i = R_i(\tau) \cdot \sum_{t=0}^{T(\tau)} \nabla_i(\log \hat{\pi}_i(\alpha_{i,t}|s_t)). \tag{11}$$

Lemma 4 below provides the vital statistics of the Reinforce estimator:

**Lemma 4.** *Suppose that each agent $i \in \mathcal{N}$ follows a stationary policy $\pi_i \in \Pi_i$. Then:*

*a)* $\mathbb{E}_{\tau\sim\text{MDP}}[\text{Reinforce}(\pi)] = v(\pi)$ (12a)

*b)* $\mathbb{E}_{\tau\sim\text{MDP}}\Big[\|\text{Reinforce}_i(\pi) - v_i(\pi)\|^2\Big] \leq \dfrac{24A_i}{\kappa_i \zeta^4}$ (12b)

*where $\kappa_i = \min_{s\in\mathcal{S},\alpha_i\in\mathcal{A}_i} \pi_i(\alpha_i|s)$.*

Therefore, if Reinforce is executed at $\hat{\pi} \leftarrow \pi_n$ at each episode $n = 1, 2, \ldots$, we will have

$$\mathbb{E}[\hat{v}_{i,n}] = v_i(\pi_n) \qquad \text{and} \qquad \mathbb{E}[\|U_{i,n}\|^2 \,|\, \mathcal{F}_n] \leq \frac{24A_i}{\zeta^4 \min_{s\in\mathcal{S},\alpha_i\in\mathcal{A}_i} \pi_{i,n}(\alpha_i|s)}. \tag{13}$$

In particular, this means that we will always have $B_n = 0$ for the bias of the estimator, but its variance could be unbounded if $\pi_n$ gets close to the boundary of $\Pi$. To avoid this, Reinforce can be paired with an explicit exploration step that modifies the sampling policy of the $n$-th episode to

$$\hat{\pi}_{i,n} = (1 - \varepsilon_n)\pi_{i,n} + \varepsilon_n \,\text{Unif}_{\mathcal{A}_i} \quad \text{for all } s \in \mathcal{S} \tag{14}$$

i.e., $\hat{\pi}_{i,n}$ is the mixture between $\pi_{i,n}$ and the uniform distribution $\text{Unif}_{\mathcal{A}_i}$ over $\mathcal{A}_i$. The resulting algorithm is known as $\varepsilon$-Greedy Policy Gradient; for a pseudocode representation, see Algorithm 2.

Importantly, by calling Reinforce at $\hat{\pi}_n$ instead of $\pi_n$, $\hat{v}_n$ becomes biased (because of the difference between $\hat{\pi}_n$ and $\pi_n$), but its variance is bounded; in particular, by invoking Lemma 4, we have

$$\mathbb{E}[\|b_{i,n}\| \,|\, \mathcal{F}_n] \leq G\varepsilon_n \qquad \text{and} \qquad \mathbb{E}[\|U_{i,n}\|^2 \,|\, \mathcal{F}_n] \leq \frac{24A_i^2}{\varepsilon_n \zeta^4} \tag{15}$$

where $G$ is a constant that depends on the smoothness of $V$ and the cardinalities of $\mathcal{A}$ and $\mathcal{S}$.[2] In this way, Algorithm 2 can be seen as a special case of (PG) with $B_n = \mathcal{O}(\varepsilon_n)$ and $\sigma_n^2 = \mathcal{O}(1/\varepsilon_n)$. ♦

## 4 Convergence analysis and results

We are now in a position to state and discuss our main results. For convenience, we will present our results in order of increasing structure, starting with stable policies, and then moving on to second-order stationary and deterministic Nash policies. All proofs are deferred to the appendix.

**4.1. Asymptotic convergence to stable Nash policies.** Our first convergence result concerns Nash policies that satisfy the stability requirement $\langle v(\pi), \pi - \pi^* \rangle < 0$ of Definition 2. In this case, we have the following guarantee:

**Theorem 1.** *Let $\pi^*$ be a stable Nash policy, and let $\pi_n$ be the sequence of play generated by (PG) with step-size $\gamma_n = \gamma/(n + m)^p$, $p \in (1/2, 1]$, and policy gradient estimates such that $p + \ell_b > 1$ and $p - \ell_\sigma > 1/2$ as per (8). Then there exists a neighborhood $\mathcal{U}$ of $\pi^*$ in $\Pi$ such that, for any given $\delta > 0$, we have*

$$\mathbb{P}(\pi_n \text{ converges to } \pi^* \,|\, \pi_1 \in \mathcal{U}) \geq 1 - \delta \tag{16}$$

*provided that $\gamma$ is small enough (or $m$ large enough) relative to $\delta$.*

---

[2]Specifically, from **??** we know that $\|v_i(\hat{\pi}_n) - v_i(\pi_n)\| \leq 3\sqrt{A}/\zeta^3 \cdot \sum_j \sqrt{A_j} \cdot \|\hat{\pi}_{j,n} - \pi_{j,n}\|$. Moreover, $|\pi_{i,n}(\alpha \mid s) - \hat{\pi}_{i,n}(\alpha \mid s)| \leq \varepsilon_n$ for all $s \in \mathcal{S}, \alpha \in \mathcal{A}_i$, so $\|\pi_{i,n} - \hat{\pi}_{i,n}\| \leq \sqrt{SA_i}\varepsilon_n$. Combining the above, it follows that we can take $G = 3NA^{3/2}\sqrt{S}/\zeta^3$.

**Corollary 1.** *Suppose that Models 1–3 are run with a step-size of the form $\gamma_n = \gamma/(n+m)^p$, $p > 1/2$, and if applicable, an exploration parameter $\varepsilon_n = \varepsilon/(n+m)^{\ell_\varepsilon}$ such that $1 - p < \ell_\varepsilon < 2p - 1$. Then:*

- *For Models 1 and 2: the conclusions of Theorem 1 hold as stated.*

- *For Model 3: the conclusions of Theorem 1 hold as long as $p > 2/3$.*

We note here that Theorem 1 provides a trajectory convergence guarantee which is otherwise quite difficult to obtain even in structured stochastic games. For example, if we zoom in on the class of stochastic potential (or min-max) games, the existing guarantees in the literature concern the "best iterate" of the algorithm, cf. [31, 61] and references therein. Because of this, said guarantees do not apply to the actual trajectory of play generated by (PG); this makes them less suitable for agent-based learning where the players involved are learning "as they go", as opposed to *simulating* the game in order to approximately compute an equilibrium policy offline.

We should also note that the convergence guarantees of Theorem 1 hold locally with arbitrarily high probability. Without further assumptions, it is not possible to obtain global trajectory convergence guarantees that hold with probability 1, even in single-state games – that is, the case of learning in finite normal form games. The reason for this locality is twofold: First, equilibrium policies are not unique in general, and gradient-based dynamics may also admit non-equilibrium attractors, such as limit cycles and the like [23, 34–36]. As a result, in the presence of multiple equilibria/attractors, the best one can hope for is a local equilibrium convergence result, conditioned on the basin of attraction of said equilibrium (as per Theorem 1).

The second obstruction to a global, unconditional convergence result is probabilistic in nature, and has to do with the randomness that enters the learning process (e.g., in the estimation of policy gradients via the REINFORCE). In this case, no matter how close one starts to an equilibrium policy, there is always a finite, non-zero probability that an unlucky realization of the noise can drive the process away from its basin, possibly never to return. This issue can only be overcome in games where $\Pi$ is partitioned (up to a set of measure zero) into basins of attraction of equilibrium policies. However, this can only occur in games with a sufficiently strong global structure, like potential stochastic games, two-player zero-sum games and the like; in complete generality, locality cannot be lifted, even in single-state problems [17, 19].

### 4.2. Convergence to second-order stationary policies.

Albeit valuable as an asymptotic convergence guarantee, Theorem 1 does not provide an indication of how long it will take players to actually converge to a Nash policy. Of course, in full generality, it is not plausible to expect to be able to derive such a convergence rate because the stability requirement provides no indication on how fast the players' policy gradients stabilize near a solution. This kind of estimate is provided by the second-order sufficient condition (SOS), which allows us to establish sufficient control over the sequence of play as indicated by the following theorem.

**Theorem 2.** *Let $\pi^*$ be a second-order stationary policy, let $\mathcal{B}$ be a neighborhood of $\pi^*$ such that (3a) holds on $\mathcal{B}$, and let $\pi_n$ be the sequence of play generated by (PG) with step-size $\gamma_n = \gamma/(n+m)^p$, $p \in (1/2, 1]$, and policy gradient estimates such that $p + \ell_b > 1$ and $p - \ell_\sigma > 1/2$ as per (8). Then:*

1. *There exists a neighborhood $\mathcal{U}$ of $\pi^*$ in $\Pi$ such that, for any confidence level $\delta > 0$, the event*

$$\mathcal{E} = \{\pi_n \in \mathcal{B} \text{ for all } n = 1, 2, \dots\} \tag{17}$$

   *occurs with probability $\mathbb{P}(\mathcal{E} \mid \pi_1 \in \mathcal{U}) \geq 1 - \delta$ if $m$ is large enough relative to $\delta$.*

2. *The sequence $\pi_n$ converges to $\pi^*$ with probability 1 on $\mathcal{E}$; in particular, we have*

$$\mathbb{P}(\pi_n \text{ converges to } \pi^* \mid \pi_1 \in \mathcal{U}) \geq 1 - \delta \tag{18}$$

   *if $m$ is large relative to $\delta$. Moreover, conditioned on $\mathcal{E}$ and taking $q = \min\{\ell_b, p - 2\ell_\sigma\}$, we have*

$$\mathbb{E}[\|\pi_n - \pi^*\|^2 \mid \mathcal{E}] = \begin{cases} \mathcal{O}(1/n^{2\mu\gamma}) & \text{if } p = 1 \text{ and } 2\mu\gamma < q, \\ \mathcal{O}(1/n^q) & \text{otherwise.} \end{cases} \tag{19}$$

**Corollary 2.** *Suppose that Models 1–3 are run with a step-size of the form $\gamma_n = \gamma/(n+m)^p$, $p > 1/2$, and if applicable, an exploration parameter $\varepsilon_n = \varepsilon/(n+m)^{p/2}$. Then:*

- *For [Models 1] and [2]: the conclusions of [Theorem 2] hold with $q = p$; in particular, [(19)] gives an $\mathcal{O}(1/n)$ rate of convergence if $p = 1$ and $2\mu\gamma > q$.*

- *For [Model 3]: the conclusions of [Theorem 2] hold for $p > 2/3$ with $q = p/2$; in particular, [(19)] gives an $\mathcal{O}(1/\sqrt{n})$ rate of convergence if $p = 1$ and $2\mu\gamma > q$.*

*Remark* 2. Getting an explicit estimate for the constant in the $\mathcal{O}(\cdot)$ guarantee of [Theorem 2] is quite involved but, up to logarithmic and subleading factors, Chung's lemma [10, 43] can be used to show that *a)* if $2\mu\gamma > q$, it scales as $(C_b + C_\sigma)/[(2\mu\gamma - q)(1 - \delta)]$ where $C_b = \sup_n \gamma_n B_n$ and $C_\sigma = \sup_n \gamma_n^2 \sigma_n^2$; *b)* if $2\mu\gamma = q$, it scales as $(C_b + C_\sigma)(1 + \max\{(2\mu\gamma)^2, 4\mu\gamma\})/(1 - \delta)$; and *c)* if $2\mu\gamma < q$ as $(C_b + C_\sigma)(1 + \max\{(2\mu\gamma)^2, 4\mu\gamma\})/[(q - 2\mu\gamma)(1 - \delta)]$.

Besides providing a general framework for achieving trajectory convergence, [Theorem 2] gives the rates of convergence of the sequence of play to the Nash policy in question. In particular, with this result in hand, one can confidently argue about the distance of the iterates of [(PG)] from equilibrium in a series of different environments. More to the point, this convergence guarantee allows the algorithm designer to adapt the parameters of the learning process according to the complexity and limitations of the environment, a feature which further highlights the significance of this result.

We should also note the delicate interplay between the method's step-size and the achieved convergence rate. In the case of [Model 1], [Corollary 2] suggests a step-size of the form $\gamma_n = \Theta(1/n)$, leading to a $\mathcal{O}(1/n)$ convergence rate. As we show in the appendix, this rate can be improved: in the deterministic case with perfect gradient information, [(PG)] with a suitably chosen constant step-size achieves a *geometric* convergence rate, i.e., $\|\pi_n - \pi^*\| = \mathcal{O}(\exp(-\rho n))$ for some $\rho > 0$ (cf. **??** in **??**). By contrast, in the case of [Model 2], the $\mathcal{O}(1/n)$ rate we provide cannot be improved, even if the quadratic minorant [(3a)] that characterizes SOS policies holds *globally* – and this because the learning process is running against standard lower bounds from convex optimization [6, 40].

Perhaps the most significant guarantee from a practical point of view is the $\mathcal{O}(1/\sqrt{n})$ convergence rate attained in [Model 3] (cf. [Algorithms 1] and [2]). This guarantee amounts to a $\mathcal{O}(1/n^{1/4})$ convergence rate in terms of the (non-squared) distance to equilibrium which, mutatis mutandis, represents a notable improvement over the $\mathcal{O}(1/n^{1/6})$ guarantee of Leonardos et al. [31] (expressed in norm values). Of course, the latter guarantee is global – because the focus of [31] is stochastic *potential* games – but it also concerns the "best iterate" of the process (not its "last iterate"), so the two results are not immediately comparable. However, a useful "best-of-both-worlds" heuristic that can be inferred by the combination of these works is that, given a budget of training episodes, [Algorithm 2] can be run with a constant step-size as per [31] for a sufficient fraction of this budget, and then with a $\mathcal{O}(1/n)$ "cooldown" schedule for the rest. In this way, after an aggressive "exploration" phase, the algorithm's $\mathcal{O}(1/n^{1/4})$ rate would kick in and supply faster stabilization to an SOS policy.

**4.3. Convergence to deterministic Nash policies.** Our last series of results concerns the rate of convergence to deterministic Nash policies in generic stochastic games. As we discussed in [Section 2], deterministic Nash policies also satisfy [(SOS)], so the rate of convergence of [(PG)] to such policies can be harvested directly from [Theorem 2]. However, as we show below, a simple projection tweak in [(SOS)] can improve this rate dramatically.

The tweak in question is inspired by the geometry of $\Pi$ around a deterministic policy: by definition, such policies are corner points of $\Pi$, so any consistent drift towards them will cause $\pi_n$ to hit the boundary of $\Pi$ in finite time. Of course, under [(PG)], the process may rebound from the boundary and return to the interior of $\Pi$ if the policy gradient estimate is not particularly good at a given iteration of the algorithm. However, if we replace the projection step of [(PG)] with a "lazy projection" in the spirit of Zinkevich [62], the aggregation of gradient steps will eventually push the process far inside the normal cone of $\Pi$ at $\pi^*$, so rebounds of this type can no longer occur.

Formally, we will consider the following *lazy policy gradient* (LPG) scheme:

$$y_{n+1} = y_n + \gamma_n \hat{v}_n \qquad \pi_{n+1} = \mathrm{proj}_\Pi(y_{n+1}) \tag{LPG}$$

where $y_n = (y_{i,n})_{i \in \mathcal{N}} \in \prod_i \mathbb{R}^{\mathcal{A}_i \times \mathcal{S}}$ is an auxiliary variable that maintains an aggregate of gradient steps *before* projecting them back to $\Pi$. We then have the following convergence result:

**Theorem 3.** *Let $\pi_n$ be the sequence of play under [(LPG)] with step-size and policy gradient estimates such that $p + \ell_b > 1$ and $p - \ell_\sigma > 1/2$ as per [(8)]. If $\pi^*$ is a deterministic Nash policy, there exists an unbounded open set $\mathcal{W} \subseteq \prod_i \mathbb{R}^{\mathcal{A}_i \times \mathcal{S}}$ of initializations such that, for any $\delta > 0$, we have*

$$\mathbb{P}(\pi_n \text{ converges to } \pi^* \mid y_1 \in \mathcal{W}) \geq 1 - \delta, \tag{20}$$

*provided that $\gamma > 0$ is small enough. Moreover, conditioned on this event, $\pi_n$ converges to $\pi^*$ at a finite number of iterations, i.e., there exists some $n_0$ such that $\pi_n = \pi^*$ for all $n \geq n_0$.*

**Corollary 3.** *Suppose that Models 1–3 are run with parameters $\gamma_n = \gamma/n^p$, $p \in (1/2, 1]$, and if applicable, $\varepsilon_n = \varepsilon/n^{\ell_\varepsilon}$ with $1 - p < \ell_\varepsilon < 2p - 1$. Then the conclusions of Theorem 3 hold.*

*Remark* 3. Getting an explicit bound for $n_0$ is quite complicated, but the last part of the proof of Theorem 3 shows that $n_0$ scales in terms of the parameters of the game and the algorithm as $n_0 = \mathcal{O}\left(\left(\frac{MSA}{c\gamma}\right)^{1/(1-p)}\right)$ where $c > 0$ measures the minimum payoff difference between equilibrium and non-equilibrium strategies at $\pi^*$, $M$ is a measure of the initial distance from $\pi^*$, and $S$ and $A$ is the number of states and pure strategies respectively.

Theorem 3 – and, by extension, Corollary 3 – are fairly unique because they provide a guarantee for convergence to an *exact* Nash equilibrium in a *finite* number of iterations. To the best of our knowledge, the only comparable result in the literature is that of [61], where the authors provide a finite-time convergence guarantee to strict equilibria with *perfect* policy gradients (as per Model 1). The result of Zhang et al. [61] echoes the convergence properties of deterministic first-order algorithms around sharp minima of convex functions [43], but the fact that Theorem 3 applies to models with *stochastic* gradient feedback of *unbounded* variance (Models 2 and 3 respectively) is a major difference. As far as we are aware, this is the first guarantee of its kind in the literature on learning in stochastic games.

## 5   Concluding remarks

A key roadblock encountered by practical applications of multi-agent reinforcement learning is the lack of universal equilibrium convergence guarantees. While the impossibility results of [21, 22] imply that unconditional convergence is not a reasonable aspiration without further assumptions on the game, the existence of local convergence results mitigates this deficiency as it provides a range of theoretically grounded stability and runtime guarantees. In this regard, deterministic policies acquire particular importance, as the convergence of policy gradient methods is especially rapid and robust and this case. Of course, this leaves open the question of non-tabular settings and parametrically encoded policies, e.g., as in the case of deep reinforcement learning; we defer these investigations to future work.

Another open issue of high practical relevance concerns policy gradient methods that do not rely on Euclidean projections to Π. In the single-state case (i.e., learning in finite normal form games), the use of methods relying on softmax choice / exponential weights is very widely used because of its regret guarantees. Whether the use of similar softmax techniques can lead to finer convergence guarantees in the context of general stochastic games is an important and intriguing question for future research.

## Acknowledgments and Disclosure of Funding

Part of this work was done while the authors were visiting the Simons Institute for the Theory of Computing. P. Mertikopoulos gratefully acknowledges financial support by the French National Research Agency (ANR) in the framework of the "Investissements d'avenir" program (ANR-15-IDEX-02), the LabEx PERSYVAL (ANR-11-LABX-0025-01), MIAI@Grenoble Alpes (ANR-19-P3IA-0003), and the bilateral ANR-NRF grant ALIAS (ANR-19-CE48-0018-01). K. Lotidis and E. Vlatakis are grateful for financial support by the Onassis Foundation (F ZR 033-1/2021-2022, 010-1/2018-2019). A. Giannou is grateful for financial support by ONR: "A Theoretically Principled Framework for Learning by Pruning". E. V. Vlatakis-Gkaragkounis is grateful for financial support by the Google-Simons Fellowship, Pancretan Association of America and Simons Collaboration on Algorithms and Geometry. This project was completed while he was a visiting research fellow at the Simons Institute for the Theory of Computing. Additionally, he would like to acknowledge the following series of NSF-CCF grants under the numbers 1763970/2107187/1563155/1814873.

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
