# On the convergence of policy gradient methods to Nash equilibria in general stochastic games

## Abstract

Multi-agent learning in stochastic $N$-player games is a notoriously difficult problem because, in addition to their changing strategic decisions, the players of the game must also contend with the fact that the game itself evolves over time, possibly in a very complicated manner. Because of this, the equilibrium convergence properties of popular learning algorithms – like policy gradient and its variants – are poorly understood, except in specific classes of games (such as potential or two-player, zero-sum games). In view of all this, we examine the long-run behavior of policy gradient methods with respect to Nash equilibrium policies that are second-order stationary (SOS) in a sense similar to the type of KKT sufficiency conditions used in optimization. Our analysis shows that SOS policies are locally attracting with high probability, and we show that policy gradient trajectories with gradient estimates provided by the REINFORCE algorithm achieve an $\mathcal{O}(1/\sqrt{n})$ convergence rate to such equilibria if the method's step-size is chosen appropriately. On the other hand, when the equilibrium in question is *deterministic*, we show that this rate can be improved dramatically and, in fact, policy gradient methods converge within a *finite* number of iterations in that case.

## 1 Introduction

Ever since they were introduced by Shapley [51] in the 1950's, stochastic games have comprised one of the staples of non-cooperative game theory, with a range of pioneering applications to multi-agent reinforcement learning [8, 28, 65], unmanned vehicles [11, 35, 48, 50, 62], general game-playing [6, 7, 38, 52, 58], etc. Informally, a stochastic game evolves in discrete time as follows: At each point in time, the players are at a given state which determines the rules of the game for that stage. The actions of the players in this state determine not only their instantaneous payoffs (as defined by the stage game), but also the transition probabilities towards the next state of the process. In this way, each player has to balance two distinct – and often competing – objectives: optimizing the payoffs of *today* versus picking a possibly suboptimal action which could yield significant benefits *tomorrow* (i.e., by influencing the transitions of the process towards a more favorable state for the player).

Since all players in the game are involved in a similar dilemma, the decision-making problem for each player is a very complicated affair. In particular, in addition to their changing strategic decisions, the players of the game must also contend with the fact that the game itself evolves over time. Because of this, even the existence of a Nash equilibrium policy – viz. a stationary Markovian policy that is stable to unilateral deviations [20] – is far more difficult to prove compared to standard, stateless normal form games; for a comprehensive survey, see [42, 53, 67] and references therein.

The question we seek to address in this paper is whether an ensemble of boundedly rational players can reach an equilibrium policy in a stochastic game. Specifically, if players do not have sufficient information – or the computational resources required – to solve a Bellman equation in very high

Submitted to 36th Conference on Neural Information Processing Systems (NeurIPS 2022). Do not distribute.

dimensions [55, 59], it is not at all clear if they would somehow end up playing a Nash policy in the long run. After all, the complexity of most games increases exponentially with the number of players, so the identification of a game's equilibria quickly becomes prohibitively difficult [17, 29, 34, 36].

**Our contributions in the context of related work.** This issue has sparked a vigorous literature with important implications for the series of applications mentioned above [3, 54, 64]. On the downside, these efforts also have to grapple with a series of strong lower bounds for computing weaker solution concepts like coarse correlated equilibria in turn-based stochastic games [16, 29]. On that account, a recent line of work has instead focused on understanding specific sub-classes of stochastic games, like *min-max* [12, 15, 49, 60] and common interest *potential* games [18, 33, 68], or computing relaxed solution concepts where either the stationarity or the Markov property has been dropped [16].

Our paper focuses on episodic playing in random stopping games – in lieu of learning in ergodic stochastic games with an infinite horizon [34, 44] – and considers the general class of policy gradient methods, first introduced by [30, 31, 56, 61] and subsequently popularized in single-agent reinforcement learning by [2, 10, 27, 63]. Concretely, this means that the sequence of play evolves episode-by-episode: within each episode, the players commit a policy and play the game, and from one episode to the next, they use an iterative gradient step to update their policy and continue playing.

Our main contributions in this general context may then be summarized as follows:

1. We introduce a flexible algorithmic template for the analysis of policy gradient methods which accounts for different information and update frameworks – from perfect policy gradients to value-based estimates obtained per episode, e.g., via the REINFORCE algorithm [4, 56, 61].

2. Within this framework, we show that Nash policies that satisfy a certain strategic stability condition are locally attracting with arbitrarily high probability. Moreover, to estimate the method's rate of convergence, we focus on Nash policies that satisfy a second-order sufficiency condition similar to the type of KKT conditions used in optimization, and we show that such policies enjoy an $\mathcal{O}(1/\sqrt{n})$ convergence rate in terms of squared distance.

3. Finally, we also consider the method's convergence to *deterministic* Nash policies and we show that, generically, the above rate can be improved dramatically. By a simple tweak to the method's projection step, we are able to show that the induced sequence of play converges to equilibrium in a *finite* number of iterations, despite all the noise and uncertainty in the process.

It is worth mentioning that our results focus squarely on the convergence of the actual, inter-episode trajectory of play – as opposed to "best-iterate" or ergodic convergence results. In addition, obtaining guarantees using stochastic estimators (cf. REINFORCE) greatly alleviate the burden of exact gradient computations that are otherwise beyond reach in low-compute / low-memory practical environments.

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

Thus, if REINFORCE is executed at $\hat{\pi} \leftarrow \pi_n$ at each episode $n = 1, 2, \ldots$, we will have

$$\mathbb{E}[\hat{v}_{i,n}] = v_i(\pi_n) \qquad \text{and} \qquad \mathbb{E}[\|U_{i,n}\|^2 \mid \mathcal{F}_n] \leq \frac{24|\mathcal{A}_i|}{\zeta^4 \min_{s \in \mathcal{S}, \alpha_i \in \mathcal{A}_i} \pi_{i,n}(\alpha_i|s)}. \tag{13}$$

This means that we will always have $B_n = 0$ for the bias of the estimator, but its variance could be unbounded if $\pi_n$ gets close to the boundary of $\Pi$. For this reason, REINFORCE is typically paired with an explicit exploration step that modifies the sampling policy of the $n$-th episode to

$$\hat{\pi}_{i,n} = (1 - \varepsilon_n)\pi_{i,n} + \varepsilon_n \operatorname{Unif}_{\mathcal{A}_i}. \tag{14}$$

i.e., $\hat{\pi}_{i,n}$ is the mixture between $\pi_{i,n}$ and the uniform distribution $\operatorname{Unif}_{\mathcal{A}_i}$ over $\mathcal{A}_i$. The resulting algorithm is known as $\varepsilon$-GREEDY POLICY GRADIENT; for a pseudocode, see Algorithm 2.

Importantly, by calling REINFORCE at $\hat{\pi}_n$, $\hat{v}_n$ becomes biased (because of the difference between $\hat{\pi}_n$ and $\pi_n$), but its variance is bounded; in particular, by invoking Lemma 4, we have

$$\mathbb{E}[\|b_{i,n}\| \mid \mathcal{F}_n] \leq G\varepsilon_n \qquad \text{and} \qquad \mathbb{E}[\|U_{i,n}\|^2 \mid \mathcal{F}_n] \leq \frac{24|\mathcal{A}_i|^2}{\varepsilon_n \zeta^4} \tag{15}$$

where $G$ is a constant that depends on the smoothness of $V$ and the cardinalities of $\mathcal{A}$ and $\mathcal{S}$. In this way, Algorithm 2 can be seen as a special case of (PG) with $B_n = \mathcal{O}(\varepsilon_n)$ and $\sigma_n = \mathcal{O}(1/\sqrt{\varepsilon_n})$. ¶

# 4 Convergence analysis and results

We are now in a position to state and discuss our main results. For convenience, we will present our results in order of increasing structure, starting with stable policies, and then moving on to second-order stationary and deterministic Nash policies. All proofs are deferred to the appendix.

**4.1. Asymptotic convergence to stable Nash policies.** Our first convergence result concerns Nash policies that satisfy the stability requirement $\langle v(\pi), \pi - \pi^* \rangle < 0$ of Definition 2. In this case, we have the following guarantee:

**Theorem 1.** *Let $\pi^*$ be a stable Nash policy, and let $\pi_n$ be the sequence of play generated by (PG) with step-size $\gamma_n = \gamma/(n + m)^p$, $p \in (1/2, 1]$, and policy gradient estimates such that $p + \ell_b > 1$ and $p - \ell_\sigma > 1/2$ as per (8). Then there exists a neighborhood $\mathcal{U}$ of $\pi^*$ in $\Pi$ such that, for any given $\delta > 0$, we have*

$$\mathbb{P}(\pi_n \text{ converges to } \pi^* \mid \pi_1 \in \mathcal{U}) \geq 1 - \delta \tag{16}$$

*provided that $\gamma$ is small enough (or $m$ large enough) relative to $\delta$.*

**Corollary 1.** *Suppose that Models 1–3 are run with a step-size of the form $\gamma_n = \gamma/(n + m)^p$, $p > 1/2$, and if applicable, an exploration parameter $\varepsilon_n = \varepsilon/(n + m)^r$ such that $1 - p < r < 2p - 1$. Then:*

- *For Models 1 and 2: the conclusions of Theorem 1 hold as stated.*

- *For Model 3: the conclusions of Theorem 1 hold as long as $p > 2/3$.*

We note here that Theorem 1 provides a trajectory convergence guarantee which is otherwise quite difficult to obtain even in structured stochastic games. For example, if we zoom in on the class of stochastic potential (or min-max) games, the existing guarantees in the literature concern the "best iterate" of the algorithm, cf. [33, 68] and references therein. Because of this, said guarantees do not apply to the actual trajectory of play generated by (PG); this makes them less suitable for agent-based learning where the players involved are learning "as they go", as opposed to *simulating* the game in order to approximately compute an equilibrium policy offline.

We should also note that the convergence guarantees of Theorem 1 hold locally with arbitrarily high probability. Without further assumptions, it is not possible to obtain global trajectory convergence guarantees that hold with probability 1, even in the simple case where the game only has a single state – that is, the case of learning in finite normal form games. In this (much simpler) setting, the

well-known impossibility result of Hart and Mas-Colell [24, 25] shows that it is not possible to expect convergence to Nash equilibrium in all games – not even locally. In this regard, the local convergence caveat in Theorem 1 cannot be lifted without further structural properties in place – such as the existence of a potential function in the spirit of [33].

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

# Organization of the appendix

| Notation | Description |
|---|---|
| $s \in \mathcal{S}$ | States of the game |
| $\alpha_i \in \mathcal{A}_i$ | Actions of agent $i \in \mathcal{N}$ |
| $\tau$ | Episode trajectory |
| $T(\tau)$ | Episode stopping time |
| $\zeta$ | Minimum stopping probability |
| $r_{i,t}$ | Realized reward of $i$-th player at time $t$ |
| $\gamma_n$ | Step size at episode $n$ |
| $\varepsilon_n$ | Explicit exploration parameter at episode $n$ |
| $v(\pi_n)$ | Policy gradients at policy $\pi_n$ of episode $n$ |
| $\hat{v}_n$ | Policy gradient proxy at episode $n$. |

**Table 1:** Index of the most common notations used in our paper.

## A  Errata and omissions

When preparing the supplementary material of our paper, we noticed a number of typographic errors and omissions in the main paper that could possibly cause confusion. We clarify those below:

- L48: The reference pointers should point to Perkins [44] and Leslie et al. [34].
- L157: (NE) should read (FOS)
- L166: Only the one-way implication is relevant; Proposition 1 was amended accordingly.
- L188: The text should read $\gamma_n = \gamma/(n + m)^p$ for some $\gamma > 0$, $m \geq 0$ and $p \geq 0$.
- L246: The text of Theorem 1 was amended to explicitly include the above clarification.
- L251–L252: the relation "$1 - p < r/2 < p - 1/2$" should read "$1 - p < r < 2p - 1$".
- L250, L283: "$\varepsilon_n = \varepsilon/n^r$" should read "$\varepsilon_n = \varepsilon/(n + m)^r$" and "$\varepsilon_n = \varepsilon/(n + m)^{p/2}$" respectively.
- L331, Eq. (20): "$\mathcal{U}$" should read "$\mathcal{W}$"
- L125, the ~~minimax~~ *mismatch coefficient* can be defined either as $\mathcal{C}_{\mathcal{G}} := \max_{\pi,\pi' \in \Pi}\{\|\tilde{d}_\rho^\pi / \tilde{d}_\rho^{\pi'}\|_\infty\}$ or simpler. $\mathcal{C}_{\mathcal{G}} := \max_{\pi, \in \Pi}\{\frac{1}{\zeta}\|d_\rho^\pi/\rho\|_\infty\}$.

The errata and omissions identified above have all been corrected in the file at hand.

## B  Asymptotic convergence to stable Nash policies

Our goal in this appendix is to prove Theorem 1 and Corollary 1, which we restate below for convenience:

**Theorem 1.** *Let $\pi^*$ be a stable Nash policy, and let $\pi_n$ be the sequence of play generated by (PG) with step-size $\gamma_n = \gamma/(n + m)^p$, $p \in (1/2, 1]$, and policy gradient estimates such that $p + \ell_b > 1$ and $p - \ell_\sigma > 1/2$ as per (8). Then there exists a neighborhood $\mathcal{U}$ of $\pi^*$ in $\Pi$ such that, for any given $\delta > 0$, we have*

$$\mathbb{P}(\pi_n \text{ converges to } \pi^* \mid \pi_1 \in \mathcal{U}) \geq 1 - \delta \tag{16}$$

*provided that $\gamma$ is small enough (or $m$ large enough) relative to $\delta$.*

**Corollary 1.** *Suppose that Models 1–3 are run with a step-size of the form $\gamma_n = \gamma/(n + m)^p$, $p > 1/2$, and if applicable, an exploration parameter $\varepsilon_n = \varepsilon/(n + m)^r$ such that $1 - p < r < 2p - 1$. Then:*

- *For Models 1 and 2: the conclusions of Theorem 1 hold as stated.*
- *For Model 3: the conclusions of Theorem 1 hold as long as $p > 2/3$.*

Our proof strategy will comprise the following basic steps:

1. To begin with, we will show that the squared distance

$$D(\pi) = \frac{1}{2}\|\pi - \pi^*\|^2 \tag{B.1}$$

   can be seen as a "local Lyapunov function" for (PG) in the sense that it is locally decreasing near $\pi^*$, up to a series of error terms – both zero-mean and non-zero-mean.

2. Due to these errors, the evolution of the iterates $D_n := D(\pi_n)$ of $D$ over time may exhibit *significant* jumps: in particular, a single "bad" realization of the noise could carry $\pi_n$ out of the basin of attraction of $\pi^*$, possibly never to return. To exclude this event, our second step will be to show that the aggregation of these errors can be controlled with probability at least $1 - \delta$.

3. Conditioned on the above, we will show that, with probability at least $1 - \delta$, the iterates $D_n$ cannot grow more than a token value. As a result, if (PG) is initialized close to $\pi^*$, it will remain in a neighborhood thereof for all $n$ (again, with probability at least $1 - \delta$).

4. Thanks to this "stochastic Lyapunov stability" result, we employ a series of martingale limit theory arguments to extract a subsequence converging to $\pi^*$.

607     5. Finally, we show that the increments of $D_n$ are summable; hence, by invoking the Gladyshev's
608       lemma [45, p. 49], we conclude that $D_n$ converges to some (finite) random variable $D_\infty$. Combin-
609       ing this fact with the existence of a convergent subsequence, we obtain the desired conclusion
610       that $\pi_n$ converges to $\pi^*$ with probability at least $1 - \delta$.

611 In the sequel, we make the above precise in a series of intermediate results.

612 **B.1. Energy inequality.** We begin by establishing a "quasi-Lyapunov" inequality for the iterates
613 $D_n = \|\pi_n - \pi^*\|^2/2$ of (B.1).

614 **Lemma B.1.** *Let $D_n := D(\pi_n)$. Then, for all $n = 1, 2, \ldots,$ we have*

$$D_{n+1} \le D_n + \gamma_n \langle v(\pi_n), \pi_n - \pi^* \rangle + \gamma_n \xi_n + \gamma_n \chi_n + \gamma_n^2 \psi_n^2, \tag{B.2}$$

615 *where the error terms $\xi_n$, $\chi_n$, and $\psi_n$ are given by*

$$\xi_n = \langle U_n, \pi_n - \pi^* \rangle, \quad \chi_n = \|\Pi\| B_n \quad and \quad \psi_n^2 = \tfrac{1}{2}\|\hat{v}_n\|^2. \tag{B.3}$$

616 *with $\|\Pi\| := \max_{\pi, \pi' \in \Pi} \|\pi - \pi'\|$.*

617 *Proof.* By the definition of the iterates of (PG), we have

$$\begin{aligned}
D_{n+1} = \frac{1}{2}\|\pi_{n+1} - \pi^*\|^2 &= \frac{1}{2}\|\mathrm{proj}_\Pi(\pi_n + \gamma_n \hat{v}_n) - \mathrm{proj}_\Pi(\pi^*)\|^2 \\
&\le \frac{1}{2}\|\pi_n + \gamma_n \hat{v}_n - \pi^*\|^2 \\
&= \frac{1}{2}\|\pi_n - \pi^*\|^2 + \gamma_n \langle \hat{v}_n, \pi_n - \pi^* \rangle + \frac{1}{2}\gamma_n^2 \|\hat{v}_n\|^2 \\
&= D_n + \gamma_n \langle v(\pi_n) + U_n + b_n, \pi_n - \pi^* \rangle + \frac{1}{2}\gamma_n^2 \|\hat{v}_n\|^2 \\
&\le D_n + \gamma_n \langle v(\pi_n), \pi_n - \pi^* \rangle + \gamma_n \xi_n + \gamma_n \chi_n + \gamma_n^2 \psi_n^2 \tag{B.4}
\end{aligned}$$

618 where we used the Cauchy-Schwarz inequality to bound the bias term as $\langle b_n, \pi_n - \pi^* \rangle \le \|b_n\| \cdot \|\pi_n -$
619 $\pi^*\| \le \|\Pi\| B_n = \chi_n$.     ■

620 **B.2. Error control and stability.** The second major step in our proof (and the most challenging one
621 from a technical standpoint) is to establish a suitable measure of control over the error increments in
622 (B.1), with the aim of showing that the process $\pi_n$ never leaves a neighborhood of $\pi^*$.

623 To make this idea precise, let $\mathcal{B} = \{\pi \in \Pi : \|\pi - \pi^*\| \le r\}$ be a ball of radius $r$ based on $\pi^*$ in $\Pi$ so that
624 $\langle v(\pi), \pi - \pi^* \rangle < 0$ for all $\pi \in \mathcal{B} \backslash \{\pi^*\}$ (without loss of generality, we can assume that $\mathcal{B}$ is maximal in
625 that regard). We will then examine the event that the aggregation of the error terms in (B.1) is not
626 sufficient to drive $\pi_n$ to escape from $\mathcal{B}$.

627 To that end, we will begin by aggregating the errors in (B.1) as

$$M_n = \sum_{k=1}^{n} \gamma_k \xi_k \quad \text{and} \quad S_n = \sum_{k=1}^{n} [\gamma_k \chi_k + \gamma_k^2 \psi_k^2]. \tag{B.5}$$

628 Since $\mathbb{E}[\xi_n \mid \mathcal{F}_n] = 0$, we have $\mathbb{E}[M_n \mid \mathcal{F}_n] = M_{n-1}$, so $M_n$ is a martingale; likewise, $\mathbb{E}[S_n \mid \mathcal{F}_n] \ge S_{n-1}$,
629 so $S_n$ is a submartingale. Then, using a technique of Hsieh et al. [26] that builds on an earlier idea by
630 Mertikopoulos and Zhou [37], we will also consider the "mean square" error process

$$R_n = M_n^2 + S_n, \tag{B.6}$$

631 and the associated indicator events

$$\mathcal{E}_n = \{\pi_k \in \mathcal{B} \text{ for all } k = 1, 2, \ldots, n\} \quad \text{and} \quad H_n = \{R_k \le a \text{ for all } k = 1, 2, \ldots, n\}, \tag{B.7a}$$

632 where, with a fair amount of hindsight, the error tolerance level $a > 0$ is such that $2a + \sqrt{a} < r$, and
633 we are employing the convention $\mathcal{E}_0 = H_0 = \Omega$ (since every statement is true for the elements of the
634 empty set). We will then assume that $\pi_1$ is initialized in a ball of radius $\sqrt{2a}$ centered at $\pi^*$, viz.

$$\mathcal{U} = \{\pi \in \Pi : D(\pi) \le a\} = \{\pi \in \Pi : \|\pi - \pi^*\|^2/2 \le a\}. \tag{B.8}$$

With all this in hand, the key to showing that $\pi_n$ remains close to $\pi^*$ with high probability is the following conditional estimate:

**Lemma B.2.** *Let $\pi_n$ be the sequence of play generated by (PG) initialized at $\pi_1 \in \mathcal{U}$. We then have:*

1. *$\mathcal{E}_{n+1} \subseteq \mathcal{E}_n$ and $H_{n+1} \subseteq H_n$ for all $n = 1, 2, \ldots$*

2. *$H_{n-1} \subseteq \mathcal{E}_n$ for all $n = 1, 2, \ldots$*

3. *Consider the "bad realization" event*

$$\tilde{H}_n := H_{n-1} \setminus H_n = \{R_k \le a \text{ for } k = 1, 2, \ldots, n-1 \text{ and } R_n > a\}, \tag{B.9}$$

*and let $\tilde{R}_n = R_n \mathbb{1}_{H_{n-1}}$ be the cumulative error subject to the noise being "small". Then we have:*

$$\mathbb{E}[\tilde{R}_n] \le \mathbb{E}[\tilde{R}_{n-1}] + \gamma_n \|\Pi\| B_n + \gamma_n^2 \|\Pi\|^2 \sigma_n^2 + \tfrac{3}{2}\gamma_n^2(G^2 + B_n^2 + \sigma_n^2) - a\,\mathbb{P}(\tilde{H}_{n-1}), \tag{B.10}$$

*where, by convention, $\tilde{H}_0 = \varnothing$ and $\tilde{R}_0 = 0$.*

*Remark.* In the above (and what follows), the notation $\mathbb{1}_A$ is used to indicate the logical indicator of an event $A \subseteq \Omega$, i.e., $\mathbb{1}_A(\omega) = 1$ if $\omega \in A$ and $\mathbb{1}_A(\omega) = 0$ otherwise.

The proof of Lemma B.2 is quite technical, so we first proceed to derive an important stability result based on this estimate.

**Proposition B.1.** *Fix some confidence threshold $\delta > 0$ and let $\pi_n$ be the sequence of play generated by (PG) with step-size and policy gradient estimates as per Theorem 1. We then have:*

$$\mathbb{P}(H_n \mid \pi_1 \in \mathcal{U}) \ge 1 - \delta \quad \text{for all } n = 1, 2, \ldots \tag{B.11}$$

*provided that $\gamma$ is small enough (or $m$ large enough) relative to $\delta$.*

*Proof.* We begin by bounding the probability of the "bad realization" event $\tilde{H}_n = H_{n-1} \setminus H_n$. Indeed, if $\pi_1 \in \mathcal{U}$, we have:

$$\mathbb{P}(\tilde{H}_n) = \mathbb{P}(H_{n-1} \setminus H_n) = \mathbb{E}[\mathbb{1}_{H_{n-1}} \times \mathbb{1}\{R_n > a\}] \le \mathbb{E}[\mathbb{1}_{H_{n-1}} \times (R_n/a)] = \mathbb{E}[\tilde{R}_n]/a \tag{B.12}$$

where, in the penultimate step, we used the fact that $R_n \ge 0$ (so $\mathbb{1}\{R_n > a\} \le R_n/a$). Telescoping (B.10) then yields

$$\mathbb{E}[\tilde{R}_n] \le \mathbb{E}[\tilde{R}_0] + \|\Pi\| \sum_{k=1}^n \gamma_k B_k + \sum_{k=1}^n \gamma_k^2 \varrho_k^2 - a \sum_{k=1}^n \mathbb{P}(\tilde{H}_{k-1}) \tag{B.13}$$

where we set

$$\varrho_n^2 = \|\Pi\|^2 \sigma_n^2 + \tfrac{3}{2}(G^2 + B_n^2 + \sigma_n^2). \tag{B.14}$$

Hence, combining (B.12) and (B.13) and invoking our stated assumptions for $\gamma_n$, $B_n$ and $\sigma_n$, we get

$$\sum_{k=1}^n \mathbb{P}(\tilde{H}_k) \le \frac{1}{a} \sum_{k=1}^n [\gamma_k B_k \|\Pi\| + \gamma_k^2 \varrho_k^2] \le \frac{C}{a} \tag{B.15}$$

for some $C \equiv C(\gamma, m) > 0$ with $\lim_{\gamma \to 0^+} C(\gamma, m) = \lim_{m \to \infty} C(\gamma, m) = 0$.

Now, by choosing $\gamma$ sufficiently small (or $m$ sufficiently large), we can ensure that $C/a < \delta$; thus, given that the events $\tilde{H}_k$ are disjoint for all $k = 1, 2, \ldots$, we get $\mathbb{P}(\bigcup_{k=1}^n \tilde{H}_k) = \sum_{k=1}^n \mathbb{P}(\tilde{H}_k) \le \delta$. In turn, this implies that $\mathbb{P}(H_n) = \mathbb{P}(\tilde{H}_1^c \cap \cdots \cap \tilde{H}_n^c) \ge 1 - \delta$, and our assertion follows. $\blacksquare$

We conclude this appendix with the proof of our technical result on the events $\mathcal{E}_n$ and $H_n$:

*Proof of Lemma B.2.* The first claim of the lemma is obvious. For the second, we proceed inductively:

1. For the base case $n = 1$, we have $\mathcal{E}_1 = \{\pi_1 \in \mathcal{B}\} \supseteq \{\pi_1 \in \mathcal{U}\} = \Omega$ (recall that $\pi_1$ is initialized in $\mathcal{U} \subseteq \mathcal{B}$). Since $H_0 = \Omega$, our claim follows.

664  2. Inductively, assume that $H_{n-1} \subseteq \mathcal{E}_n$ for some $n \geq 1$. To show that $H_n \subseteq \mathcal{E}_{n+1}$, suppose that
665  $R_k \leq a$ for all $k = 1, 2, \ldots, n$. Since $H_n \subseteq H_{n-1}$, this implies that $\mathcal{E}_n$ also occurs, i.e., $\pi_k \in \mathcal{B}$ for
666  all $k = 1, 2, \ldots, n$; as such, it suffices to show that $\pi_{n+1} \in \mathcal{B}$. To do so, given that $\pi_k \in \mathcal{U} \subseteq \mathcal{B}$
667  for all $k = 1, 2, \ldots n$, telescoping the bound (B.2) over $k = 1, 2, \ldots, n$ gives

$$D_{k+1} \leq D_k + \gamma_k \xi_k + \gamma_k \chi_k + \gamma_k^2 \psi_k^2, \quad \text{for all } k = 1, 2, \ldots n, \tag{B.16}$$

668  and hence, after telescoping over $k = 1, 2, \ldots, n$, we get

$$D_{n+1} \leq D_1 + M_n + S_n \leq D_1 + \sqrt{R_n} + R_n \leq a + \sqrt{a} + a = 2a + \sqrt{a}. \tag{B.17}$$

669  We conclude that $D(\pi_{n+1}) \leq 2a + \sqrt{a}$, i.e., $\pi_{n+1} \in \mathcal{B}$, as required for the induction.

670  For our third claim, note first that

$$R_n = (M_{n-1} + \gamma_n \xi_n)^2 + S_{n-1} + \gamma_n \chi_n + \gamma_n^2 \psi_n^2$$
$$= R_{n-1} + 2\gamma_n \xi_n M_{n-1} + \gamma_n^2 \xi_n^2 + \gamma_n \chi_n + \gamma_n^2 \psi_n^2, \tag{B.18}$$

671  so, after taking expectations, we get

$$\mathbb{E}[R_n \mid \mathcal{F}_n] = R_{n-1} + 2M_{n-1}\gamma_n \mathbb{E}[\xi_n \mid \mathcal{F}_n] + \mathbb{E}[\gamma_n^2 \xi_n^2 + \gamma_n \chi_n + \gamma_n^2 \psi_n^2 \mid \mathcal{F}_n] \geq R_{n-1}, \tag{B.19}$$

672  i.e., $R_n$ is a submartingale. To proceed, let $\tilde{R}_n = R_n \mathbb{1}_{H_{n-1}}$ so

$$\tilde{R}_n = R_n \mathbb{1}_{H_{n-1}} = R_{n-1} \mathbb{1}_{H_{n-1}} + (R_n - R_{n-1}) \mathbb{1}_{H_{n-1}}$$
$$= R_{n-1} \mathbb{1}_{H_{n-2}} - R_{n-1} \mathbb{1}_{\tilde{H}_{n-1}} + (R_n - R_{n-1}) \mathbb{1}_{H_{n-1}},$$
$$= \tilde{R}_{n-1} + (R_n - R_{n-1}) \mathbb{1}_{H_{n-1}} - R_{n-1} \mathbb{1}_{\tilde{H}_{n-1}}, \tag{B.20}$$

673  where we used the fact that $H_{n-1} = H_{n-2} \setminus \tilde{H}_{n-1}$ so $\mathbb{1}_{H_{n-1}} = \mathbb{1}_{H_{n-2}} - \mathbb{1}_{\tilde{H}_{n-1}}$ (since $H_{n-1} \subseteq H_{n-2}$). Then,
674  (B.18) yields

$$R_n - R_{n-1} = 2M_{n-1}\gamma_n \xi_n + \gamma_n^2 \xi_n^2 + \gamma_n \chi_n + \gamma_n^2 \psi_n^2 \tag{B.21}$$

675  and hence, given that $H_{n-1}$ is $\mathcal{F}_n$-measurable, we get:

$$\mathbb{E}[(R_n - R_{n-1}) \mathbb{1}_{H_{n-1}}] = 2 \mathbb{E}[\gamma_n M_{n-1} \xi_n \mathbb{1}_{H_{n-1}}] \tag{B.22a}$$
$$+ \mathbb{E}[\gamma_n^2 \xi_n^2 \mathbb{1}_{H_{n-1}}] \tag{B.22b}$$
$$+ \mathbb{E}[(\gamma_n \chi_n + \gamma_n^2 \psi_n^2) \mathbb{1}_{H_{n-1}}]. \tag{B.22c}$$

676  However, since $H_{n-1}$ and $M_{n-1}$ are both $\mathcal{F}_n$-measurable, we have the following estimates:

677  1. For the noise term in (B.22a), we have:

$$\mathbb{E}[M_{n-1} \xi_n \mathbb{1}_{H_{n-1}}] = \mathbb{E}[M_{n-1} \mathbb{1}_{H_{n-1}} \mathbb{E}[\xi_n \mid \mathcal{F}_n]] = 0. \tag{B.23}$$

678  2. The term (B.22b) is where the reduction to $H_{n-1}$ kicks in; indeed, we have:

$$\mathbb{E}[\xi_n^2 \mathbb{1}_{H_{n-1}}] = \mathbb{E}[\mathbb{1}_{H_{n-1}} \mathbb{E}[|\langle \pi_n - \pi^*, U_n \rangle|^2 \mid \mathcal{F}_n]]$$
$$\leq \mathbb{E}[\mathbb{1}_{H_{n-1}} \|\pi_n - \pi^*\|^2 \mathbb{E}[\|U_n\|^2 \mid \mathcal{F}_n]] \qquad \text{\# by Cauchy–Schwarz}$$
$$\leq \mathbb{E}[\mathbb{1}_{\mathcal{E}_n} \|\pi_n - \pi^*\|^2 \mathbb{E}[\|U_n\|^2 \mid \mathcal{F}_n]] \qquad \text{\# because } H_{n-1} \subseteq \mathcal{E}_n$$
$$\leq \|\Pi\|^2 \sigma_n^2. \tag{B.24}$$

679  3. Finally, for the term (B.22c), we have:

$$\mathbb{E}[\psi_n^2 \mathbb{1}_{H_{n-1}}] \leq \tfrac{3}{2}[G^2 + B_n^2 + \sigma_n^2] \tag{B.25}$$

680  where we used the bound $\|v(\pi)\| \leq G$. Likewise, $\chi_n \mathbb{1}_{H_{n-1}} \leq \|\Pi\| B_n$, so

$$(B.22c) \leq \gamma_n \|\Pi\| B_n + \tfrac{3}{2}\gamma_n^2 (G^2 + B_n^2 + \sigma_n^2) \tag{B.26}$$

681  Thus, putting together all of the above, we obtain:

$$\mathbb{E}[(R_n - R_{n-1}) \mathbb{1}_{H_{n-1}}] \leq \gamma_n \|\Pi\| B_n + \gamma_n^2 \|\Pi\|^2 \sigma_n^2 + \tfrac{3}{2}\gamma_n^2 (G^2 + B_n^2 + \sigma_n^2) \tag{B.27}$$

682  Going back to (B.20), we have $R_{n-1} > a$ if $\tilde{H}_{n-1}$ occurs, so the last term becomes

$$\mathbb{E}[R_{n-1} \mathbb{1}_{\tilde{H}_{n-1}}] \geq a \mathbb{E}[\mathbb{1}_{\tilde{H}_{n-1}}] = a \mathbb{P}(\tilde{H}_{n-1}). \tag{B.28}$$

683  Our claim then follows by combining Eqs. (B.20), (B.25), (B.26) and (B.28). ∎

**B.3. Extraction of a convergent subsequence.** Our next step is to show that any realization $\pi_n$ of (PG) that is contained in $\mathcal{B}$ admits a subsequence $\pi_{n_k}$ converging to $\pi^*$.

**Proposition B.2.** *Let $\pi^*$ be a stable Nash policy, and let $\pi_n$ be the sequence of play generated by (PG) with step-size and policy gradient estimates such that $p + \ell_b > 1$ and $p - \ell_\sigma > 1/2$ as per (8). Then $\pi_n$ admits a subsequence $\pi_{n_k}$ that converges to $\pi^*$ with probability 1 on the event $\mathcal{E} = \bigcap_n \mathcal{E}_n = \{\pi_n \in \mathcal{B} \text{ for all } n = 1, 2, \dots\}$.*

*Proof.* Let $\mathcal{Q} = \{\pi_n \in \mathcal{B} \text{ for all } n\} \cap \{\liminf_n \|\pi_n - \pi^*\| > 0\}$ denote the event that $\pi_n$ is contained in $\mathcal{B}$ but the sequence $\pi_n$ does not admit a subsequence converging to $\pi^*$. We will show that $\mathbb{P}(\mathcal{Q}) = 0$.

Indeed, assume ad absurdum that $\mathbb{P}(\mathcal{Q}) > 0$. Hence, with probability 1 on $\mathcal{Q}$, there exists some positive constant $c > 0$ (again, possibly random) such that $\langle v(\pi_n), \pi_n - \pi^* \rangle \leq -c < 0$ for all $n$. Thus, going back to (B.1), we get

$$D_{n+1} \leq D_n - \gamma_n c + \gamma_n \xi_n + \gamma_n \chi_n + \gamma_n^2 \psi_n^2, \tag{B.29}$$

so if we let $\tau_n = \sum_{k=1}^n \gamma_k$ and telescope the above, we obtain the bound

$$D_{n+1} \leq D_1 - \tau_n \left[ c - \frac{M_n}{\tau_n} - \frac{S_n}{\tau_n} \right] \tag{B.30}$$

with $\xi_n, \chi_n$ and $\psi_n$ given by (B.3), and $M_n = \sum_{k=1}^n \gamma_k \xi_k$, $S_n = \sum_{k=1}^n [\gamma_k \chi_k + \gamma_k^2 \psi_k^2]$ defined as in (D.10). Also, (7) readily gives

$$\sum_{n=1}^\infty \mathbb{E}[\gamma_n^2 \xi_n^2 \mid \mathcal{F}_n] \leq \sum_{n=1}^\infty \gamma_n^2 \mathbb{E}[\|\pi_n - \pi^*\|^2 \|U_n\|^2 \mid \mathcal{F}_n] \leq \|\Pi\|^2 \sum_{n=1}^\infty \gamma_n^2 \sigma_n^2 < \infty \tag{B.31}$$

so, by the strong law of large numbers for martingale difference sequences [23, Theorem 2.18], we conclude that $M_n/\tau_n$ converges to 0 with probability 1. In a similar vein, for the submartingale $S_n$ we have

$$\mathbb{E}[S_n] = \sum_{k=1}^n \gamma_k \chi_k \sum_{k=1}^n \gamma_k^2 \mathbb{E}[\psi_k^2] \leq \|\Pi\| \sum_{k=1}^n \gamma_k B_k + \frac{3}{2} \sum_{k=1}^n \gamma_k^2 [G^2 + B_k^2 + \sigma_k^2], \tag{B.32}$$

so, by (7) and the stated conditions for the method's step-size and bias/noise parameters, it follows that $S_n$ is bounded in $L^1$. Therefore, by Doob's submartingale convergence theorem [23, Theorem 2.5], we further deduce that $S_n$ converges with probability 1 to some (finite) random variable $S_\infty$.

Going back to (B.30) and letting $n \to \infty$, the above shows that $D_n \to -\infty$ with probability 1 on $\mathcal{Q}$. Since $D$ is nonnegative by construction and $\mathbb{P}(\mathcal{Q}) > 0$ by assumption, we obtain a contradiction and our proof is complete. ∎

**B.4. Convergence of the energy values.** Our last auxiliary result concerns the convergence of the values of the dual energy function $D$. We encode this as follows.

**Proposition B.3.** *If (PG) is run with assumptions as in Proposition B.1, there exists a finite random variable $D_\infty$ such that*

$$\mathbb{P}(D_n \to D_\infty \text{ as } n \to \infty \mid \pi_n \in \mathcal{B} \text{ for all } n) = 1. \tag{B.33}$$

*Proof.* Let $\mathcal{E}_n = \{\pi_k \in \mathcal{B} \text{ for all } k = 1, 2, \dots, n\}$ be defined as in (B.7), and let $\tilde{D}_n = \mathbb{1}_{\mathcal{E}_n} D_n$. Then, by the energy inequality (B.2) and the fact that $\mathcal{E}_{n+1} \subseteq \mathcal{E}_n$, we get

$$\begin{aligned}
\tilde{D}_{n+1} = \mathbb{1}_{\mathcal{E}_{n+1}} D_{n+1} &\leq \mathbb{1}_{\mathcal{E}_n} D_{n+1} \\
&\leq \mathbb{1}_{\mathcal{E}_n} D_n + \mathbb{1}_{\mathcal{E}_n} \gamma_n \langle v(\pi_n), \pi_n - \pi^* \rangle + (\gamma_n \xi_n + \gamma_n \chi_n + \gamma_n^2 \psi_n^2) \mathbb{1}_{\mathcal{E}_n} \\
&\leq \tilde{D}_n + \gamma_n \mathbb{1}_{\mathcal{E}_n} \xi_n + (\gamma_n \chi_n + \gamma_n^2 \psi_n^2) \mathbb{1}_{\mathcal{E}_n},
\end{aligned} \tag{B.34}$$

where we used the fact that that $\langle v(\pi_k), \pi_k - \pi^* \rangle \leq 0$ for all $k = 1, 2, \dots, n$ if $\mathcal{E}_n$ occurs. Since $\mathcal{E}_n$ is $\mathcal{F}_n$-measurable, conditioning on $\mathcal{F}_n$ and taking expectations yields

$$\mathbb{E}[\tilde{D}_{n+1} \mid \mathcal{F}_n] \leq \tilde{D}_n + \gamma_n \mathbb{1}_{\mathcal{E}_n} \mathbb{E}[\xi_n \mid \mathcal{F}_n] + \mathbb{1}_{\mathcal{E}_n} \gamma_n \chi_n + \mathbb{1}_{\mathcal{E}_n} \mathbb{E}[\gamma_n^2 \psi_n^2 \mid \mathcal{F}_n]$$

$$\leq \tilde{D}_n + \gamma_n \|\Pi\| B_n + \gamma_n \chi_n + \mathbb{E}[\gamma_n^2 \psi_n^2 \,|\, \mathcal{F}_n]$$

$$\leq \tilde{D}_n + \gamma_n \|\Pi\| B_n + \frac{3}{2}[G^2 + B_n^2 + \sigma_n^2]. \tag{B.35}$$

By our step-size assumptions, we have $\sum_n \gamma_n^2(1 + B_n^2 + \sigma_n^2) < \infty$ and $\sum_n \gamma_n B_n < \infty$, which means that $\tilde{D}_n$ is an almost supermartingale with almost surely summable increments, i.e.,

$$\sum_{n=1}^{\infty} \left[ \mathbb{E}[\tilde{D}_{n+1} \,|\, \mathcal{F}_n] - \tilde{D}_n \right] < \infty \quad \text{with probability 1} \tag{B.36}$$

Therefore, by Gladyshev's lemma [45, p. 49], we conclude that $\tilde{D}_n$ converges almost surely to some (finite) random variable $D_\infty$. Since $\mathbb{1}_{\mathcal{E}_n} = 1$ for all $n$ if and only if $\pi_n \in \mathcal{B}$ for all $n$, we conclude that $\mathbb{P}(D_n \text{ converges} \,|\, \pi_n \in \mathcal{B} \text{ for all } n) = \mathbb{P}(\tilde{D}_n \text{ converges}) = 1$, and our claim follows. $\blacksquare$

**B.5. Putting everything together.** We are now in a position to prove Theorem 1 and Corollary 1.

*Proof of Theorem 1.* Let $\mathcal{E} = \bigcap_n \mathcal{E}_n = \{\pi_n \in \mathcal{B} \text{ for all } n\}$ denote the event that $\pi_n$ lies in $\mathcal{B}$ for all $n$. By Proposition B.1, if $\pi_1$ is initialized within the neighborhood $\mathcal{U}$ defined in (B.8), we have $\mathbb{P}(\mathcal{E} \,|\, \pi_1 \in \mathcal{U}) \geq 1 - a$, noting also that the neighborhood $\mathcal{U}$ is independent of the required confidence level $a$. Then, by Propositions B.2 and B.3, it follows that *a)* $\liminf_n \|\pi_n - \pi^*\| = 0$; and *b)* $D_n$ converges, both events occurring with probability 1 on the set $\mathcal{E} \cap \{\pi_1 \in \mathcal{U}\}$. We thus conclude that $\lim_{n\to\infty} D_n = 0$ and hence

$$\mathbb{P}(\pi_n \to \pi^* \,|\, \pi_1 \in \mathcal{U}) \geq \mathbb{P}(\mathcal{E} \cap \{\pi_n \to \pi^*\} \,|\, \pi_1 \in \mathcal{U})$$
$$= \mathbb{P}(\pi_n \to \pi^* \,|\, \pi_1 \in \mathcal{U}, \mathcal{E}) \times \mathbb{P}(\mathcal{E} \,|\, \pi_1 \in \mathcal{U}) \geq 1 - \delta,$$

and our proof is complete. $\blacksquare$

*Proof of Corollary 1.* For Models 1 and 2, taking $\ell_b = \infty, \ell_\sigma = 0$, we obtain $p > 1/2$. Since we have that $\sum_{n=1}^{\infty} \gamma_n = \infty$, we get that $p \leq 1$, i.e., $p \in (1/2, 1]$.

For Model 3, we have that $B_n = \mathcal{O}(\varepsilon_n)$ and $\sigma_n = \mathcal{O}(1/\sqrt{\varepsilon_n})$, i.e., $\ell_b = r$ and $\ell_\sigma = r/2$. Now, since $p \leq 1, p + \ell_b > 1$ and $p - \ell_\sigma > 1/2$, we obtain that $p \in (2/3, 1]$ and $(1-p)/2 < r/2 < p - 1/2$. $\blacksquare$

# C   Rate of convergence to second-order stationary policies

We now proceed with the proof of Theorem 2, which we again restate below for convenience:

**Theorem 2.** *Let $\pi^*$ be a Nash policy such that (SOS) holds on some open set $\mathcal{B}$ containing $\pi^*$, and let $\pi_n$ be the sequence of play generated by (PG) with step-size $\gamma_n = \gamma/(n+m)^p$, $p \in (1/2, 1]$, and policy gradient estimates such that $p + \ell_b > 1$ and $p - \ell_\sigma > 1/2$ as per (8). Then:*

1. *There exists a neighborhood $\mathcal{U}$ of $\pi^*$ in $\Pi$ such that, for any confidence level $\delta > 0$, the event*

$$\mathcal{E} = \{\pi_n \in \mathcal{B} \text{ for all } n = 1, 2, \dots\} \tag{17}$$

   *occurs with probability $\mathbb{P}(\mathcal{E} \,|\, \pi_1 \in \mathcal{U}) \geq 1 - \delta$ if $m$ is large enough relative to $\delta$.*

2. *The sequence $\pi_n$ converges to $\pi^*$ with probability 1 on $\mathcal{E}$; in particular, we have*

$$\mathbb{P}(\pi_n \text{ converges to } \pi^* \,|\, \pi_1 \in \mathcal{U}) \geq 1 - \delta \tag{18}$$

   *if $m$ is large relative to $\delta$. Moreover, conditioned on $\mathcal{E}$ and taking $q = \min\{\ell_b, p - 2\ell_\sigma\}$, we have*

$$\mathbb{E}[\|\pi_n - \pi^*\|^2 \,|\, \mathcal{E}] = \begin{cases} \mathcal{O}(1/n^{2\mu\gamma}) & \text{if } p = 1 \text{ and } 2\mu\gamma < q, \\ \mathcal{O}(1/n^q) & \text{otherwise.} \end{cases} \tag{19}$$

*Proof.* We will follow an approach similar to Theorem 1 for the first part of the theorem. More precisely, let $\mathcal{B} = \{\pi \in \Pi : \|\pi - \pi^*\| \leq r\}$ be a ball of radius $r$ centered at $\pi^*$ in $\Pi$ such that (SOS) holds

743 for all $\pi \in \mathcal{B}$. Then, for all $\pi \in \mathcal{B}\setminus\{\pi^*\}$, we have $\langle v(\pi), \pi - \pi^* \rangle \leq -\mu\|\pi - \pi^*\| < 0$ by Proposition 1.

744 Hence, defining the events $\mathcal{E}_n$ and $H_n$ as in Eq. (B.7), and assuming that $\pi_1$ is initialized in a ball of

745 radius $\sqrt{2a}$ centered at $\pi^*$, viz.

$$\mathcal{U} = \{\pi \in \Pi : D(\pi) \leq a\} = \{\pi \in \Pi : \|\pi - \pi^*\|^2/2 \leq a\}. \tag{C.1}$$

746 then, by Lemma B.2 and Proposition B.1, we readily obtain that

$$\mathbb{P}(H_n \mid \pi_1 \in \mathcal{U}) \geq 1 - \delta \quad \text{for all } n = 1, 2, \dots \tag{C.2}$$

747 which implies that

$$\mathbb{P}(\mathcal{E} \mid \pi_1 \in \mathcal{U}) \geq 1 - \delta \tag{C.3}$$

748 if $m$ is large enough relative to $\delta$.

749 For the second part, constraining Eq. (B.2) on the event $\mathcal{E}_n$, we get:

$$D_{n+1} \mathbb{1}_{\mathcal{E}_n} \leq D_n \mathbb{1}_{\mathcal{E}_n} + \gamma_n \langle v(\pi_n), \pi_n - \pi^* \rangle \mathbb{1}_{\mathcal{E}_n} + \mathbb{1}_{\mathcal{E}_n}\left(\gamma_n \xi_n + \gamma_n \chi_n + \gamma_n^2 \psi_n^2\right)$$

$$\leq (1 - 2\mu\gamma_n)D_n \mathbb{1}_{\mathcal{E}_n} + \mathbb{1}_{\mathcal{E}_n}\left(\gamma_n \xi_n + \gamma_n \chi_n + \gamma_n^2 \psi_n^2\right) \tag{C.4}$$

750 where the last inequality comes from (SOS). Therefore, taking expectations, we obtain:

$$\mathbb{E}[D_{n+1} \mathbb{1}_{\mathcal{E}_n}] \leq (1 - 2\mu\gamma_n)\mathbb{E}[D_n \mathbb{1}_{\mathcal{E}_n}] + \mathbb{E}\left[\mathbb{1}_{\mathcal{E}_n}\left(\gamma_n \xi_n + \gamma_n \chi_n + \gamma_n^2 \psi_n^2\right)\right]$$

$$\leq (1 - 2\mu\gamma_n)\mathbb{E}[D_n \mathbb{1}_{\mathcal{E}_n}] + \gamma_n \mathbb{E}[\mathbb{1}_{\mathcal{E}_n} \xi_n] + \gamma_n \mathbb{E}[\mathbb{1}_{\mathcal{E}_n} \chi_n] + \gamma_n^2 \mathbb{E}[\mathbb{1}_{\mathcal{E}_n} \psi_n^2]$$

$$= (1 - 2\mu\gamma_n)\mathbb{E}[D_n \mathbb{1}_{\mathcal{E}_n}] + \gamma_n \mathbb{E}[\mathbb{1}_{\mathcal{E}_n} \chi_n] + \gamma_n^2 \mathbb{E}[\mathbb{1}_{\mathcal{E}_n} \psi_n^2]$$

$$\leq (1 - 2\mu\gamma_n)\mathbb{E}[D_n \mathbb{1}_{\mathcal{E}_n}] + \|\Pi\| \mathbb{P}(\mathcal{E}_n)\gamma_n B_n + \mathbb{P}(\mathcal{E}_n)\left(G\gamma_n^2 + 3\gamma_n^2\sigma_n^2 + 3\gamma_n^2 B_n^2\right) \tag{C.5}$$

751 where the equality in the third line comes from the fact that

$$\mathbb{E}[\mathbb{1}_{\mathcal{E}_n} \xi_n] = \mathbb{E}[\mathbb{E}[\xi_n \mathbb{1}_{\mathcal{E}_n} \mid \mathcal{F}_n]] = \mathbb{E}[\mathbb{1}_{\mathcal{E}_n} \mathbb{E}[\xi_n \mid \mathcal{F}_n]] = 0. \tag{C.6}$$

752 Now, since $\mathbb{1}_{\mathcal{E}_{n+1}} \leq \mathbb{1}_{\mathcal{E}_n}$, we further have

$$\mathbb{E}[D_{n+1} \mathbb{1}_{\mathcal{E}_{n+1}}] \leq \mathbb{E}[D_{n+1} \mathbb{1}_{\mathcal{E}_n}] \tag{C.7}$$

753 and hence, setting $\bar{D}_n \coloneqq \mathbb{E}[D_n \mathbb{1}_{\mathcal{E}_n}]$, we get

$$\bar{D}_{n+1} \leq (1 - 2\mu\gamma_n)\bar{D}_n + \|\Pi\| \mathbb{P}(\mathcal{E}_n)\gamma_n B_n + \mathbb{P}(\mathcal{E}_n)\left(G\gamma_n^2 + 3\gamma_n^2\sigma_n^2 + 3\gamma_n^2 B_n^2\right)$$

$$\leq (1 - 2\mu\gamma_n)\bar{D}_n + \|\Pi\|\gamma_n B_n + G\gamma_n^2 + 3\gamma_n^2\sigma_n^2 + 3\gamma_n^2 B_n^2. \tag{C.8}$$

754 Therefore, taking $\gamma_n, B_n, \sigma_n$ as per the statement of the theorem and noting that the terms $\gamma_n^2$ and $\gamma_n^2 B_n^2$

755 are respectively dominated by the terms $\gamma_n^2\sigma_n^2$ and $\gamma_n B_n$, we obtain

$$\bar{D}_{n+1} \leq \left(1 - \frac{2\mu\gamma}{(n+m)^p}\right)\bar{D}_n + \frac{C_1}{(n+m)^{p+\ell_b}} + \frac{C_2}{(n+m)^{2p-2\ell_\sigma}}$$

$$\leq \left(1 - \frac{2\mu\gamma}{(n+m)^p}\right)\bar{D}_n + \frac{C_1 + C_2}{(n+m)^{p+q}} \tag{C.9}$$

756 for some $C_1, C_2 > 0$, where $q = \min\{\ell_b, p - 2\ell_\sigma\}$, as per the theorem's statement. Therefore, by a

757 straightforward modification of Chung's lemma [14, Lemmas 2&3], [45, p. 45], we get

$$\bar{D}_n = \begin{cases} \mathcal{O}(1/n^{2\mu\gamma}) & \text{if } p = 1 \text{ and } 2\mu\gamma < q, \\ \mathcal{O}(1/n^q) & \text{otherwise.} \end{cases} \tag{C.10}$$

758 Accordingly, letting $n \to \infty$ and recalling that $\mathbb{E}[D_n \mathbb{1}_{\mathcal{E}}] \leq \mathbb{E}[D_n \mathbb{1}_{\mathcal{E}_n}] = \bar{D}_n$

$$\lim_{n\to\infty} \mathbb{E}[D_n \mathbb{1}_{\mathcal{E}}] = 0. \tag{C.11}$$

759 Then, by Fatou's lemma [21], we obtain

$$0 \leq \mathbb{E}[\liminf_{n\to\infty} D_n \mathbb{1}_{\mathcal{E}}] \leq \liminf_{n\to\infty} \mathbb{E}[D_n \mathbb{1}_{\mathcal{E}}] = 0, \tag{C.12}$$

which readily shows that $\mathbb{E}[\liminf_{n\to\infty} D_n \mathbb{1}_{\mathcal{E}}] = 0$. Finally, since $\liminf_{n\to\infty} D_n \mathbb{1}_{\mathcal{E}} \geq 0$ (a.s.) and $\mathbb{E}[\liminf_{n\to\infty} D_n \mathbb{1}_{\mathcal{E}}] = 0$, we get that

$$\liminf_{n\to\infty} D_n \mathbb{1}_{\mathcal{E}} = 0 \quad \text{with probability 1.} \tag{C.13}$$

Therefore, there exists a subsequence $D_{n_k}$ that converges to 0 with probability 1 on the event $\mathcal{E}$, i.e., $\pi_{n_k}$ converges to $\pi^*$. Hence, invoking Proposition B.3, we further deduce that $D_n$ converges to some $D_\infty$ with probability 1 on $\mathcal{E}$, and thus, we obtain that $\lim_{n\to\infty} D_n = 0$ on $\mathcal{E}$. We thus get

$$\mathbb{P}(\pi_n \to \pi^* \mid \pi_1 \in \mathcal{U}) \geq \mathbb{P}(\mathcal{E} \cap \{\pi_n \to \pi^*\} \mid \pi_1 \in \mathcal{U})$$
$$= \mathbb{P}(\pi_n \to \pi^* \mid \pi_1 \in \mathcal{U}, \mathcal{E}) \times \mathbb{P}(\mathcal{E} \mid \pi_1 \in \mathcal{U}) \geq 1 - \delta, \tag{C.14}$$

as claimed.

For the last part of the theorem, note that

$$\bar{D}_n = \mathbb{E}[D_n \mathbb{1}_{\mathcal{E}_n}] \geq \mathbb{E}[D_n \mathbb{1}_{\mathcal{E}}] = \mathbb{E}[\mathbb{E}[D_n \mid \sigma(\mathcal{E})] \mathbb{1}_{\mathcal{E}}]$$
$$= \mathbb{E}[\mathbb{E}[D_n \mid \mathcal{E}] \mathbb{1}_{\mathcal{E}}]$$
$$= \mathbb{E}[D_n \mid \mathcal{E}] \mathbb{E}[\mathbb{1}_{\mathcal{E}}]$$
$$= \mathbb{E}[D_n \mid \mathcal{E}] \mathbb{P}(\mathcal{E}) \tag{C.15}$$

where we used the fact that $\mathbb{E}[D_n \mid \sigma(\mathcal{E})] \mathbb{1}_{\mathcal{E}} = \mathbb{E}[D_n \mid \mathcal{E}] \mathbb{1}_{\mathcal{E}}$. We thus conclude that

$$\mathbb{E}\left[\|\pi_n - \pi^*\|^2 \,\big|\, \mathcal{E}\right] = 2\,\mathbb{E}[D_n \mid \mathcal{E}] \leq \frac{2}{\mathbb{P}(\mathcal{E})} \bar{D}_n \leq \frac{2}{1-\delta} \bar{D}_n$$

$$\tag{C.16}$$

and hence

$$\mathbb{E}\left[\|\pi_n - \pi^*\|^2 \,\big|\, \mathcal{E}\right] = \begin{cases} \mathcal{O}(1/n^{2\mu\gamma}) & \text{if } p = 1 \text{ and } 2\mu\gamma < q, \\ \mathcal{O}(1/n^q) & \text{otherwise.} \end{cases} \qquad \blacksquare$$

*Proof of Corollary 2.* For Models 1 and 2, taking $\ell_b = \infty, \ell_\sigma = 0$ we readily get that $q = p$ and $p > 1/2$. Since we require that $\sum_{n=1}^\infty \gamma_n = \infty$, we obtain that $p \in (1/2, 1]$. Hence, for $p = 1$ and $2\mu\gamma > 1$ we obtain $\mathcal{O}(1/n)$ rate of convergence.

For Model 3, we have that $B_n = \mathcal{O}(\varepsilon_n)$ and $\sigma_n = \mathcal{O}(1/\sqrt{\varepsilon_n})$, i.e., $\ell_b = p/2$ and $\ell_\sigma = p/4$, and, hence, we readily get that $q = p/2$. Now, since $p \leq 1$, $p + \ell_b > 1$ and $p - \ell_\sigma > 1/2$, we obtain that $p \in (2/3, 1]$. Hence, for $p = 1$ and $\mu\gamma > 1$, we obtain $\mathcal{O}(1/\sqrt{n})$ rate of convergence. $\blacksquare$

# D   Rate of convergence to strict Nash policies

**D.1. Structural preliminaries.** To prove Theorem 3, we will first require some notions describing the geometry of $\Pi$ near $\pi^*$. Referring to [47] for a full treatment, we have:

**Definition 3.** Let $\mathcal{C}$ be a convex set and let $x \in \mathcal{C}$. Then the tangent cone $\mathrm{TC}_{\mathcal{C}}(x)$ is defined as the set of all rays emanating from $x$ and intersecting $\mathcal{C}$ to at least one other point different from $x$. The *polar cone* $\mathrm{PC}_{\mathcal{C}}(x)$ to $\mathcal{C}$ at $x$ is then defined $\mathrm{PC}_{\mathcal{C}}(x) = \{y : \langle y, z \rangle \leq 0 \text{ for all } z \in \mathrm{TC}_{\mathcal{C}}(x)\}$, where $y$ belong in the dual space of the vector space in which $\mathcal{C}$ is defined.

With these general definitions in hand, we proceed to characterize some further projections of Euclidean projections on $\Pi$ that will play an important role in the sequel. For notational simplicity, we suppress the player and state indices in the statement and proof of the next lemma.

**Lemma D.1.** $x = \mathrm{proj}(y)$ *if and only if there exist* $\mu \in \mathbb{R}$ *and* $\nu_\alpha \in \mathbb{R}_+$ *such that, for all* $\alpha \in \mathcal{A}$, *we have* $y_\alpha = x_\alpha + \mu - \nu_\alpha$ *with* $\nu_\alpha \geq 0$ *and* $x_\alpha \nu_\alpha = 0$.

*Proof.* Recall that $\mathrm{proj}(y) = \arg\min_{x \in \Delta(\mathcal{A})} \|y - x\|^2$. Our result then follows by applying the KKT conditions to this optimization problem and noting that, since the constraints are affine, the KKT conditions are sufficient for optimality. Our Langragian is

$$\mathcal{L}(x, \mu, \nu) = \sum_{\alpha \in \mathcal{A}} \frac{1}{2}(y_\alpha - x_\alpha)^2 - \mu\left(\sum_{\alpha \in \mathcal{A}} x_\alpha - 1\right) + \sum_{\alpha \in \mathcal{A}} \nu_\alpha x_\alpha$$

where the set of constraints (i) of the statement of the lemma are the stationarity constraints, which in our case are $\nabla \mathcal{L}(x, \mu, \nu) = 0 \Leftrightarrow \nabla(\sum_{\alpha \in \mathcal{A}} \frac{1}{2}(y_\alpha - x_\alpha)^2) = \mu \nabla(\sum_{\alpha \in \mathcal{A}} x_\alpha - 1) - \sum_{\alpha \in \mathcal{A}} \nu_\alpha \nabla x_\alpha$, while the set of constraints (ii) of the statement of the lemmas are the complementary slackness constraints. Note that complementary slackness implies $\nu_\alpha > 0$ whenever $\alpha \notin \text{supp}(x)$, so our proof is complete. ∎

Our next result is a concrete consequence of [Proposition 1] which will be very useful in establishing the stability estimates required for the proof of [Theorem 3].

**Lemma D.2.** *Let $\pi^* = (\alpha_{i,s}^*)_{i \in \mathcal{N}, s \in \mathcal{S}}$ be a strict Nash policy. Then there exists a neighborhood $\mathcal{U}$ of $\pi^*$ and constants $c_{i,s}$ such that for each player $i \in \mathcal{N}$ and state $s \in \mathcal{S}$, we have:*

$$v_{i\alpha_{i,s}^*}(\pi) - v_{i\alpha_{i,s}}(\pi) \geq c_{i,s} \ \text{ for all } \pi \in \mathcal{U} \text{ and } \alpha_i \neq \alpha_i^*, \ \alpha_i \in \mathcal{A}_i. \tag{D.1}$$

*Proof.* Our claim is a consequence of the definition of strict Nash policies. Specifically, from [Proposition 1] we have

$$\langle v(\pi^*), z \rangle < 0 \quad \text{for all} \quad z \in \text{TC}(\pi^*), z \neq 0 \tag{D.2}$$

Let $z = e_{i,\alpha_{i,s}} - e_{i,\alpha_{i,s}^*}$, then we get that

$$v_{i\alpha_{i,s}^*}(\pi^*) - v_{i\alpha_{i,s}}(\pi^*) > 0 \tag{D.3}$$

where $e_{i,\alpha_{i,s}}$ is the vector that has one only in the index and zero anywhere else. By continuity there exists a neighborhood $\mathcal{U} \subseteq \mathcal{X}$ and $c_{i,s} > 0$ for each player $i \in \mathcal{N}$ such that

$$v_{i\alpha_{i,s}^*}(\pi) - v_{i\alpha_{i,s}}(\pi) \geq c_{i,s} \quad \text{for all} \quad \pi \in \mathcal{U} \qquad \qquad ∎$$

Our final result is intimately tied to the lazy projection step in ([LPG]), and quantifies the relation between initializations in $\prod_i (\mathbb{R}^{\mathcal{A}_i})^{\mathcal{S}}$ and $\Pi$.

**Lemma D.3.** *Let $\pi^* = (\alpha_{i,s}^*)_{i \in \mathcal{N}, s \in \mathcal{S}}$, be a deterministic policy. For each agent $i \in \mathcal{N}$ and each state $s \in \mathcal{S}$, let $y_{i,\alpha_{i,s}} - y_{i,\alpha_{i,s}^*}$ be the difference of the aggregated gradients between the strategy of the equilibrium and any other strategy $\alpha_i^* \neq \alpha_i \in \mathcal{A}_i$. Then for any $\varepsilon > 0$ such that $\mathcal{U}_\varepsilon = \{\pi : \pi_{i,\alpha_{i,s}^*} \geq 1 - \varepsilon$ for all $i \in \mathcal{N}$ and $s \in \mathcal{S}\}$, there exist $M_{i,\varepsilon,s}$ such that if $\mathcal{W}_{i,s} = \{y \in \mathbb{R}^{\mathcal{A}_i} : y_{i,\alpha_{i,s}} - y_{i,\alpha_{i,s}^*} < -M_{i,\varepsilon,s}\}$ then $\prod_{i \in \mathcal{N}, s \in \mathcal{S}} \text{proj}_{\Pi_i}(\mathcal{W}_{i,s}) \subseteq \mathcal{U}_\varepsilon$.*

*Proof.* Consider an arbitrary player $i \in \mathcal{N}$, a state $s \in \mathcal{S}$, and let $\mathcal{W}_i(M_{i,\varepsilon,s})$ be an open set as defined in the statement of the lemma. For notational simplicity, we will drop the index $s$. We will show that any $M_{i,\varepsilon} > 1 - \frac{\varepsilon}{|\mathcal{A}_i|} > 0$ satisfies our claim. By using [Lemma D.1] for a $y_i \in \mathcal{W}_i(M_{i,\varepsilon})$ with $\pi_i = \text{proj}(y_i)$ we have that

$$y_{i\alpha_i^*} - y_{i\alpha_i} > M_{i,\varepsilon} \tag{D.4}$$

$$\pi_{i\alpha_i^*} - \pi_{i\alpha_i} - (\nu_{\alpha_i^*} - \nu_{\alpha_i}) > M_{i,\varepsilon} \tag{D.5}$$

with $\nu_{\alpha_i} \geq 0$ and $\pi_{i\alpha_i} = 0$ whenever $\nu_{\alpha_i} > 0$. Notice that since $M_{i,\varepsilon} > 1 - \frac{\varepsilon}{|\mathcal{A}_i|}$ we have that $\pi_{i\alpha_i^*} > \pi_{i\alpha_i} + 1 - \frac{\varepsilon}{|\mathcal{A}_i|} + (\nu_{\alpha_i^*} - \nu_{\alpha_i})$ or

$$\pi_{i\alpha_i} < \pi_{i\alpha_i^*} - 1 + \frac{\varepsilon}{|\mathcal{A}_i|} - (\nu_{\alpha_i^*} - \nu_{\alpha_i}) < \frac{\varepsilon}{|\mathcal{A}_i|} \tag{D.6}$$

Hence, by summing over all strategies of player $i$ we get the desired result. ∎

**D.2. Proof of the main theorem.** We are now in a position to prove our main result on the rate of convergence towards strict Nash policies. For ease of reference, we restate [Theorem 3] below.

**Theorem 3.** *Let $\pi_n$ be the sequence of play under ([LPG]) with step-size and policy gradient estimates such that $p + \ell_b > 1$ and $p - \ell_\sigma > 1/2$ as per ([8]). If $\pi^*$ is a deterministic Nash policy, there exists an unbounded open set $\mathcal{W} \subseteq \prod_i (\mathbb{R}^{\mathcal{A}_i})^{\mathcal{S}}$ of initializations such that, for any $\delta > 0$, we have*

$$\mathbb{P}(\pi_n \text{ converges to } \pi^* \mid y_1 \in \mathcal{W}) \geq 1 - \delta, \tag{20}$$

*provided that $\gamma > 0$ is small enough. Moreover, conditioned on this event, $\pi_n$ converges to $\pi^*$ at a finite number of iterations, i.e., there exists some $n_0$ such that $\pi_n = \pi^*$ for all $n \geq n_0$.*

*Proof of Theorem 3.* We start by fixing a confidence level $\delta > 0$ and all the parameters of the algorithm, such that all the assumptions stated in the theorem are satisfied and. We will prove that for each agent $i \in \mathcal{N}$, $s \in \mathcal{S}$ there exist $M_{1,i,s} > 0$, $\mathcal{W}_{1,i,s} = \{y \in \mathbb{R}^{\mathcal{A}_i} : y_{i,\alpha_i} - y_{i,\alpha_i^*} < -M_{1,i,s} \text{ for all } \alpha_i \in \mathcal{A}_i, \alpha_i \neq \alpha_i^*\}$, such that if $y_1 \in \mathcal{W}_1 := \prod_{i \in \mathcal{N}, s \in \mathcal{S}} \mathcal{W}_{1,i,s}$ then the agents' sequence of play, converge to the deterministic Nash policy, in finite number of iterations.

To simplify the notation, we will drop the indices $s$ and $i$ referring to the states and agents, accordingly, and we will focus on a specific agent and a specific state. From Lemma D.3, Lemma D.2 we have that there exist constants $c$, $M$, neighborhood $\mathcal{U}_c = \{\pi \in \Pi : \|\pi - \pi^*\| \leq \beta\}$ and open set $\mathcal{W}_M$ such that

$$v_{\alpha^*}(\pi) - v_\alpha(\pi) \geq c \qquad \text{for all } \alpha \neq \alpha^*, \alpha \in \mathcal{A} \text{ and } \pi \in \mathcal{U}_c \tag{D.7}$$

$$y_{\alpha^*} - y_\alpha > M_c \quad \text{for all } \alpha \neq \alpha^*, \alpha \in \mathcal{A} \text{ and } \pi = \text{proj}(y) \in \mathcal{U}_c \tag{D.8}$$

The first step is to prove that for an appropriate initialization for $y_1$, we have $y_n \in \mathcal{W}(M_c)$ for all $n = 1, 2, \ldots$, with probability at least $1 - \delta$. Assume that $y_k \in \mathcal{W}(M_c)$ for all $k = 1, \ldots, n$; then for the differences of the scores at a round $n + 1$ between any $\alpha \in \mathcal{A}$ and the equilibrium strategy $\alpha^*$, we have

$$\begin{aligned}
y_{\alpha,n+1} - y_{\alpha^*,n+1} &= y_{\alpha,n} - y_{\alpha^*,n} + (\hat{v}_{\alpha,n} - \hat{v}_{\alpha^*,n}) \\
&= y_{\alpha,1} - y_{\alpha^*,1} + \sum_{k=1}^n \gamma_k[(v_{\alpha,k} - v_{\alpha^*,k}) + (U_{\alpha,k} - U_{\alpha^*,k}) + (b_{\alpha,k} - b_{\alpha^*,k})] \\
&\leq -M_1 + \sum_{k=1}^n \gamma_k[(v_{\alpha,k} - v_{\alpha^*,k}) + (U_{\alpha,k} - U_{\alpha^*,k}) + (b_{\alpha,k} - b_{\alpha^*,k})] \\
&\leq -M_1 - c\sum_{k=1}^n \gamma_k + \sum_{k=1}^n \gamma_k[(U_{\alpha,k} - U_{\alpha^*,k}) + (b_{\alpha,k} - b_{\alpha^*,k})] \\
&\leq -M_1 - c\sum_{k=1}^n \gamma_k + \sum_{k=1}^n \gamma_k[\xi_k + \chi_k] \tag{D.9}
\end{aligned}$$

where $\xi_k = (U_{\alpha,k} - U_{\alpha^*,k})$ and $\chi_k = 2\|b_k\|$. Now, similarly to the proofs of Theorems 1 and 2 we will proceed to control the aggregate error terms

$$R_n = \sum_{k=1}^n \gamma_k \xi_k \quad \text{and} \quad S_n = \sum_{k=1}^n \gamma_k \chi_k. \tag{D.10}$$

Since $\mathbb{E}[\xi_n \,|\, \mathcal{F}_n] = 0$, we have $\mathbb{E}[R_n \,|\, \mathcal{F}_n] = R_{n-1}$, so $R_n$ is a martingale; likewise, $\mathbb{E}[S_n \,|\, \mathcal{F}_n] \geq S_{n-1}$, so $S_n$ is a sub-martingale. Furthermore from (7) we have:

I. $\mathbb{E}[\xi_n^2] \leq \mathbb{E}[\|U_n\|^2] \leq \mathbb{E}[\mathbb{E}[\|U_n\|^2 \,|\, \mathcal{F}_n]] \leq \sigma_n^2$

II. $\mathbb{E}[\chi_n] = 2\,\mathbb{E}[\|b_n\|] \leq \mathbb{E}[\mathbb{E}[\|b_n\| \,|\, \mathcal{F}_n]] \leq B_n$

Moreover, for any $\eta_1 > 0$, we get by Doob's Maximal Inequality:

$$\mathbb{P}\left(\sup_{1 \leq k \leq n} R_k \geq \eta_1\right) \leq \frac{\mathbb{E}[R_n^2]}{\eta_1^2} \overset{(a)}{=} \frac{\sum_{k=1}^n \gamma_k^2 \, \mathbb{E}[\xi_k^2]}{\eta_1^2} \overset{(I)}{\leq} \frac{\sum_{k=1}^n \gamma_k^2 \sigma_k^2}{\eta_1^2} \tag{D.11}$$

where $(a)$ comes from the fact that $\mathbb{E}[\xi_i \xi_j] = 0$ for $i \neq j$. Since $\gamma_n = \gamma/n^p$, $\sigma_n = \mathcal{O}(n^{\ell_\sigma})$ and $p - \ell_\sigma > 1/2$, there exists $\gamma_1$ sufficiently small such that if $\gamma \leq \gamma_1$ then

$$\sum_{k=1}^\infty \gamma_k^2 \sigma_k^2 < \frac{\delta \eta_1^2}{2} \tag{D.12}$$

and so we automatically get that

$$\mathbb{P}\left(\sup_{1 \leq k \leq n} R_k \geq \eta_1\right) \leq \frac{\delta}{2} \tag{D.13}$$

Furthermore, notice that the term $\{S_n\}_{n\in\mathbb{N}}$ is a sub-martingale, since $\mathbb{E}[|S_n|\,|\,\mathcal{F}_n] < \infty$ and $\mathbb{E}[S_{n+1}\,|\,\mathcal{F}_n] > S_n$, for all $n$. As before, using Doob's Maximal Inequality, we get for any $\eta_2 > 0$:

$$\mathbb{P}\left(\sup_{1\leq k\leq n} S_k \geq \eta_2\right) \leq \frac{\mathbb{E}[S_n]}{\eta_2} = \frac{\sum_{k=1}^n \gamma_k \,\mathbb{E}[\chi_k]}{\eta_2} \leq \frac{2\sum_{k=1}^n \gamma_k B_k}{\eta_2} \tag{D.14}$$

So, since $p + \ell_b > 1$ there exists $\gamma_2$ sufficiently small such that if $\gamma \leq \gamma_2$ then

$$\sum_{k=1}^n \gamma_k B_k \leq \frac{\eta_2 \delta}{4} \tag{D.15}$$

which immidiately implies that

$$\mathbb{P}\left(\sup_{1\leq k\leq n} S_k \geq \eta_2\right) \leq \frac{\delta}{2} \tag{D.16}$$

By choosing $\gamma \leq \min\{\gamma_1, \gamma_2\}$ we get that

$$\mathbb{P}\left(\sup_{1\leq k\leq n} R_n + S_n \leq M_c\right) \geq 1 - \delta. \tag{D.17}$$

Notice now that by choosing $M_1 > M_c + \eta_1 + \eta_2$, from (D.9) we have that with probability at least $1 - \delta$, $y_{\alpha,n+1} - y_{\alpha^*,n+1} < -M_c$, which implies that $\pi_{n+1} \in \mathcal{U}_c$.

Defining the sequences of "good" events $\{\mathcal{E}_n\}_{n\in\mathbb{N}}$ and $\{\mathcal{E}'_n\}_{n\in\mathbb{N}}$ as $\mathcal{E}_n := \{\pi_k \in \mathcal{U}_c, \forall k = 1, \ldots, n\}$ and $\mathcal{E}'_n := \left\{\sup_{1\leq k\leq n} R_k + S_k \leq \eta_1 + \eta_2\right\}$, accordingly, we get that $\mathcal{E}'_n \subseteq \mathcal{E}_n$ for all $n$. Because $\mathbb{P}(\mathcal{E}'_n) \geq 1 - \delta$, we get that

$$\mathbb{P}(\mathcal{E}_n) \geq 1 - \delta \tag{D.18}$$

and since $\{\mathcal{E}_n\}_{n\in\mathbb{N}}$ is a decreasing sequence converging to $\mathcal{E} := \{\pi_n \in \mathcal{U}_c, \forall n \in \mathbb{N}\}$, we obtain

$$\mathbb{P}(\mathcal{E}) \geq 1 - \delta. \tag{D.19}$$

i.e.,

$$\mathbb{P}(\pi_n \in \mathcal{U}_c, \ \forall n \mid y_1 \in \mathcal{W}_1) \geq 1 - \delta \tag{D.20}$$

Notice that the above conclusions immediately imply convergence in finite time. More specifically, constrained to the event $\mathcal{E}$ with probability at least $1 - \delta$, from Eq. (D.9) we have

$$y_{\alpha,n+1} - y_{\alpha^*,n+1} \leq -M_c - c\sum_{k=1}^n \gamma_k \tag{D.21}$$

for all $n = 1, 2, \ldots$. Assume ad absurdum that there exists at least one strategy $\alpha \neq \alpha^*, \alpha \in \mathcal{A}$ such that $\limsup_{n\to\infty} \pi_{\alpha,n} \geq \varepsilon > 0$. for all sufficiently large $n$. Recall also that for $\pi \in \mathcal{U}_c$, it holds that $\pi_{\alpha^*} > 0$ by construction. Using Lemma D.1 we get

$$y_{\alpha,n+1} - y_{\alpha^*,n+1} = \pi_{\alpha,n+1} - \pi_{\alpha^*,n+1} \leq -M_c - c\sum_{k=1}^n \gamma_k \tag{D.22}$$

Notice that the L.H.S. of this inequality is bounded, while the R.H.S. goes to $-\infty$, which is a contradiction. Thus, with probability at least $1 - \delta$, $\pi_n \to \pi^*$ as $n \to \infty$.

We can rewrite the previous inequality as

$$\pi_{\alpha,n+1} \leq 1 - M_c - c\sum_{k=1}^n \gamma_k \quad \text{for all} \quad \alpha^* \neq \alpha \in \mathcal{A} \tag{D.23}$$

Now aggregating over all strategies, on the previous inequality, we get that

$$\|\pi_{n+1} - \pi^*\|_1 = 2(1 - \pi_{\alpha^*,n+1}) \leq 2\sum_{\alpha^*\neq\alpha\in\mathcal{A}} \left(1 - M_c - c\sum_{k=1}^n \gamma_k\right) \tag{D.24}$$

Thus, once $\sum_{k=1}^n \gamma_k$ becomes at least $(1 - M_c)/c$, which occurs in finite time, the convergence is implied. ∎

*Proof of Corollary 3.* For Models 1 and 2, taking $\ell_b = \infty, \ell_\sigma = 0$ we readily get that $p > 1/2$. Since we require that $\sum_{n=1}^\infty \gamma_n = \infty$, we obtain that $p \in (1/2, 1]$.

For Model 3, we have that $B_n = \mathcal{O}(\varepsilon_n)$ and $\sigma_n = \mathcal{O}(1/\sqrt{\varepsilon_n})$, i.e., $\ell_b = r$ and $\ell_\sigma = r/2$. Now, since $p \leq 1$, $p + \ell_b > 1$ and $p - \ell_\sigma > 1/2$, we obtain that $p \in (2/3, 1]$. ∎

# E    Structural properties of policy gradient methods

In this part of the appendix we will establish the necessary properties about the value function, its gradient. More precisely,
- In Lemma E.1 we prove that in the random stopping episodic framework visitation the notion of discounted state visitation distribution is well-defined.
- In Lemma 1, we prove the conversion lemma, a standard lemma that connects a sample by visitation distribution and a random trajectory.
- In Lemma E.4, we establish different versions of Policy Gradient theorem via $Q$-value function for the random stopping episodic framework.
- In Lemma E.5 and E.7, we establish the boundedness and the Lipschitz smoothness of policy gradient vector field, i.e., $v(\pi) = (v_i(\pi))_{i \in \mathcal{N}}$ where $v_i(\pi) = \nabla_{\pi_i} V_{i,s}(\pi)$

For a policy profile $\pi \in \Pi$ and an arbitrary initial state distribution $s_0 \sim \rho$, let's recall the definition of discounted state visitation measure/distribution as

$$\tilde{d}_\rho^\pi(s) = \mathbb{E}_{\tau \sim \mathrm{MDP}}\left[\sum_{t=0}^{T(\tau)} \mathbb{1}\{s_t = s\} \Big| s_0 \sim \rho\right], \quad d_\rho^\pi(s) := \tilde{d}_\rho^\pi(s)/Z_\rho^\pi$$

To begin with, we prove formally that the above definition is well-posed for the random stopping episodic framework described above, i.e., $\tilde{d}_\rho^\pi(s) < \infty$, so $Z_\rho^\pi := \sum_{s \in \mathcal{S}} \tilde{d}_\rho^\pi(s)$ is well-defined.

**Lemma E.1.** *For any $s \in \mathcal{S}$, $\tilde{d}_\rho^\pi(s) < \infty$ and $Z_\rho^\pi \leq \frac{1}{\zeta}$.*

*Proof.* For the sake of the proof, we define a new state $s_f$, indicating that the game has stopped. In other words, we have that $P(s_f \mid s, \alpha) = \zeta_{s,\alpha} \geq \zeta > 0$ for all $\alpha \in \mathcal{A}, s \in \mathcal{S}$. Hence, for $s \in \mathcal{S}$ we obtain:

$$\tilde{d}_\rho^\pi(s) = \mathbb{E}_{\tau \sim \mathrm{MDP}}\left[\sum_{t=0}^{T(\tau)} \mathbb{1}\{s_t = s\} \Big| s_0 \sim \rho\right] \tag{E.1}$$

$$= \mathbb{E}_{\tau \sim \mathrm{MDP}}\left[\sum_{t=0}^{\infty} \mathbb{1}\{s_t = s, s_i \neq s_f, 1 \leq i \leq t\} \mid s_0 \sim \rho\right] \tag{E.2}$$

$$\leq \sum_{s \in \mathcal{S}} \tilde{d}_\rho^\pi(s) \tag{E.3}$$

$$= \mathbb{E}_{\tau \sim \mathrm{MDP}}\left[\sum_{t=0}^{\infty} \mathbb{1}\{s_i \neq s_f, 1 \leq i \leq t\} \mid s_0 \sim \rho\right] \tag{E.4}$$

$$= \sum_{t=0}^{\infty} \mathbb{P}(s_i \neq s_f, 1 \leq i \leq t \mid s_0 \sim \rho) \tag{E.5}$$

$$= \sum_{t=0}^{\infty} \prod_{i=1}^{t} \mathbb{P}(s_i \neq s_f \mid s_0 \sim \rho, s_j \neq s_f, 1 \leq j \leq i-1) \tag{E.6}$$

$$\leq \sum_{t=0}^{\infty} (1 - \zeta)^t \leq \frac{1}{\zeta} \tag{E.7}$$

$$< \infty. \tag{E.8}$$

$\blacksquare$

**Lemma 1.** [Conversion Lemma] *For an arbitrary state-action function $f: \mathcal{S} \times \mathcal{A} \to \mathbb{R}$, a policy profile $\pi$ and an initial state distribution $s_0 \sim \rho$, we have*

$$\mathbb{E}_{\tau \sim \mathrm{MDP}}\left[\sum_{t=0}^{T(\tau)} f(s_t, \alpha_t)\right] = Z_\rho^\pi \, \mathbb{E}_{s \sim d_\rho^\pi} \mathbb{E}_{\alpha \sim \pi(\cdot|s)}[f(s, \alpha)] \tag{2}$$

*Proof.*

$$\mathbb{E}_{\tau \sim \mathrm{MDP}}\left[\sum_{t=0}^{T(\tau)} f(s_t, \alpha_t)\right] = \sum_{t=0}^{\infty} \sum_{s \in \mathcal{S}} \sum_{\alpha \in \mathcal{A}} \mathbb{E}_{\tau \sim \mathrm{MDP}}\left[\mathbb{1}\{t \leq T(\tau), s_t = s, \alpha_t = \alpha\} f(s, \alpha)\right]$$

$$= \sum_{s \in \mathcal{S}} \sum_{t=0}^{\infty} \sum_{\alpha \in \mathcal{A}} \mathbb{P}^{\pi}(s = s_t \mid s_0 \sim \rho) \pi(\alpha \mid s) f(s, \alpha)$$

$$= \sum_{s \in \mathcal{S}} \sum_{t=0}^{\infty} \mathbb{P}^{\pi}(s = s_t \mid s_0 \sim \rho) \sum_{\alpha \in \mathcal{A}} \pi(\alpha \mid s) f(s, \alpha)$$

$$= \sum_{s \in \mathcal{S}} \tilde{d}_{\rho}^{\pi}(s) \, \mathbb{E}_{\alpha \sim \pi(\cdot \mid s)} \left[ f(s, \alpha) \right]$$

$$= Z_{\rho}^{\pi} \, \mathbb{E}_{s \sim d_{\rho}^{\pi}} \, \mathbb{E}_{\alpha \sim \pi(\cdot \mid s)} [f(s, \alpha)] \tag{E.9}$$

where $Z_{\rho}^{\pi} := \mathbb{E}_{s \sim \text{Unif}(\mathcal{S})}[\tilde{d}_{\rho}^{\pi}(s)] \cdot |\mathcal{S}|$ is well-defined by E.1. $\blacksquare$

An equivalent but very useful way to describe compactly the aforementioned lemma is via the matrix representation of the discounted visitation distribution:

**Lemma E.2** ( Conversion Lemma (Matrix form) )**.** *For an arbitrary state-action function $f : \mathcal{S} \times \mathcal{A} \to \mathbb{R}$ and a policy profile $\pi$, we have*

$$\mathbb{E}_{\tau \sim \text{MDP}} \left[ \sum_{t=0}^{T(\tau)} f(s_t, \alpha_t) \mid \alpha_0 = \alpha, s_0 = s \right] = e_{s,\alpha}^{\top} \mathcal{T}(\pi) f \tag{E.10}$$

*where $\mathcal{T}$ is a discounted visitation distribution (action-state)-matrix under poliy profile $\pi$ i.e.,* $[\mathcal{T}(\pi)]\underbrace{(\alpha, s)}_{\text{Row Index}} \to \underbrace{(\alpha', s')}_{\text{Column Index}} = \sum_{t=0}^{\infty} \mathbb{P}^{\pi}(s_t = s', \alpha_t = \alpha' \mid s_0 = s, \alpha_0 = \alpha)$

*Proof.* By definition we have

$$e_{s,\alpha}^{\top} \mathcal{T}(\pi) f = \langle e_{s,\alpha}^{\top} \mathcal{T}(\pi), f \rangle \tag{E.11}$$

$$= \sum_{s' \in \mathcal{S}} \sum_{\alpha' \in \mathcal{A}} \left( e_{s,\alpha}^{\top} \mathcal{T}(\pi) \right)_{(s',\alpha')} \cdot f(s', \alpha') \tag{E.12}$$

$$= \sum_{s' \in \mathcal{S}} \sum_{\alpha' \in \mathcal{A}} e_{s,\alpha}^{\top} \mathcal{T}(\pi) e_{s',\alpha'} \cdot f(s', \alpha') \tag{E.13}$$

$$= \sum_{s' \in \mathcal{S}} \sum_{\alpha' \in \mathcal{A}} \sum_{t=0}^{\infty} \mathbb{P}^{\pi}(s_t = s', \alpha_t = \alpha' \mid s_0 = s, \alpha_0 = \alpha) \cdot f(s', \alpha') \tag{E.14}$$

$$= \sum_{t=0}^{\infty} \sum_{s' \in \mathcal{S}} \sum_{\alpha' \in \mathcal{A}} \mathbb{E}_{\tau \sim \text{MDP}} \left[ \mathbb{1}\{t \le T(\tau), s'_t = s, \alpha'_t = \alpha, \} f(s, \alpha) \mid s_0 = s, \alpha_0 = \alpha \right] \tag{E.15}$$

$$= \mathbb{E}_{\tau \sim \text{MDP}} \left[ \sum_{t=0}^{T(\tau)} f(s_t, \alpha_t) \mid \alpha_0 = \alpha, s_0 = s \right] \tag{E.16}$$

$\blacksquare$

*Remark* 1. Notice that $\mathcal{T}$ is a well-defined matrix. Indeed, let's us define $\mathcal{P}(\pi)$ as the state-action one step transition matrix:

$$[\mathcal{P}(\pi)]\underbrace{(\alpha, s)}_{\text{Row Index}} \to \underbrace{(\alpha', s')}_{\text{Column Index}} = \mathbb{P}^{\pi}(s_1 = s', \alpha_1 = \alpha' \mid s_0 = s, \alpha_0 = \alpha) = \pi(\alpha' \mid s') P(s' \mid s, \alpha).$$

Notice that $\mathcal{P}(\pi)$ is a substochastic matrix and therefore spectral($\mathcal{P}(\pi)$) $< 1$ or equivalently $(I - \mathcal{P}(\pi))^{-1}$ is invertible. Thus using Neumann series we have that $(I - \mathcal{P}(\pi))^{-1} = \sum_{t=0}^{\infty} \mathcal{P}(\pi)^t$. By induction, a folklore probabilistic-graph theoretic fact, we can show that $\sum_{t=0}^{\infty} \mathcal{P}(\pi)^t = \mathcal{T}(\pi)$.

In order to analyze the gradient of MARL policy gradient methods, we will introduce the notions $Q, A$ and their per-player averages that are useful in the MDP analysis.

**Definition 4.** For a state $s \in \mathcal{S}$, a policy $\pi$ and $\alpha = (\alpha_1, \dots, \alpha_N) \in \mathcal{A}$, we define:

(i) The $Q$-value function of player $i$ as:

$$Q_i^{\pi}(s, \alpha) := \mathbb{E}_{\tau \sim \text{MDP}(\pi \mid s)} \left[ \sum_{t=0}^{T(\tau)} R_i(s_t(\tau), \alpha_t(\tau)) \mid s_0 = s, \alpha_0 = \alpha \right] \tag{E.17}$$

903 (ii) The *Advantage*-function of player $i$ as:

$$A_i^\pi(s, \alpha) := Q_i^\pi(s, \alpha) - V_{i,s}(\pi) \tag{E.18}$$

904 We also define $\overline{Q_i^\pi}, \overline{A_i^\pi}$ to be the averaged for $i$-th player single MDP $Q$-value and advantage functions:

905 (i) The averaged $\overline{Q_i^\pi}$-value function of player $i$ as:

$$\overline{Q_i^\pi}(s, \alpha_i) := \mathbb{E}_{\alpha_{-i} \sim \pi_{-i}(\cdot|s)} \left[ Q_i^\pi(s, (\alpha_i; \alpha_{-i})) \right] \tag{E.19}$$

906 (ii) The averaged *Advantage* , $\overline{A_i^\pi}$-function of player $i$ as:

$$\overline{A_i^\pi}(s, \alpha_i) := \mathbb{E}_{\alpha_{-i} \sim \pi_{-i}(\cdot|s)} \left[ A_i^\pi(s, (\alpha_i; \alpha_{-i})) \right], \tag{E.20}$$

907 Using Remark 1, we can rewrite the above notations using $\mathcal{T}, \mathcal{P}$.

908 **Lemma E.3.** *For a policy profile $\pi$, we have that*

909 *1. $Q_i^\pi(s, \alpha) = e_{s,\alpha}^\top \mathcal{T}(\pi) r_i$*

910 *2. $\tilde{d}_\rho^\pi(s) = [\sum_{s' \in \mathcal{S}} \rho(s') \sum_{\alpha' \in \mathcal{A}} \pi(\alpha' \mid s') e_{s',\alpha'}]^\top \mathcal{T}(\pi) \sum_{\alpha \in \mathcal{A}} e_{s,\alpha}$*

911 *Proof.* We separately have using Lemma E.3 and Remark 1.

912 1. $Q_i^\pi(s, \alpha) = \mathbb{E}_{\tau \sim \text{MDP}(\pi|s)} \left[ \sum_{t=0}^{T(\tau)} R_i(s_t(\tau), \alpha_t(\tau)) \mid s_0 = s, \alpha_0 = \alpha \right] = e_{s,\alpha}^\top \mathcal{T}(\pi) R_i$

  2.

$$\tilde{d}_\rho^\pi(s) = \mathbb{E}_{\tau \sim \text{MDP}} \left[ \sum_{t=0}^{T(\tau)} \mathbb{1}\{s_t = s\} \Big| s_0 \sim \rho \right] \tag{E.21}$$

$$= \mathbb{E}_{s' \sim \rho} \mathbb{E}_{\tau \sim \text{MDP}} \left[ \sum_{t=0}^{T(\tau)} \sum_{\alpha \in \mathcal{A}} \mathbb{1}\{s_t = s, \alpha_t = \alpha\} \Big| s_0 = s' \right] \tag{E.22}$$

$$= \mathbb{E}_{s' \sim \rho} \mathbb{E}_{\alpha' \sim \pi(\cdot|s)} \mathbb{E}_{\tau \sim \text{MDP}} \left[ \sum_{t=0}^{T(\tau)} \sum_{\alpha \in \mathcal{A}} \mathbb{1}\{s_t = s, \alpha_t = \alpha\} \Big| s_0 = s', \alpha_0 = \alpha' \right] \tag{E.23}$$

$$= \mathbb{E}_{s' \sim \rho} \mathbb{E}_{\alpha' \sim \pi(\cdot|s)} \left[ e_{s',\alpha'}^\top \mathcal{T}(\pi) \sum_{\alpha \in \mathcal{A}} e_{s,\alpha} \right] \tag{E.24}$$

$$= \left[ \sum_{s' \in \mathcal{S}} \rho(s') \sum_{\alpha' \in \mathcal{A}} \pi(\alpha' \mid s') e_{s',\alpha'} \right]^\top \mathcal{T}(\pi) \sum_{\alpha \in \mathcal{A}} e_{s,\alpha} \tag{E.25}$$

913                                      ■

914 Having defined the above notions, we are ready to provide equivalent forms of the $v(\pi)$ operator that
915 will permit us to prove its boundedness and smoothness. We start with the following versions of
916 Policy gradient theorem for random stopping setting:

917 **Lemma E.4.** *For the independent gradient operator $v(\pi)$ per player the following expressions are*
918 *equal to $v_i(\pi)$:*

919 *1. $v_i(\pi) = \mathbb{E}_{\tau \sim \text{MDP}} \left[ \sum_{t=0}^{T(\tau)} \nabla_i (\log \pi_i(\alpha_{i,t}(\tau) \mid s_t(\tau))) \overline{Q_i^\pi}(s_t(\tau), \alpha_{i,t}(\tau)) \right]$*

920 *2. $v_i(\pi) = Z_\rho^\pi \mathbb{E}_{s \sim d_\rho^\pi} \mathbb{E}_{\alpha_i \sim \pi_i(\cdot|s)} \left[ \nabla_i (\log \pi_i(\alpha_i \mid s)) \overline{Q_i^\pi}(s, \alpha_i) \right]$*

921 *3. $(v_i(\pi))_{\alpha_i^\circ, s^\circ} = \frac{\partial V_{i,\rho}(\pi)}{\partial \pi_i(\alpha_i^\circ | s^\circ)} = \tilde{d}_\rho^\pi(s^\circ) \overline{Q_i^\pi}(s^\circ, \alpha_i^\circ) = Z_\rho^\pi d_\rho^\pi(s^\circ) \overline{Q_i^\pi}(s^\circ, \alpha_i^\circ)$*

922 *Proof.* Let as recall again the definition of our independent gradient operator $v(\pi)$:

$$v_i(\pi) = \nabla_i V_{i,\rho}(\pi)$$

First, we will show that:

$$\nabla_i \left( V_{i,\rho}(\pi) \right) = \mathbb{E}_{\tau \sim \text{MDP}} \left[ \sum_{t=0}^{T(\tau)} \nabla_i \left( \log \pi_i(\alpha_{i,t}(\tau) \mid s_t(\tau)) \right) \overline{Q_i^\pi}(s_t(\tau), \alpha_{i,t}(\tau)) \right] \tag{E.26}$$

We will start with an arbitrary $s_0$, and by linearity of $\nabla_{\pi_i}(\cdot)$ and $\mathbb{E}_{s_0 \sim \rho}[\cdot]$, we will obtain the result.

$$
\begin{aligned}
\nabla_i \left( V_{i,s_0}(\pi) \right) &= \nabla_i \left( \mathbb{E}_\tau [R_i(\tau)] \right) \\
&= \nabla_i \left( \mathbb{E}_{\alpha_i \sim \pi_i(\cdot|s_0)} \left[ \overline{Q_i^\pi}(s_0, \alpha_i) \right] \right) \\
&= \nabla_i \left( \sum_{\alpha_i \in \mathcal{A}_i} \pi_i(\alpha_i \mid s_0) \overline{Q_i^\pi}(s_0, \alpha_i) \right) \\
&= \sum_{\alpha_i \in \mathcal{A}_i} \nabla_i \left( \pi_i(\alpha_i \mid s_0) \right) \overline{Q_i^\pi}(s_0, \alpha_i) + \pi_i(\alpha_i \mid s_0) \nabla_i \left( \overline{Q_i^\pi}(s_0, \alpha_i) \right) \\
&= \sum_{\alpha_i \in \mathcal{A}_i} \nabla_i \left( \log \pi_i(\alpha_i \mid s_0) \right) \pi_i(\alpha_i \mid s_0) \overline{Q_i^\pi}(s_0, \alpha_i) + \pi_i(\alpha_i \mid s_0) \nabla_i \left( \overline{Q_i^\pi}(s_0, \alpha_i) \right) \\
&= \mathbb{E}_{\alpha_i \sim \pi_i(\cdot|s_0)} \left[ \nabla_i \left( \log \pi_i(\alpha_i \mid s_0) \right) \overline{Q_i^\pi}(s_0, \alpha_i) \right] \\
&\quad + \sum_{\alpha_i \in \mathcal{A}_i} \pi_i(\alpha_i \mid s_0) \nabla_i \left( \mathbb{E}_{\alpha_{-i} \sim \pi_{-i}(\cdot|s_0)} \left[ R_i(s_0, \alpha) + \sum_{s_1 \in \mathcal{S}} P(s_1 \mid s_0, \alpha) V_{i,s_1}(\pi) \right] \right) \\
&= \mathbb{E}_{\alpha_i \sim \pi_i(\cdot|s_0)} \left[ \nabla_i \left( \log \pi_i(\alpha_i \mid s_0) \right) \overline{Q_i^\pi}(s_0, \alpha_i) \right] \\
&\quad + \sum_{\alpha_i \in \mathcal{A}_i} \pi_i(\alpha_i \mid s_0) \mathbb{E}_{\alpha_{-i} \sim \pi_{-i}(\cdot|s_0)} \left[ \sum_{s_1 \in \mathcal{S}} P(s_1 \mid s_0, \alpha) \nabla_i \left( V_{i,s_1}(\pi) \right) \right] \\
&= \mathbb{E}_{\alpha_i \sim \pi_i(\cdot|s_0)} \left[ \nabla_i \left( \log \pi_i(\alpha_i \mid s_0) \right) \overline{Q_i^\pi}(s_0, \alpha_i) \right] \\
&\quad + \mathbb{E}_{\alpha \sim \pi(\cdot|s_0)} \left[ \sum_{s_1 \in \mathcal{S}} P(s_1 \mid s_0, \alpha) \nabla_i \left( V_{i,s_1}(\pi) \right) \right]
\end{aligned}
\tag{E.27}
$$

Thus, we can rewrite it as:

$$
\begin{aligned}
\nabla_i \left( V_{i,s_0}(\pi) \right) &= \mathbb{E}_{\alpha_i \sim \pi_i(\cdot|s_0)} \left[ \nabla_i \left( \log \pi_i(\alpha_i \mid s_0) \right) \overline{Q_i^\pi}(s_0, \alpha_i) \right] \\
&\quad + \mathbb{E}_{\alpha \sim \pi(\cdot|s_0)} \left[ \sum_{s_1 \in \mathcal{S}} P(s_1 \mid s_0, \alpha) \nabla_i \left( V_{i,s_1}(\pi) \right) \right] \\
&= \mathbb{E}_{\tau \sim \text{MDP}(\pi|s_0)} \left[ \nabla_i \left( \log \pi_i(\alpha_{i,0}(\tau) \mid s_0) \right) \overline{Q_i^\pi}(s_0, \alpha_{i,0}(\tau)) \right] \\
&\quad + \mathbb{E}_{\tau \sim \text{MDP}(\pi|s_0)} \left[ \mathbb{1} \left\{ T(\tau) \geq 1 \right\} \nabla_i \left( V_{i,s_1(\tau)}(\pi) \right) \right] \\
&= \sum_{t=0}^\infty \mathbb{E}_{\tau \sim \text{MDP}(\pi|s_0)} \left[ \mathbb{1}\{t \leq T(\tau)\} \nabla_i \left( \log \pi_i(\alpha_{i,t}(\tau) \mid s_t(\tau)) \right) \overline{Q_i^\pi}(s_t(\tau), \alpha_{i,t}(\tau)) \right] \\
&\quad + \mathbb{E}_{\tau \sim \text{MDP}(\pi|s_0)} \left[ \mathbb{1}\{T(\tau) = \infty\} A_\infty \right] \\
&\overset{(a)}{=} \mathbb{E}_{\tau \sim \text{MDP}(\pi|s_0)} \left[ \sum_{t=0}^{T(\tau)} \nabla_i \left( \log \pi_i(\alpha_{i,t}(\tau) \mid s_t(\tau)) \right) \overline{Q_i^\pi}(s_t(\tau), \alpha_{i,t}(\tau)) \right]
\end{aligned}
\tag{E.28}
$$

where $(a)$ holds because $\mathbb{P}(T(\tau) = \infty) = 0$, and $A_\infty$ is some limiting quantity.

Hence, we readily obtain:

$$\nabla_i \left( V_{i,\rho}(\pi) \right) = \mathbb{E}_{s_0 \sim \rho} \left[ \nabla_i \left( V_{i,s_0}(\pi) \right) \right] \tag{E.29}$$

Now we are ready to utilize our Lemma 1:

$$\nabla_i \left( V_{i,\rho}(\pi) \right) = Z_\rho^\pi \, \mathbb{E}_{s \sim d_\rho^\pi} \, \mathbb{E}_{\alpha \sim \pi(\cdot|s)} \left[ \nabla_i \left( \log \pi_i(\alpha_i \mid s) \right) \overline{Q_i^\pi}(s, \alpha_i) \right] \tag{E.30}$$

$$= Z_\rho^\pi \, \mathbb{E}_{s \sim d_\rho^\pi} \, \mathbb{E}_{\alpha_i \sim \pi_i(\cdot|s)} \Big[ \nabla_i \left( \log \pi_i(\alpha_i \mid s) \right) \overline{Q_i^\pi}(s, \alpha_i) \Big] \tag{E.31}$$

Decoupling $\nabla_i$ per a state $s^\circ$ and action $\alpha_i^\circ$, we get

$$\frac{\partial V_{i,\rho}(\pi)}{\partial \pi_i(\alpha_i^\circ \mid s^\circ)} = Z_\rho^\pi \, \mathbb{E}_{s \sim d_\rho^\pi} \, \mathbb{E}_{\alpha_i \sim \pi_i(\cdot|s)} \left[ \frac{\partial \left( \log \pi_i(\alpha_i \mid s) \right)}{\partial \pi_i(\alpha_i^\circ \mid s^\circ)} \overline{Q_i^\pi}(s, \alpha_i) \right] \tag{E.32}$$

$$= Z_\rho^\pi \, \mathbb{E}_{s \sim d_\rho^\pi} \, \mathbb{E}_{\alpha_i \sim \pi_i(\cdot|s)} \left[ \mathbb{1}\{\alpha_i^\circ = \alpha_i, s^\circ = s\} \frac{1}{\pi_i(\alpha_i^\circ \mid s^\circ)} \overline{Q_i^\pi}(s^\circ, \alpha_i^\circ) \right] \tag{E.33}$$

$$= \sum_{s \in \mathcal{S}} \tilde{d}_\rho^\pi(s) \sum_{\alpha_i \in \mathcal{A}_i} \pi_i(\alpha_i \mid s) \mathbb{1}\{\alpha_i^\circ = \alpha_i, s^\circ = s\} \frac{1}{\pi_i(\alpha_i^\circ \mid s^\circ)} \overline{Q_i^\pi}(s^\circ, \alpha_i^\circ) \tag{E.34}$$

$$= \tilde{d}_\rho^\pi(s^\circ) \overline{Q_i^\pi}(s^\circ, \alpha_i^\circ) = Z_\rho^\pi d_\rho^\pi(s^\circ) \overline{Q_i^\pi}(s^\circ, \alpha_i^\circ) \tag{E.35}$$

∎

We are ready to bound the amplitude of the independent player gradient operator:

**Lemma E.5.** *For a given initial state distribution $\rho$, the independent player policy gradient operator $v(\pi)$ is bounded. More precisely,*

$$\|v_i(\pi)\| \le \frac{\sqrt{|\mathcal{A}_i|}}{\zeta^2} \quad \& \quad \|v(\pi)\| \le \frac{\sum_{i \in \mathcal{N}} \sqrt{|\mathcal{A}_i|}}{\zeta^2}$$

*Proof.* We start by analyzing $\|v_i(\pi)\|^2$ using the aforementioned Lemma E.4.

$$\|v_i(\pi)\|^2 = \sum_{\alpha_i^\circ, s^\circ, \in \mathcal{A}_i, \mathcal{S}} (v_i(\pi)_{\alpha_i^\circ, s^\circ})^2$$

$$= \sum_{s^\circ \in \mathcal{S}} \sum_{\alpha_i^\circ \in \mathcal{A}_i} \left( \frac{\partial V_{i,\rho}(\pi)}{\partial \pi_i(\alpha_i^\circ \mid s^\circ)} \right)^2$$

$$= \sum_{s^\circ \in \mathcal{S}} \sum_{\alpha_i^\circ \in \mathcal{A}_i} (Z_\rho^\pi d_\rho^\pi(s^\circ) \overline{Q_i^\pi}(s^\circ, \alpha_i^\circ))^2$$

$$\le (Z_\rho^\pi)^2 \max_{\alpha_i^\circ, s^\circ, \in \mathcal{A}_i, \mathcal{S}} (\overline{Q_i^\pi}(s^\circ, \alpha_i^\circ))^2 \sum_{s^\circ \in \mathcal{S}} \sum_{\alpha_i^\circ \in \mathcal{A}_i} d_\rho^\pi(s^\circ)^2$$

$$\le \frac{1}{\zeta^2} \max_{\alpha_i^\circ, s^\circ, \in \mathcal{A}_i, \mathcal{S}} (\mathbb{E}_{\alpha_{-i} \sim \pi_{-i}(\cdot|s)} \left[ Q_i^\pi \left( s^\circ, (\alpha_i^\circ; \alpha_{-i}) \right) \right])^2 \sum_{s^\circ \in \mathcal{S}} \sum_{\alpha_i^\circ \in \mathcal{A}_i} d_\rho^\pi(s^\circ)$$

$$\le \frac{1}{\zeta^2} \max_{\alpha^\circ, s^\circ, \in \mathcal{A}, \mathcal{S}} (Q_i^\pi(s^\circ, \alpha^\circ))^2 \sum_{\alpha_i^\circ \in \mathcal{A}_i} \sum_{s^\circ \in \mathcal{S}} d_\rho^\pi(s^\circ)$$

$$\le \frac{1}{\zeta^2} \max_{\alpha^\circ, s^\circ, \in \mathcal{A}, \mathcal{S}} \left( \mathbb{E}_{\tau \sim \mathrm{MDP}(\pi|s)} \left[ \sum_{t=0}^{T(\tau)} R_i(s_t(\tau), \alpha_t(\tau)) \mid s_0 = s^\circ, \alpha_0 = \alpha^\circ \right] \right)^2 \sum_{\alpha_i^\circ \in \mathcal{A}_i} 1$$

$$\le \frac{|\mathcal{A}_i|}{\zeta^2} \left( \mathbb{E}_{\tau \sim \mathrm{MDP}(\pi|s)} \left[ \sum_{t=0}^{T(\tau)} 1 \mid s_0 = s^\circ, \alpha_0 = \alpha^\circ \right] \right)^2$$

$$\le \frac{|\mathcal{A}_i|}{\zeta^4}$$

Thus we conclude that

$$\|v_i(\pi)\| \le \frac{\sqrt{|\mathcal{A}_i|}}{\zeta^2} \quad \& \quad \|v(\pi)\| \le \frac{\sum_{i \in \mathcal{N}} \sqrt{|\mathcal{A}_i|}}{\zeta^2}$$

∎

To prove the smoothness of the policy gradient operator, we have first to establish the performance lemma for our setting. Respectively, we get

**Lemma E.6** (Performance lemma). *For any pair of policy profiles $\pi = (\pi_i, \pi_{-i}), \pi' = (\pi'_i, \pi'_{-i})$, it holds*

$$V_{i,\rho}(\pi_i, \pi_{-i}) - V_{i,\rho}(\pi'_i, \pi'_{-i}) = \mathbb{E}_{\tau \sim \text{MDP}(\pi|\rho)}\left[\sum_{t=0}^{T(\tau)} A_i^{\pi'_i, \pi'_{-i}}(s_t, \alpha_t)\right] \tag{E.36}$$

*where $\text{MDP}(\pi|\rho)$ signifies that players follow $\pi$ as policy profile with $\rho$ as the initial state distribution.*

*Proof.* We will initial prove the aforementioned result for an arbitrary deterministic initial state $s_0 = s$:

$$V_{i,s}(\pi) - V_{i,s}(\pi') = \mathbb{E}_{\tau \sim \text{MDP}(\pi|\rho)}\left[\sum_{t=0}^{T(\tau)} R_i(s_t, \alpha_t)\right] - V_{i,s}(\pi') \tag{E.37}$$

$$= \mathbb{E}_{\tau \sim \text{MDP}(\pi|s)}\left[\sum_{t=0}^{T(\tau)} \left(R_i(s_t, \alpha_t) + V_{i,s_t}(\pi') - V_{i,s_t}(\pi')\right)\right] - V_{i,s}(\pi') \tag{E.38}$$

$$= \mathbb{E}_{\tau \sim \text{MDP}(\pi|s)}\left[\sum_{t=0}^{T(\tau)} R_i(s_t, \alpha_t) + \sum_{t=0}^{T(\tau)} \left(V_{i,s_t}(\pi') - V_{i,s}(\pi') - V_{i,s_t}(\pi')\right)\right] \tag{E.39}$$

$$= \mathbb{E}_{\tau \sim \text{MDP}(\pi|s)}\left[\sum_{t=0}^{T(\tau)} \left(R_i(s_t, \alpha_t) + \mathbb{1}\{T(\tau) \geq t+1\}V_{i,s_{t+1}}(\pi')\right) - V_{i,s_t}(\pi')\right] \tag{E.40}$$

$$= \mathbb{E}_{\tau \sim \text{MDP}(\pi|s)}\left[\sum_{t=0}^{T(\tau)} \left(Q_i^{\pi'}(s_t, \alpha_t) - V_{i,s_t}(\pi')\right)\right] \tag{E.41}$$

$$= \mathbb{E}_{\tau \sim \text{MDP}(\pi|s)}\left[\sum_{t=0}^{T(\tau)} A_i^{\pi'}(s_t, \alpha_t)\right] \tag{E.42}$$

where in the last equation we recall the definition of the Advantage function and in the pre-last the equivalent definitions of $Q_i^{\pi}(s, \alpha)$

$$Q_i^{\pi}(s, \alpha) = \mathbb{E}_{\tau \sim \text{MDP}(\pi|s)}\left[\sum_{t=0}^{T(\tau)} R_i(s_t(\tau), \alpha_t(\tau)) \mid s_0 = s, \alpha_0 = \alpha\right]$$

$$= R_i(s, \alpha) + \mathbb{E}_{\tau \sim \text{MDP}(\pi|s)}\left[\mathbb{1}\{T(\tau) \geq 1\}V_{i,s_1}(\pi) \mid s_0 = s, \alpha_0 = \alpha\right] \tag{E.43}$$

Applying the linearity of $\mathbb{E}_{s \sim \rho}[\cdot]$, we get the desired result:

$$V_{i,\rho}(\pi) - V_{i,\rho}(\pi') = \mathbb{E}_{\tau \sim \text{MDP}(\pi|\rho)}\left[\sum_{t=0}^{T(\tau)} A_i^{\pi'}(s_t, \alpha_t)\right] = Z_\rho^\pi \mathbb{E}_{s \sim d_\rho^\pi} \mathbb{E}_{\alpha \sim \pi(\cdot|s)}\left[A_i^{\pi'}(s, \alpha)\right] \tag{E.44}$$

where the last expression comes from Lemma 1. ∎

Before closing this section by proving the Lipschitz-smoothness of our operator, we describe a useful observation that would be helpful in the smoothness bounds.

**Proposition E.1.** *For any pair of policy profiles $\pi = (\pi_i, \pi_{-i}), \pi' = (\pi'_i, \pi'_{-i})$ and an arbitrary initial state distribution $\rho$ and a subset $\mathcal{M} \subseteq \mathcal{N}$, it holds that:*

$$\sum_s d_\rho^\pi(s) \sum_{\alpha_\mathcal{M}} |(\pi_\mathcal{M} - \pi'_\mathcal{M})(\alpha_\mathcal{M} \mid s)| \leq \sum_{i \in \mathcal{M}} \sqrt{|\mathcal{A}_i|}\|\pi_i - \pi'_i\|$$

*where $\pi_\mathcal{M} = (\pi_i)_{i \in \mathcal{M}}$ and $\alpha_\mathcal{M} = (\alpha_i)_{i \in \mathcal{M}}$, correspondingly.*

*Proof.*

$$\sum_s d_\rho^\pi(s) \sum_{\alpha_\mathcal{M}} |(\pi_\mathcal{M} - \pi'_\mathcal{M})(\alpha_\mathcal{M} \mid s)| = 2 \sum_s d_\rho^\pi(s) \frac{1}{2}\|(\pi_\mathcal{M} - \pi'_\mathcal{M})\|_1 \tag{E.45}$$

$$= 2 \sum_s d_\rho^\pi(s) \frac{1}{2} d_{\mathrm{TV}}(\pi_{\mathcal{M}}(\cdot|s), \pi_{\mathcal{M}}'(\cdot|s)) \tag{E.46}$$

$$\leq 2 \sum_s d_\rho^\pi(s) \sum_{i \in \mathcal{M}} \frac{1}{2} d_{\mathrm{TV}}(\pi_i(\cdot|s), \pi_i'(\cdot|s)) \tag{E.47}$$

$$= \sum_s d_\rho^\pi(s) \sum_{i \in \mathcal{M}} \|(\pi_i(\cdot|s) - \pi_i'(\cdot|s))\|_1 \tag{E.48}$$

$$= \sum_s d_\rho^\pi(s) \sum_{i \in \mathcal{M}} \sqrt{|\mathcal{A}_i|} \|\pi_i - \pi_i'\|_2 \tag{E.49}$$

$$= \sum_{i \in \mathcal{M}} \sqrt{|\mathcal{A}_i|} \|\pi_i - \pi_i'\|_2 \left( \sum_s d_\rho^\pi(s) \right) \tag{E.50}$$

$$= \sum_{i \in \mathcal{M}} \sqrt{|\mathcal{A}_i|} \|\pi_i - \pi_i'\|_2 \tag{E.51}$$

where $d_{\mathrm{TV}}$ corresponds to the total variation distance. Indeed notice that $d_{\mathrm{TV}}$ actually equals to the normalized difference of the histograms between two distributions. Additionally, the first inequality is derived by the "triangle inequality" that holds for $d_{\mathrm{TV}}$ in product-measure distributions. ∎

**Lemma E.7.** *For a given initial state distribution $\rho$, the independent player policy gradient operator $v(\pi)$ is lipschitz-smooth. More precisely, for any pair of policy profiles $\pi = (\pi_i, \pi_{-i}), \pi' = (\pi_i', \pi_{-i}')$, it holds*

$$\|v_i(\pi) - v_i(\pi')\| = \|\nabla_i(V_{i,\rho}(\pi) - \nabla_i(V_{i,\rho}(\pi')\| \leq \frac{3\sqrt{|\mathcal{A}_i|}}{\zeta^3} \sum_{j=1}^N \sqrt{|\mathcal{A}_i|} \|\pi_j - \pi_j'\| \quad \forall i \in \mathcal{N}$$

*and consequently,*

$$\|v(\pi) - v(\pi')\| \leq \frac{3|\mathcal{A}|}{\zeta^3} \|\pi - \pi'\|$$

*Proof.* For the proof, we will follow the approach of Zhang et al. [68] and Agarwal et al. [1]. Our first task is to bound the directional derivative of the $i$-th player's value function. We start by setting some notation. Let $\pi, \pi' \in \Pi$ and pert $\in \mathcal{S} \times \mathcal{A}$ such that $\|\text{pert}\| = 1$. Then, we define the following $\lambda$-almost perturbed policies:

$$\begin{cases} \pi_\lambda^{\mathbb{A}}(\alpha \mid s) = (\pi_i + \lambda\text{pert}, \pi_{-i}) \\ \pi_\lambda^{\mathbb{B}}(\alpha \mid s) = (\pi_i' + \lambda\text{pert}, \pi_{-i}') \end{cases}$$

$$\left| \frac{\partial V_{i,\rho}(\pi_\lambda^{\mathbb{A}})}{\partial \lambda} - \frac{\partial V_{i,\rho}(\pi_\lambda^{\mathbb{B}})}{\partial \lambda} \right| = \left| \frac{\partial V_{i,\rho}(\pi_\lambda^{\mathbb{A}}) - V_{i,\rho}(\pi_\lambda^{\mathbb{B}})}{\partial \lambda} \right| \tag{E.52}$$

$$= \left| \frac{\partial \left( V_{i,\rho}(\pi_\lambda^{\mathbb{A}}) - V_{i,\rho}(\pi_\lambda^{\mathbb{B}}) \right)}{\partial \lambda} \right| \tag{E.53}$$

$$= \left| \frac{\partial \left( Z_\rho^{\pi_\lambda^{\mathbb{A}}} \mathbb{E}_{s \sim d_\rho^{\pi_\lambda^{\mathbb{A}}}} \mathbb{E}_{\alpha \sim \pi_\lambda^{\mathbb{A}}(\cdot|s)} \left[ A_i^{\pi_\lambda^{\mathbb{B}}}(s, \alpha) \right] \right)}{\partial \lambda} \right| \tag{E.54}$$

$$= \left| \frac{\partial \left( Z_\rho^{\pi_\lambda^{\mathbb{A}}} \mathbb{E}_{s \sim d_\rho^{\pi_\lambda^{\mathbb{A}}}} \mathbb{E}_{\alpha \sim \pi_\lambda^{\mathbb{A}}(\cdot|s)} \left[ A_i^{\pi_\lambda^{\mathbb{B}}}(s, \alpha) \right] \right)}{\partial \lambda} \right| \tag{E.55}$$

$$= \left| \frac{\partial \left( Z_\rho^{\pi_\lambda^{\mathbb{A}}} \sum_{s,\alpha} d_\rho^{\pi_\lambda^{\mathbb{A}}}(s)(\pi_\lambda^{\mathbb{A}} - \pi_\lambda^{\mathbb{B}})(\alpha \mid s) A_i^{\pi_\lambda^{\mathbb{B}}}(s, \alpha) \right)}{\partial \lambda} \right| \tag{E.56}$$

$$= \left| \frac{\partial \left( Z_\rho^{\pi_\lambda^{\mathbb{A}}} \sum_{s,\alpha} d_\rho^{\pi_\lambda^{\mathbb{A}}}(s)(\pi_\lambda^{\mathbb{A}} - \pi_\lambda^{\mathbb{B}})(\alpha \mid s) Q_i^{\pi_\lambda^{\mathbb{B}}}(s,\alpha) \right)}{\partial \lambda} \right| \qquad (E.57)$$

$$= \left| \frac{\partial \left( \sum_{s,\alpha} \tilde{d}_\rho^{\pi_\lambda^{\mathbb{A}}}(s)(\pi_\lambda^{\mathbb{A}} - \pi_\lambda^{\mathbb{B}})(\alpha \mid s) Q_i^{\pi_\lambda^{\mathbb{B}}}(s,\alpha) \right)}{\partial \lambda} \right| \qquad (E.58)$$

965 where (E.54) leverages the Performance Lemma E.6 and (E.56) uses the fact $\sum_{\alpha \in \mathcal{A}} \pi(\alpha \mid s) A_i^\pi(s,\alpha) =$,
966 for all $s \in \mathcal{S}$ and the last one is derived by the definition $d_\rho^\pi(s) := \tilde{d}_\rho^\pi(s)/Z_\rho^\pi$.

967 By triangular inequality, the linearity of $\partial$ operator and Lemma E.1, we have:

$$\left| \frac{\partial(V_{i,\rho}(\pi_\lambda^{\mathbb{A}}) - V_{i,\rho}(\pi_\lambda^{\mathbb{B}}))}{\partial \lambda} \right|_{\lambda=0} \right| \leq \left| \sum_{s,\alpha} \frac{\partial \tilde{d}_\rho^{\pi_\lambda^{\mathbb{A}}}(s)}{\partial \lambda} \right|_{\lambda=0} (\pi - \pi^{'})(\alpha \mid s) Q_i^{\pi^{'}}(s,\alpha) \right|$$

$$+ Z_\rho^{\pi^{\mathbb{A}}} \left| \sum_{s,\alpha} d_\rho^\pi(s) \frac{\partial(\pi_\lambda^{\mathbb{A}} - \pi_\lambda^{\mathbb{B}})(\alpha \mid s)}{\partial \lambda} \right|_{\lambda=0} Q_i^{\pi^{'}}(s,\alpha) \right|$$

$$+ Z_\rho^{\pi^{\mathbb{A}}} \left| \sum_{s,\alpha} d_\rho^\pi(s)(\pi - \pi^{'})(\alpha \mid s) \frac{\partial Q_i^{\pi_\lambda^{\mathbb{B}}}(s,\alpha)}{\partial \lambda} \right|_{\lambda=0} \right| \qquad (E.59)$$

968 We will bound the following three terms separately:

$$\begin{cases} \text{Term}_A = \left| \sum_{s,\alpha} \frac{\partial \tilde{d}_\rho^{\pi_\lambda^{\mathbb{A}}}(s)}{\partial \lambda} \right|_{\lambda=0} (\pi - \pi^{'})(\alpha \mid s) Q_i^{\pi^{'}}(s,\alpha) \right| \\ \text{Term}_B = \left| \sum_{s,\alpha} d_\rho^\pi(s) \frac{\partial(\pi_\lambda^{\mathbb{A}} - \pi_\lambda^{\mathbb{B}})(\alpha|s)}{\partial \lambda} \right|_{\lambda=0} Q_i^{\pi^{'}}(s,\alpha) \right| \\ \text{Term}_C = \left| \sum_{s,\alpha} d_\rho^\pi(s)(\pi - \pi^{'})(\alpha \mid s) \frac{\partial Q_i^{\pi_\lambda^{\mathbb{B}}}(s,\alpha)}{\partial \lambda} \right|_{\lambda=0} \right| \end{cases}$$

969 For $\text{Term}_A$, we will use Lemma E.3 in order to compute compactly the derivative:

$$\frac{\partial \tilde{d}_\rho^{\pi_\lambda^{\mathbb{A}}}(s)}{\partial \lambda} = \frac{\partial \left( \left[ \sum_{s' \in \mathcal{S}} \rho(s') \sum_{\alpha' \in \mathcal{A}} \pi_\lambda^{\mathbb{A}}(\alpha' \mid s') e_{s',\alpha'} \right]^\top \mathcal{T}(\pi_\lambda^{\mathbb{A}}) \sum_{\alpha \in \mathcal{A}} e_{s,\alpha} \right)}{\partial \lambda} \qquad (E.60)$$

$$= \left( \left[ \sum_{s' \in \mathcal{S}} \rho(s') \sum_{\alpha' \in \mathcal{A}} \frac{\partial \pi_\lambda^{\mathbb{A}}(\alpha' \mid s')}{\partial \lambda} e_{s',\alpha'} \right]^\top \mathcal{T}(\pi_\lambda^{\mathbb{A}}) \sum_{\alpha \in \mathcal{A}} e_{s,\alpha} \right)$$

$$+ \left( \left[ \sum_{s' \in \mathcal{S}} \rho(s') \sum_{\alpha' \in \mathcal{A}} \pi_\lambda^{\mathbb{A}}(\alpha' \mid s') e_{s',\alpha'} \right]^\top \frac{\partial \mathcal{T}(\pi_\lambda^{\mathbb{A}})}{\partial \lambda} \sum_{\alpha \in \mathcal{A}} e_{s,\alpha} \right) \qquad (E.61)$$

$$= \left( \left[ \sum_{s' \in \mathcal{S}} \rho(s') \sum_{\alpha' \in \mathcal{A}} \text{pert}(\alpha_i' \mid s') \cdot \pi_{-i}(\alpha_{-i}' \mid s') e_{s',\alpha'} \right]^\top \mathcal{T}(\pi_\lambda^{\mathbb{A}}) \sum_{\alpha \in \mathcal{A}} e_{s,\alpha} \right)$$

$$+ \left( \left[ \sum_{s' \in \mathcal{S}} \rho(s') \sum_{\alpha' \in \mathcal{A}} \pi_\lambda^{\mathbb{A}}(\alpha' \mid s') e_{s',\alpha'} \right]^\top \frac{\partial (I - \mathcal{P}(\pi_\lambda^{\mathbb{A}}))^{-1}}{\partial \lambda} \sum_{\alpha \in \mathcal{A}} e_{s,\alpha} \right) \qquad (E.62)$$

$$= \left( \left[ \sum_{s' \in \mathcal{S}} \rho(s') \sum_{\alpha' \in \mathcal{A}} \text{pert}(\alpha_i' \mid s') \cdot \pi_{-i}(\alpha_{-i}' \mid s') e_{s',\alpha'} \right]^\top \mathcal{T}(\pi_\lambda^{\mathbb{A}}) \sum_{\alpha \in \mathcal{A}} e_{s,\alpha} \right)$$

$$+ \left( \left[ \sum_{s' \in \mathcal{S}} \rho(s') \sum_{\alpha' \in \mathcal{A}} \pi_\lambda^{\mathbb{A}}(\alpha' \mid s') e_{s',\alpha'} \right]^\top (\mathcal{T}(\pi_\lambda^{\mathbb{A}}) \frac{\partial \mathcal{P}(\pi_\lambda^{\mathbb{A}})}{\partial \lambda} \mathcal{T}(\pi_\lambda^{\mathbb{A}})) \sum_{\alpha \in \mathcal{A}} e_{s,\alpha} \right) \qquad (E.63)$$

970 Thus for $\lambda = 0$, we get

$$\frac{\partial \tilde{d}_\rho^{\pi_\lambda^{\mathbb{A}}}(s)}{\partial \lambda} \right|_{\lambda=0} = \left( \left[ \sum_{s' \in \mathcal{S}} \rho(s') \sum_{\alpha' \in \mathcal{A}} \text{pert}(\alpha_i' \mid s') \cdot \pi_{-i}(\alpha_{-i}' \mid s') e_{s',\alpha'} \right]^\top \mathcal{T}(\pi) \sum_{\alpha \in \mathcal{A}} e_{s,\alpha} \right)$$

$$+ \left( \left[ \sum_{s' \in \mathcal{S}} \rho(s') \sum_{\alpha' \in \mathcal{A}} \pi(\alpha' \mid s') e_{s',\alpha'} \right]^{\top} (\mathcal{T}(\pi) \frac{\partial \mathcal{P}(\pi_\lambda^{\mathbb{A}})}{\partial \lambda} \Big|_{\lambda=0} \mathcal{T}(\pi)) \sum_{\alpha \in \mathcal{A}} e_{s,\alpha} \right) \qquad (\text{E.64})$$

Notice that $\left[ \frac{\partial \mathcal{P}(\pi_\lambda^{\mathbb{A}})}{\partial \lambda} \Big|_{\lambda=0} \right]_{(s^\circ, \alpha^\circ) \to (s^\star, \alpha^\star)} = \text{pert}(\alpha_i^\star \mid s^\star) \cdot \pi_{-i}(\alpha_{-i}^\star \mid s^\star) P(s^\star \mid s^\circ, \alpha^\circ).$

To compactify the notation let us call $\text{aux}_A := \left[ \sum_{s' \in \mathcal{S}} \rho(s') \sum_{\alpha' \in \mathcal{A}} \text{pert}(\alpha_i' \mid s') \cdot \pi_{-i}(\alpha_{-i}' \mid s') e_{s',\alpha'} \right]$,
$\text{aux}_B := [\sum_{s' \in \mathcal{S}} \rho(s') \sum_{\alpha' \in \mathcal{A}} \pi(\alpha' \mid s') e_{s',\alpha'}]$ and $\text{aux}_C(s) := \sum_{\alpha \in \mathcal{A}} e_{s,\alpha}$.

Then, we get that:

$$\text{Term}_A = \left| \sum_{s,\alpha} \frac{\partial \tilde{d}_\rho^{\pi_\lambda^{\mathbb{A}}}(s)}{\partial \lambda} \Big|_{\lambda=0} (\pi_\prime - \pi^{'})(\alpha \mid s) Q_i^{\pi_{s'}^{'}}(s, \alpha) \right| \qquad (\text{E.65})$$

$$= \left| \sum_{s,\alpha} \left( \text{aux}_A^{\top} \mathcal{T}(\pi) \text{aux}_C(s) + \text{aux}_B^{\top} (\mathcal{T}(\pi) \frac{\partial \mathcal{P}(\pi_\lambda^{\mathbb{A}})}{\partial \lambda} \Big|_{\lambda=0} \mathcal{T}(\pi)) \text{aux}_C(s) \right) (\pi - \pi^{'})(\alpha \mid s) Q_i^{\pi^{'}}(s, \alpha) \right| \qquad (\text{E.66})$$

$$= \left| \left( \text{aux}_A^{\top} \mathcal{T}(\pi) + \text{aux}_B^{\top} (\mathcal{T}(\pi) \frac{\partial \mathcal{P}(\pi_\lambda^{\mathbb{A}})}{\partial \lambda} \Big|_{\lambda=0} \mathcal{T}(\pi)) \right) \underbrace{\sum_{s,\alpha} (\pi - \pi^{'})(\alpha \mid s) Q_i^{\pi^{'}}(s, \alpha) \text{aux}_C(s)}_{\text{aux}_D} \right| \qquad (\text{E.67})$$

$$\leq \|\text{aux}_A\|_1 \|\mathcal{T}(\pi) \text{aux}_D\|_\infty + \|\text{aux}_B\|_1 \|(\mathcal{T}(\pi) \frac{\partial \mathcal{P}(\pi_\lambda^{\mathbb{A}})}{\partial \lambda} \Big|_{\lambda=0} \mathcal{T}(\pi)) \text{aux}_D\|_\infty \qquad (\text{E.68})$$

It is easy to see that $\|\text{aux}_A\|_1 \leq \sqrt{|\mathcal{A}_i|}$, $\|\text{aux}_B\|_1 = 1$. Indeed,

$$\|\text{aux}_A\|_1 = \sum_{s' \in \mathcal{S}} \rho(s') \sum_{\alpha' \in \mathcal{A}} |\text{pert}(\alpha_i' \mid s')| \cdot \pi_{-i}(\alpha_{-i}' \mid s') = \sum_{s' \in \mathcal{S}} \rho(s') \sum_{\alpha_i' \in \mathcal{A}_i} |\text{pert}(\alpha_i' \mid s')|$$

$$= \sum_{s' \in \mathcal{S}} \rho(s') \|\text{pert}_{i|s'}\|_1 \leq \sum_{s' \in \mathcal{S}} \rho(s') \sqrt{|\mathcal{A}_i|} \|\text{pert}_{i|s'}\|_2 \leq \sqrt{|\mathcal{A}_i|} \qquad (\text{E.69})$$

$$\|\text{aux}_B\|_1 = \sum_{s' \in \mathcal{S}} \rho(s') \sum_{\alpha' \in \mathcal{A}} \pi(\alpha' \mid s') = 1 \qquad (\text{E.70})$$

Additionally by Conversion Lemma in Matrix form (See Lemma E.2), we have that:

$$\|\mathcal{T}(\pi) x\|_\infty = \max_{s,\alpha} |e_{s,\alpha}^{\top} \mathcal{T}(\pi) x| = \max_{s,\alpha} \left| \mathbb{E}_{\tau \sim \text{MDP}} \left[ \sum_{t=0}^{T(\tau)} x(s_t, \alpha_t) \mid \alpha_0 = \alpha, s_0 = s \right] \right| \leq \frac{1}{\zeta} \|x\|_\infty \qquad (\text{E.71})$$

Similarly, for the matrix $\frac{\partial \mathcal{P}(\pi_\lambda^{\mathbb{A}})}{\partial \lambda} \Big|_{\lambda=0}$, we have that

$$\|\frac{\partial \mathcal{P}(\pi_\lambda^{\mathbb{A}})}{\partial \lambda} \Big|_{\lambda=0} x\|_\infty = \max_{s,\alpha} |e_{s,\alpha}^{\top} \frac{\partial \mathcal{P}(\pi_\lambda^{\mathbb{A}})}{\partial \lambda} \Big|_{\lambda=0} x| = \max_{s,\alpha} \left| \sum_{s',\alpha'} \text{pert}(\alpha_i' \mid s') \cdot \pi_{-i}(\alpha_{-i}' \mid s') P(s' \mid s, \alpha) x_{s',\alpha'} \right|$$

$$\leq \sum_{s',\alpha'} |\text{pert}(\alpha_i' \mid s')| \cdot \pi_{-i}(\alpha_{-i}' \mid s') P(s' \mid s, \alpha) \leq \sqrt{|\mathcal{A}_i|} \|\text{pert}_{i|s'}\|_2 \|x\|_\infty \leq \sqrt{|\mathcal{A}_i|} \|x\|_\infty \qquad (\text{E.72})$$

since $\|\text{pert}\|_2 = 1$. Then, using (E.72) and (E.71) in (E.68) we get that :

$$\text{Term}_A \leq \frac{\sqrt{|\mathcal{A}_i|}}{\zeta} \|\text{aux}_D\|_\infty + \frac{\sqrt{|\mathcal{A}_i|}}{\zeta^2} \|\text{aux}_D\|_\infty \qquad (\text{E.73})$$

$$\leq \frac{\sqrt{|\mathcal{A}_i|}}{\zeta} (1 + \frac{1}{\zeta}) \left\| \sum_{s,\alpha} (\pi - \pi^{'})(\alpha \mid s) Q_i^{\pi^{'}}(s, \alpha) \text{aux}_C(s) \right\|_\infty \qquad (\text{E.74})$$

$$\leq \frac{\sqrt{|\mathcal{A}_i|}}{\zeta^2} (1 + \frac{1}{\zeta}) \max_s \left| \sum_\alpha (\pi - \pi^{'})(\alpha \mid s) \right| \|\text{aux}_C(s)\|_\infty \qquad (\text{E.75})$$

$$\leq \frac{\sqrt{|\mathcal{A}_i|}}{\zeta^2}(1 + \frac{1}{\zeta})\sum_{j=1}^{N}\sqrt{|\mathcal{A}_i|}\|\pi_j - \pi'_j\| \leq \frac{\sqrt{|\mathcal{A}_i|}}{\zeta^3}\sum_{j=1}^{N}\sqrt{|\mathcal{A}_i|}\|\pi_j - \pi'_j\| \tag{E.76}$$

where we used above the fact that $Q$ function is bounded by $1/\zeta$, $\|\text{pert}\| = 1$ and the proposition E.1 to bound the difference of the policy profiles.

For the $\text{Term}_B$, we have that:

$$\text{Term}_B = \left|\sum_{s,\alpha} d_\rho^\pi(s)\frac{\partial(\pi_\lambda^{\mathbb{A}} - \pi_\lambda^{\mathbb{B}})(\alpha \mid s)}{\partial\lambda}\Big|_{\lambda=0} Q_i^{\pi'}(s,\alpha)\right| \tag{E.77}$$

$$= \left|\sum_{s,\alpha} d_\rho^\pi(s)\text{pert}(\alpha_i \mid s)(\pi_{-i} - \pi'_{-i})(\alpha \mid s)Q_i^{\pi'}(s,\alpha)\right| \tag{E.78}$$

$$\leq \frac{1}{\zeta}\left|\sum_{s} d_\rho^\pi(s)\sum_{\alpha_i}\text{pert}(\alpha_i \mid s)\sum_{\alpha_{-i}}(\pi_{-i} - \pi'_{-i})(\alpha \mid s)\right| \tag{E.79}$$

$$\leq \frac{1}{\zeta}\sum_{s}\left|d_\rho^\pi(s)\max_{s}\sum_{\alpha_i}|\text{pert}(\alpha_i \mid s)|\sum_{\alpha_{-i}}(\pi_{-i} - \pi'_{-i})(\alpha \mid s)\right| \tag{E.80}$$

$$\leq \frac{1}{\zeta}\max_{s}\|\text{pert}_{i|s}\|_1\sum_{s} d_\rho^\pi(s)\sum_{\alpha_{-i}}|(\pi_{-i} - \pi'_{-i})(\alpha \mid s)| \tag{E.81}$$

$$\leq \frac{\sqrt{|\mathcal{A}_i|}}{\zeta}\max_{s}\|\text{pert}_{i|s}\|_2(\sum_{s} d_\rho^\pi(s)\sum_{\alpha_{-i}}|(\pi_{-i} - \pi'_{-i})(\alpha \mid s)|) \tag{E.82}$$

$$\leq \frac{\sqrt{|\mathcal{A}_i|}}{\zeta}\sum_{j\in\mathcal{N}\setminus\{i\}}\sqrt{|\mathcal{A}_i|}\|\pi_j - \pi'_j\| \leq \frac{\sqrt{|\mathcal{A}_i|}}{\zeta}\sum_{j=1}^{N}\sqrt{|\mathcal{A}_i|}\|\pi_j - \pi'_j\| \tag{E.83}$$

where we used again the fact that $Q$ function is bounded by $1/\zeta$ and the proposition E.1 to bound the difference of the policy profiles.

For the $\text{Term}_C$, we get that:

$$\text{Term}_C = \left|\sum_{s,\alpha} d_\rho^\pi(s)(\pi - \pi')(\alpha \mid s)\frac{\partial Q_i^{\pi_\lambda^{\mathbb{B}}}(s,\alpha)}{\partial\lambda}\Big|_{\lambda=0}\right| \tag{E.84}$$

$$\leq \max_{s,\alpha}\left|\frac{\partial Q_i^{\pi_\lambda^{\mathbb{B}}}(s,\alpha)}{\partial\lambda}\Big|_{\lambda=0}\right|\left|\sum_{s,\alpha} d_\rho^\pi(s)\right|(\pi - \pi')(\alpha \mid s)| \tag{E.85}$$

$$\leq \max_{s,\alpha}\left|\frac{\partial Q_i^{\pi_\lambda^{\mathbb{B}}}(s,\alpha)}{\partial\lambda}\Big|_{\lambda=0}\right|\left|\sum_{j=1}^{N}\sqrt{|\mathcal{A}_i|}\|\pi_j - \pi'_j\|\right| \tag{E.86}$$

$$\leq \max_{s,\alpha}\left|e_{s,\alpha}^\top\frac{\partial\mathcal{T}(\pi_\lambda^{\mathbb{B}})}{\partial\lambda}\Big|_{\lambda=0}r_i\right|\sum_{j=1}^{N}\sqrt{|\mathcal{A}_i|}\|\pi_j - \pi'_j\| \tag{E.87}$$

$$\leq \max_{s,\alpha}\left|e_{s,\alpha}^\top\frac{\partial(I - \mathcal{P}(\pi_\lambda^{\mathbb{A}}))^{-1}}{\partial\lambda}\Big|_{\lambda=0}r_i\right|\sum_{j=1}^{N}\sqrt{|\mathcal{A}_i|}\|\pi_j - \pi'_j\| \tag{E.88}$$

$$\leq \max_{s,\alpha}\left|e_{s,\alpha}^\top(\mathcal{T}(\pi)\frac{\partial\mathcal{P}(\pi_\lambda^{\mathbb{A}})}{\partial\lambda}\Big|_{\lambda=0}\mathcal{T}(\pi))r_i\right|\sum_{j=1}^{N}\sqrt{|\mathcal{A}_i|}\|\pi_j - \pi'_j\| \tag{E.89}$$

$$\leq \frac{\sqrt{|\mathcal{A}_i|}}{\zeta^2}\sum_{j=1}^{N}\sqrt{|\mathcal{A}_i|}\|\pi_j - \pi'_j\| \tag{E.90}$$

using again (E.72) and (E.71) and proposition E.1. Thus, we are ready now to bound the gradient per player:

$$\left|\frac{\partial(V_{i,\rho}(\pi_\lambda^{\mathbb{A}}) - V_{i,\rho}(\pi_\lambda^{\mathbb{B}}))}{\partial\lambda}\Big|_{\lambda=0}\right| \leq \text{Term}_A + Z_\rho^{\pi^{\mathbb{A}}}(\text{Term}_B + \text{Term}_C) \leq \frac{3\sqrt{|\mathcal{A}_i|}}{\zeta^3}\sum_{j=1}^{N}\sqrt{|\mathcal{A}_i|}\|\pi_j - \pi'_j\|$$

987 where we recall that $Z_\rho^{\pi^A} \le \frac{1}{\zeta}$ Since we prove it for an arbitrary perturbation vector pert for the
988 directional derivative, for the independent player's policy gradient it holds also that:

$$\|v_i(\pi) - v_i(\pi')\| = \|\nabla_i(V_{i,\rho}(\pi) - \nabla_i(V_{i,\rho}(\pi')\| \le \frac{3\sqrt{|\mathcal{A}_i|}}{\zeta^3} \sum_{j=1}^{N} \sqrt{|\mathcal{A}_i|} \|\pi_j - \pi'_j\| \quad \forall i \in \mathcal{N}$$

989 Finally for the concatenated gradient operator we get:

$$\|v(\pi) - v(\pi')\| = \sqrt{\sum_{i\in\mathcal{N}} \|v_i(\pi) - v_i(\pi')\|^2} = \sqrt{\sum_{i\in\mathcal{N}} \|\nabla_i(V_{i,\rho}(\pi) - \nabla_i(V_{i,\rho}(\pi')\|^2} \tag{E.91}$$

$$\le \sqrt{\sum_{i\in\mathcal{N}} \frac{9|\mathcal{A}_i|}{\zeta^6} (\sum_{j\in\mathcal{N}} \sqrt{|\mathcal{A}_i|} \|\pi_j - \pi'_j\|)^2} \le \sqrt{\sum_{i\in\mathcal{N}} \frac{9|\mathcal{A}_i|}{\zeta^6} \sum_{j\in\mathcal{N}} |\mathcal{A}_i| \sum_{j\in\mathcal{N}} \|\pi_j - \pi'_j\|^2} \tag{E.92}$$

$$\le \frac{3}{\zeta^3} \sqrt{(\sum_{i\in\mathcal{N}} |\mathcal{A}_i|)^2 \|\pi - \pi'\|^2} \le \frac{3|\mathcal{A}|}{\zeta^3} \|\pi - \pi'\| \tag{E.93}$$

990 ∎

# F   Statistics of Reinforce

Let's first recall our notation: We will write $\nabla_i$ to denote the gradient of the quantity in question with respect to $\pi_i$, i.e., when $\pi_{-i}$ is kept fixed and only $\pi_i$ is varied. For concision, we will write $v_i(\pi) = \nabla_i V_{i,\rho}(\pi)$ for the individual gradient of player $i$'s value function, and $v(\pi) = (v_i(\pi))_{i\in\mathcal{N}}$ for the ensemble thereof. Below we present two fundamental properties of Reinforce Policy Gradient estimator that we will utilize later in the our analysis.
- Reinforce is an unbiased estimator of $v(\pi)$.
- Reinforce's variance is bounded by $\mathcal{O}(1/\min_{s\in\mathcal{S},\alpha_i\in\mathcal{A}_i} \pi_i(\alpha_i|s))$ for each $i \in \mathcal{N}$.

993 **Lemma 4.** *Suppose that each agents $i \in \mathcal{N}$ follows a stationary policy $\pi_i \in \Pi_i$. Then, letting*
994 $\kappa_i = \min_{s\in\mathcal{S},\alpha_i\in\mathcal{A}_i} \pi_i(\alpha_i|s)$ *for each $i \in \mathcal{N}$, we have*

a)   $\mathbb{E}_{\tau\sim\text{MDP}}[\text{Reinforce}(\pi)] = v(\pi).$ (12a)

b)   $\mathbb{E}_{\tau\sim\text{MDP}}\Big[\|\text{Reinforce}_i(\pi) - v_i(\pi)\|^2\Big] \le \frac{24|\mathcal{A}_i|}{\kappa_i\zeta^4}.$ (12b)

995 *Proof.* In order to prove $\mathbb{E}_{\tau\sim\text{MDP}}[\text{Reinforce}(\pi)] = v(\pi)$, it is equivalent to prove that

$$\mathbb{E}_{\tau\sim\text{MDP}}[\text{Reinforce}_i(\pi)] = v_i(\pi) \text{ for each } i \in \mathcal{N}.$$

996 Without loss of generality let's assume that $\text{MDP} \equiv \text{MDP}(\pi \mid \rho)$ for some initial state distribution $\rho$.
997 Additionally, we denote $\mathbb{P}^\pi(\tau)$ the induced probability of a random trajectory $\tau = (s_t, \alpha_t, r_t)_{t\le T(\tau)}$.

$$\mathbb{E}_{\tau\sim\text{MDP}}[\hat{v}_i] = \mathbb{E}_{\tau\sim\text{MDP}}[R_i(\tau) \cdot \Lambda_i(\tau)] = \sum_{\tau\in\mathcal{T}} \mathbb{P}^\pi(\tau) R_i(\tau) \cdot \Lambda_i(\tau) \tag{F.1}$$

$$= \sum_{\tau\in\mathcal{T}} \mathbb{P}^\pi(\tau) R_i(\tau) \cdot [\sum_{t=0}^{T(\tau)} \nabla_i(\log \pi_i(a_{i,t}|s_t))] \tag{F.2}$$

$$= \sum_{\tau\in\mathcal{T}} \mathbb{P}^\pi(\tau) R_i(\tau) \cdot \nabla_i \Big[\sum_{t=0}^{T(\tau)} \log \pi_i(a_{i,t}|s_t)\Big] \tag{F.3}$$

$$= \sum_{\tau\in\mathcal{T}} \mathbb{P}^\pi(\tau) R_i(\tau) \nabla_i \sum_{t=0}^{T(\tau)} \log \pi_i(a_{i,t}|s_t)$$

$$+ \sum_{\tau\in\mathcal{T}} \mathbb{P}^\pi(\tau) R_i(\tau) \Big( \nabla_i \sum_{j\in\mathcal{N}\setminus\{i\}} \sum_{t=0}^{T(\tau)} \log \pi_j(\alpha_{j,t}|s_t) + \nabla_i \sum_{t=0}^{T(\tau)} \log \mathbb{P}(s_t \mid s_{t-1}, a_{t-1}) \Big) \tag{F.4}$$

$$+ \sum_{\tau\in\mathcal{T}} \mathbb{P}^\pi(\tau) R_i(\tau) \nabla_i \log \rho(s_0)$$

$$= \sum_{\tau \in \mathcal{T}} \mathbb{P}^\pi(\tau) R_i(\tau) \nabla_i(\log \mathbb{P}^\pi(\tau)) = \sum_{\tau \in \mathcal{T}} (\nabla_i \mathbb{P}^\pi(\tau)) R_i(\tau) = \nabla_i(\sum_{\tau \in \mathcal{T}} \mathbb{P}^\pi(\tau) R_i(\tau)) \tag{F.5}$$

$$= \nabla_i V_{i,\rho}(\pi) \tag{F.6}$$

where in the second to last inequality we used the definition for the derivative of the logarithm. We also note here that

$$\mathbb{E}_{\tau \sim \text{MDP}}[\hat{v}_i] = \mathbb{E}_{\tau \sim \text{MDP}}[R_i(\tau) \nabla_i(\log \mathbb{P}^\pi(\tau))] \tag{F.7}$$

For the variance of REINFORCE estimator we have that

$$
\begin{aligned}
\mathbb{E}_{\tau \sim \text{MDP}}\Big[\|\text{REINFORCE}_i(\pi) - v_i(\pi)\|^2\Big] = {} & \mathbb{E}_{\tau \sim \text{MDP}}\Big[\|\text{REINFORCE}_i(\pi)\|^2\Big] \\
& - 2\,\mathbb{E}_{\tau \sim \text{MDP}}[\langle \text{REINFORCE}_i(\pi), v_i(\pi)\rangle] \\
& + \mathbb{E}_{\tau \sim \text{MDP}}\Big[\|v_i(\pi)\|^2\Big]
\end{aligned}
$$

or equivalently $\mathbb{E}_{\tau \sim \text{MDP}}\Big[\|\text{REINFORCE}_i(\pi) - v_i(\pi)\|^2\Big] = \mathbb{E}_{\tau \sim \text{MDP}}\Big[\|\text{REINFORCE}_i(\pi)\|^2\Big] - \mathbb{E}_{\tau \sim \text{MDP}}\Big[\|v_i(\pi)\|^2\Big]$.
Therefore, we have that

$$\mathbb{E}_{\tau \sim \text{MDP}}\Big[\|\text{REINFORCE}_i(\pi) - v_i(\pi)\|^2\Big] \leq \mathbb{E}_{\tau \sim \text{MDP}}\Big[\|\text{REINFORCE}_i(\pi)\|^2\Big] = \mathbb{E}[\|\hat{v}_i\|^2] \tag{F.8}$$

$$\mathbb{E}[\|\hat{v}_i\|^2] = \mathbb{E}_{\tau \sim \text{MDP}}[\|R_i(\tau)\Lambda_i(\tau)\|^2] \leq \mathbb{E}_{\tau \sim \text{MDP}}[\|R_i(\tau)\|^2 \|\Lambda_i(\tau)\|^2] \tag{F.9}$$

$$\leq \mathbb{E}_{\tau \sim \text{MDP}}[(T(\tau) + 1)^2 \|\sum_{t=0}^{T(\tau)} \nabla_i \log \pi_i(a_{i,t}, s_t)\|^2] \tag{F.10}$$

$$\leq \mathbb{E}_{\tau \sim \text{MDP}}[(T(\tau) + 1)^3 \sum_{t=0}^{\infty} \sum_{s,a \in \mathcal{S} \times \mathcal{A}_i} \mathbb{1}\{t \leq T\}\mathbb{1}\{s_t = s, a_{i,t} = a\}\|\nabla_i \log \pi_i(a, s)\|^2] \tag{F.11}$$

$$= \sum_{t=0}^{\infty} \sum_{s,a \in \mathcal{S} \times \mathcal{A}_i} \mathbb{E}_{\tau \sim \text{MDP}}[(T(\tau) + 1)^3 \mathbb{1}\{t \leq T\}\mathbb{1}\{s_t = s, a_{i,t} = a\}\frac{1}{(\pi_i(a,s))^2}] \tag{F.12}$$

$$\leq \sum_{t=0}^{\infty} \sum_{s,a \in \mathcal{S} \times \mathcal{A}_i} \frac{1}{(\pi_i(a,s))^2} \mathbb{E}_{\tau \sim \text{MDP}}[(T(\tau) + 1)^3 \mathbb{1}\{t \leq T\}\mathbb{1}\{s_t = s, a_{i,t} = a\}] \tag{F.13}$$

$$\leq \sum_{t=0}^{\infty} \sum_{s,a \in \mathcal{S} \times \mathcal{A}_i} \frac{1}{\pi_i(a,s)} \mathbb{E}_{\tau \sim \text{MDP}}[(T(\tau) + 1)^3 \mathbb{1}\{t \leq T\}\mathbb{1}\{s_t = s\}] \tag{F.14}$$

$$\leq \sum_{t=0}^{\infty} \sum_{s,a \in \mathcal{S} \times \mathcal{A}_i} \frac{1}{\kappa_i}\{(T(\tau) + 1)^3 \mathbb{1}\{t \leq T\}\mathbb{1}\{s_t = s\}\} \tag{F.15}$$

$$= \sum_{t=0}^{\infty} \sum_{s \in \mathcal{S}} \frac{|A_i|}{\kappa_i} \mathbb{E}_{\tau \sim \text{MDP}}[(T(\tau) + 1)^3 \mathbb{1}\{t \leq T\}\mathbb{1}\{s_t = s\}] \tag{F.16}$$

$$= \frac{|A_i|}{\kappa_i} \mathbb{E}_{\tau \sim \text{MDP}}[(T(\tau) + 1)^3 \sum_{t=0}^{T} \mathbb{1}\{t \leq T\}] \tag{F.17}$$

$$\leq \frac{|A_i|}{\kappa_i} \mathbb{E}_{\tau \sim \text{MDP}}[(T(\tau) + 1)^4] \tag{F.18}$$

$$\leq \frac{|A_i|}{\kappa_i} \sum_{t=0}^{\infty} (1 - \zeta)^t \zeta(t+1)^4 \leq \frac{24}{\zeta^4} \frac{|A_i|}{\kappa_i} \tag{F.19}$$

we note that to go from the first to the second inequality we used the boundeness by one of the rewards, while from the second to the third using Jensen's inequality. ∎

 # G Solution concepts

> In this part, we will establish three important facts that certifies the leitmotif of our focus to variational optima. More precisely,
> - In Lemma 2, we prove the crucial property of Gradient Dominance for the multi-agent random stopping setting.
> - In Lemma 3, we establish that any stationary point corresponds to Nash Equilibria.
> - In Proposition 1, we prove the "drift" inequalities for all the different types of stationary points.
> Proposition 1 will be crucial to prove the corresponding rate of convergence at the following sections of the supplement

 **Lemma 2.** [Gradient dominance property] *For any policy profile $\pi = (\pi_i)_{i \in \mathcal{N}} \in \Pi$, we have that*

$$V_{i,\rho}(\pi_i'; \pi_{-i}) - V_{i,\rho}(\pi_i; \pi_{-i}) \le \mathcal{C}_{\mathcal{G}} \max_{\bar{\pi}_i \in \Pi_i} \langle \nabla_i V_{i,\rho}(\pi), \bar{\pi}_i - \pi_i \rangle \tag{GDP}$$

 *for any unilateral deviation $\pi_i' \in \Pi_i$ of each player $i \in \mathcal{N}$.*

 *Proof.* We start by rewriting the LHS of the demanded expression using Performance Lemma E.6
 and Conversion Lemma 1 for $\pi^{\mathbb{A}} = (\pi_i'; \pi_{-i})$ and $\pi^{\mathbb{B}} = (\pi_i; \pi_{-i})$:

$$V_{i,\rho}(\pi^{\mathbb{A}}) - V_{i,\rho}(\pi^{\mathbb{B}}) = \sum_{s \in \mathcal{S}} \tilde{d}_\rho^{\pi^{\mathbb{A}}}(s) \, \mathbb{E}_{\alpha \sim \pi^{\mathbb{A}}(\cdot|s)} \left[ A_i^{\pi_i^{\mathbb{B}}}(s, \alpha) \right] \tag{G.1}$$

$$= \sum_{s \in \mathcal{S}} \tilde{d}_\rho^{\pi^{\mathbb{A}}}(s) \sum_{a_i \in \mathcal{A}_i} \pi_i'(a_i|s) \sum_{a_{-i} \in \mathcal{A}_{-i}} \pi_{-i}(a_{-i}|s) A_i^{\pi^{\mathbb{B}}}(s, \alpha) \tag{G.2}$$

$$= \sum_{s \in \mathcal{S}} \tilde{d}_\rho^{\pi^{\mathbb{A}}}(s) \sum_{a_i \in \mathcal{A}_i} \pi_i'(a_i|s) \overline{A}_i^{\pi^{\mathbb{B}}}(s, a_i) \tag{G.3}$$

$$\le \sum_{s \in \mathcal{S}} \tilde{d}_\rho^{\pi^{\mathbb{A}}}(s) \sum_{a_i \in \mathcal{A}_i} \pi_i'(a_i|s) \max_{a_i \in \mathcal{A}_i} \overline{A}_i^{\pi^{\mathbb{B}}}(s, a_i) \tag{G.4}$$

$$V_{i,\rho}(\pi^{\mathbb{A}}) - V_{i,\rho}(\pi^{\mathbb{B}}) \le \max_{\tilde{\pi}_i \in \Delta(\mathcal{A})^{\mathcal{S}}} \sum_{s \in \mathcal{S}} \tilde{d}_\rho^{\pi^{\mathbb{A}}}(s) \sum_{a_i \in \mathcal{A}_i} \tilde{\pi}_i(a_i|s) \overline{A}_i^{\pi^{\mathbb{B}}}(s, a_i) \tag{G.5}$$

$$\le \max_{\tilde{\pi}_i \in \Delta(\mathcal{A})^{\mathcal{S}}} \sum_{s \in \mathcal{S}} \tilde{d}_\rho^{\pi^{\mathbb{A}}}(s) \sum_{a_i \in \mathcal{A}_i} (\tilde{\pi}_i(a_i|s) - \pi_i(a_i|s)) \overline{A}_i^{\pi^{\mathbb{B}}}(s, a_i) \tag{G.6}$$

$$\le \max_{\tilde{\pi}_i \in \Delta(\mathcal{A})^{\mathcal{S}}} \sum_{s \in \mathcal{S}} \frac{\tilde{d}_\rho^{\pi^{\mathbb{A}}}(s)}{\tilde{d}_\rho^{\pi^{\mathbb{B}}}(s)} \tilde{d}_\rho^{\pi^{\mathbb{B}}}(s) \sum_{a_i \in \mathcal{A}_i} (\tilde{\pi}_i(a_i|s) - \pi_i(a_i|s)) \overline{A}_i^{\pi^{\mathbb{B}}}(s, a_i) \tag{G.7}$$

$$\le \left\| \frac{\tilde{d}_\rho^{\pi^{\mathbb{A}}}(s)}{\tilde{d}_\rho^{\pi^{\mathbb{B}}}(s)} \right\|_\infty \max_{\tilde{\pi}_i \in \Delta(\mathcal{A})^{\mathcal{S}}} \sum_{s \in \mathcal{S}} \sum_{a_i \in \mathcal{A}_i} \tilde{d}_\rho^{\pi^{\mathbb{B}}}(s)(\tilde{\pi}_i(a_i|s) - \pi_i(a_i|s)) \overline{Q}_i^{\pi^{\mathbb{B}}}(s, a_i) \tag{G.8}$$

$$\le \left\| \frac{\tilde{d}_\rho^{\pi^{\mathbb{A}}}(s)}{\tilde{d}_\rho^{\pi^{\mathbb{B}}}(s)} \right\|_\infty \max_{\tilde{\pi}_i \in \Delta(\mathcal{A})^{\mathcal{S}}} \sum_{s \in \mathcal{S}, a_i \in \mathcal{A}_i} (\tilde{\pi}_i(a_i|s) - \pi_i(a_i|s)) \tilde{d}_\rho^{\pi^{\mathbb{B}}}(s) \overline{Q}_i^{\pi^{\mathbb{B}}}(s, a_i) \tag{G.9}$$

$$\le \left\| \frac{\tilde{d}_\rho^{\pi^{\mathbb{A}}}(s)}{\tilde{d}_\rho^{\pi^{\mathbb{B}}}(s)} \right\|_\infty \max_{\tilde{\pi}_i \in \Delta(\mathcal{A})^{\mathcal{S}}} \sum_{s \in \mathcal{S}, a_i \in \mathcal{A}_i} (\tilde{\pi}_i(a_i|s) - \pi_i(a_i|s)) \frac{\partial V_{i,\rho}(\pi)}{\partial \pi_i(\alpha_i \mid s)} \tag{G.10}$$

$$V_{i,\rho}(\pi_i'; \pi_{-i}) - V_{i,\rho}(\pi_i; \pi_{-i}) \le \mathcal{C}_{\mathcal{G}} \max_{\bar{\pi}_i \in \Pi_i} \langle \nabla_i V_{i,\rho}(\pi), \bar{\pi}_i - \pi_i \rangle \tag{G.11}$$

 Notice that we have assumed that $\tilde{d}_\rho^{\pi^{\mathbb{B}}} > 0$. If this wasn't the case we could take a trivial bound of $\infty$.

 ∎

 **Lemma 3.** [First-order stationary policies are Nash] *A profile $\pi^* = (\pi_i^*)_{i \in \mathcal{N}} \in \Pi$ is a Nash policy
 profile if and only if it satisfies the first-order stationary condition*

$$\langle v(\pi^*), \pi - \pi^* \rangle \le 0 \quad \text{for all } \pi \in \Pi. \tag{FOS}$$

*Proof.* Let's apply the definition of first-order stationary point for the pair of policy profiles $\{\pi^*, \pi\}$:
$\pi^* = (\pi_i^*, \pi_{-i}^*)$ and $\pi = (\pi_i, \pi_{-i})$:

$$\langle v(\pi^*), \pi^* - \pi \rangle \geq 0 \qquad\qquad \Leftrightarrow \qquad\qquad \text{(G.12)}$$

$$\langle v(\pi^*), (\pi_i^*, \pi_{-i}^*) - (\pi_i, \pi_{-i}^*) \rangle \geq 0 \qquad\qquad \Leftrightarrow \qquad\qquad \text{(G.13)}$$

$$\langle v(\pi^*), (\pi_i^* - \pi_i, 0) \rangle \geq 0 \qquad\qquad \Leftrightarrow \qquad\qquad \text{(G.14)}$$

$$\langle v_i(\pi^*), \pi_i^* - \pi_i \rangle \geq 0 \qquad\qquad \Leftrightarrow \qquad\qquad \text{(G.15)}$$

$$\langle \nabla_i V_{i,\rho}(\pi^*), \pi_i^* - \pi_i \rangle \geq 0 \qquad\qquad \Leftrightarrow \qquad\qquad \text{(G.16)}$$

$$\min_{\bar{\pi}_i \in \Pi_i} \langle \nabla_i V_{i,\rho}(\pi^*), \pi_i^* - \bar{\pi}_i \rangle \geq 0 \qquad\qquad \Leftrightarrow \qquad\qquad \text{(G.17)}$$

$$\max_{\bar{\pi}_i \in \Pi_i} \langle \nabla_i V_{i,\rho}(\pi^*), \pi_i - \bar{\pi}_i^* \rangle \leq 0 \qquad\qquad \Leftrightarrow \qquad\qquad \text{(G.18)}$$

$$\text{(G.19)}$$

By Gradient Dominance Property and Lemma 2, we have that

$$V_{i,\rho}(\pi_i; \pi_{-i}^*) - V_{i,\rho}(\pi_i^*; \pi_{-i}^*) \leq \mathcal{C}_{\mathcal{G}} \max_{\bar{\pi}_i \in \Pi_i} \langle \nabla_i V_{i,\rho}(\pi^*), \bar{\pi}_i - \pi_i^* \rangle \leq 0 \Rightarrow \qquad \text{(G.20)}$$

$$V_{i,\rho}(\pi_i; \pi_{-i}^*) \leq V_{i,\rho}(\pi_i^*; \pi_{-i}^*) \quad \forall \pi_i \in \Pi_i. \qquad \text{(G.21)}$$

∎

With all this in place, we are finally in a position to prove the characterization of second-order stationary and strict Nash policies that of Proposition 1. For ease of reference, we restate the relevant claims below.

**Proposition 1.** *Let $\pi^* = (\pi_i^*)_{i \in \mathcal{N}} \in \Pi$ be a Nash policy. Then:*

a) *If $\pi^*$ is second-order stationary, there exists some $\mu > 0$ such that*

$$\langle v(\pi), \pi - \pi^* \rangle \leq -\mu \|\pi - \pi^*\|^2 \qquad \text{for all } \pi \text{ sufficiently close to } \pi^*. \qquad \text{(3a)}$$

b) *If $\pi^*$ is strict, there exists some $\mu > 0$ such that*

$$\langle v(\pi), \pi - \pi^* \rangle \leq -\mu \|\pi - \pi^*\| \qquad \text{for all } \pi \text{ sufficiently close to } \pi^*. \qquad \text{(3b)}$$

*Proof.* We begin with the characterization of second-order stationary policies. To that end, let $d = |\mathcal{S}| \sum_i |\mathcal{A}_i|$ denote the ambient dimension of $\prod_i \left(\mathbb{R}^{\mathcal{A}_i}\right)^{\mathcal{S}}$ and consider the mapping $\varphi : \mathbb{R}^{d \times d} \to \mathbb{R}$ mapping $H \mapsto \max\{z^\top H z : z \in \mathrm{TC}(\pi^*), \|z\| = 1\}$. Clearly, $\varphi$ is convex as the pointwise maximum of a set of linear – and hence convex – functions. This in turn implies the continuity of $\varphi$ as every convex function is continuous on the interior of its effective domain. Since $\pi^*$ satisfies (SOS) by assumption, we have $\varphi(\mathrm{Jac}_v(\pi^*)) < 0$, so, by continuity and the convexity of $\Pi$, there exists some $\mu > 0$ and a convex neighborhood $\mathcal{U}$ of $\pi^*$ in $\Pi$ such that $\varphi(\mathrm{Jac}_v(\pi)) \leq -\mu$ for all $\pi \in \mathcal{U}$.

With this in mind, letting $z = \pi - \pi^* \in \mathrm{TC}(\pi^*)$ for some $\pi \in \mathcal{U}$, a straightforward Taylor expansion with integral remainder yields

$$v(\pi) - v(\pi^*) = \int_0^1 \mathrm{Jac}_v(\pi^* + \tau z) z \, d\tau \qquad \text{(G.22)}$$

and hence, setting $\pi_\tau = \pi^* + \tau z$, we get

$$\langle v(\pi) - v(\pi^*), \pi - \pi^* \rangle = \int_0^1 z^\top \mathrm{Jac}_v(\pi_\tau) z \, d\tau$$

$$\leq \|z\|^2 \int_0^1 \varphi(\mathrm{Jac}_v(\pi_\tau)) \, d\tau \leq -\mu \|z\|^2 = -\mu \|\pi - \pi^*\|^2 \qquad \text{(G.23)}$$

However, by (FOS), we have $\langle v(\pi^*), \pi - \pi^* \rangle \leq 0$ which, combined with the above, yields $\langle v(\pi), \pi - \pi^* \rangle \leq -\mu \|\pi - \pi^*\|^2$, as claimed.

1038 For the second part of our lemma, pick some $\pi \neq \pi^*$ and let $z = (\pi - \pi^*)/\|\pi - \pi^*\|$, so $z \in \mathrm{TC}(\pi^*)$
1039 and $\|z\| = 1$. Then, given that (FOS) is satisfied as a strict inequality for all $\pi \neq \pi^*$, we readily get
1040 $\langle v(\pi^*), z \rangle < 0$ for all $z \in \mathrm{TC}(\pi^*)$ with $\|z\| = 1$. Thus, by the joint continuity of the function $\langle v(\pi), z \rangle$
1041 in $\pi$ and $z$, there exists a compact convex neighborhood $\mathcal{K}$ of $\pi^*$ in $\Pi$ such that $\mu := \min\{\langle v(\pi), z \rangle :$
1042 $\pi \in \mathcal{K}, z \in \mathrm{TC}(\pi^*), \|z\| = 1\} < 0$. Thus, letting $z = (\pi - \pi^*)/\|\pi - \pi^*\|$ as above, we conclude that
1043 $\langle v(\pi), \pi - \pi^* \rangle \leq -\mu\|\pi - \pi^*\|$, as claimed. $\blacksquare$