# OpenReview forum: "On the convergence of policy gradient methods to Nash equilibria in general stochastic games"
_NeurIPS.cc/2022/Conference — NeurIPS 2022 Accept_

### Official Review · Reviewer_kenS · 2022-07-01

**Rating:** 8
**Confidence:** 5
**Soundness:** 4 excellent
**Presentation:** 4 excellent
**Contribution:** 4 excellent

**Summary:**

This paper studies the local stability of stationary Nash equilibria under the projected gradient (PG) algorithm in episodic finite (states and actions) stochastic games under various assumptions on the players observations --full, intermediate (stochastic gradient) or low (value based)-- Form the fact that stationary equilibria of finite stochastic games satisfies a version of Polyak-Łojasiewicz condition (a consequence of the semi-algebraicity), stationary Nash equilibria satisfy a gradient dominance property which easily implies that equilibria can be expressed as solutions of a variational inequality (VI). The rewriting of the equilibrium condition as a VI allows to classify the stability of Nash equilibria from stable, to second-order stationary (as in Mertikopoulos, Zhou Math Prog 2019). They then use the variational inequality conditions to establish their main results, namely 1) if an equilibrium is Nash stable, then if one starts closely to it, PG converges to it with high probability 2) when the equilibrium is second-order stable then the convergence time can be estimated (roughly 1/n or 1/n^1/2)...This rate can be improved to finite time convergence for pure stationary equilibria.

**Questions:**

- Can this result be coupled with the one (in zero-sum and potential stochastic games) where best iterate convergence results have been proved? namely, can we deduce from your analysis a convergence a global convergence is some zero-sum and potential SGs --

- Can you give some non trivial examples of finite stochastic games with a stationary equilibrium which is second order stable (or stable) without being strict or the game zero-sum?


**Limitations:**

 - Can you please discuss the classes of games, beyond the class of stochastic games, where your results apply?

- Can your results extend to ergodic stochastic games -where there is no ending, and the learning is obtained during the play-?

**Strengths And Weaknesses:**

Strengths: this is an excellent paper. It shows the local stability of stationary Nash equilibria under the projected gradient under several observational assumptions whenever that Nash equilibrium has some additional stability properties. The fact that it uses only the variational inequality property of the Nash equilibrium implies that the result extend to a wider class of game, beyond the one studied in the paper (for example concave continuous games as studied in Mertikopoulos, Zhou 2019). The main difference with the MZ paper is that MZ studies Online Mirror Descent while this paper studies projected gradient descent.


Some weaknesses: only local convergence results, only on a subclass of stochastic games (episodic stochastic games). Also, It seems to me that the existence of stable and second order stationary equilibria in finite stochastic games is rather hard to obtain (outside strict pure equilibria or zero sum games).

---

> ### Author Response · Authors · 2022-08-02
> **Replies**
>
> Thank you for your very encouraging comments and your positive evaluation. We reply to your main questions point-by-point below, and we have colored all relevant revisions in our paper in $\color{purple}{\textrm{purple}}$.
>
> 1. **On the locality of our convergence results.**
> Indeed, our convergence results are local, but this cannot be avoided. In general, equilibrium policies are not unique, and gradient-based dynamics may also admit non-equilibrium attractors, such as limit cycles and the like. As a result, in the presence of multiple equilibria/attractors, the best one can hope for is a local equilibrium convergence result, conditioned on the basin of attraction of said equilibrium.
>
>     These issues can only be overcome in games with a sufficiently strong global struture – such as potential or min-max games – *but not otherwise*. We have revised the relevant part of our paper to make this clear.
>
>
> 1. **On non-strict second-order stationary policies.**
> Second-order stationary policies are characterized by a negative-definite Jacobian of the individual gradient field $v(\pi)$. Admittedly, because of the difficulty involved in calculating - and manipulating - the value functions of a stochastic game in closed form, we acknowledge that we don't have an explicit numerical example of a stochastic game with a non-strict SOS policy. However, given that negative-definiteness of the Jacobian is a hyperbolicity assumption, we don't see a fundamental obstruction to the existence of such sets. In addition, we should also point out that SOS policies are standard in the case of non-tabular stochastic games, see e.g., the recent paper of Zhang et al. (SICON, 2020) which provides a wide range of examples of non-strict SOS policies in non-tabular problems. Our theory also applies to this setting, but such an extension lies beyond the scope of this work, so we did not undertake it.
>
>
> 1. **On the relation to best-iterate convergence results.**
> The template of "best-iterate" results can be summarized as follows: if an algorithm is run for $T$ iterations, then, at least one of the generated iterates will have near-stationary individual payoff gradients (with the exact distance from stationarity determined by the associated best-iterate convergence rate).
>     A major difficulty in this setting is that, in the stochastic case, it is impossible to determine which of these $T$ iterates is near-stationary, and this because the players only have access to *stochastic* gradients (not their mean values). In our context however, we do not need to: if we consider a game for which a best-iterate result is available (for example, as you suggest, potential stochastic games or the like), this guarantees that at least one iterate will be sufficiently close to a stationary policy. Given this "asymptotic closeness" result, our paper's analysis guarantees that stable policies will capture the policy gradient process with high probability, thus turning the "best-iterate" analysis to a "last-iterate" analysis.
>
>     That said, in the case of stochastic potential games, it would be simpler to analyze the policy gradient method directly as a constrained stochastic gradient algorithm with possibly biased gradients (and a variance that grows to infinity), rather than going through a "best-iterate" type of result. We conjecture that such an analysis is indeed possible - for both the standard policy gradient method and its "lazy" variant - but it would take us too far afield, so we defer it to the future.
>
> 1. **On more general classes of games where our results apply.**
> A highly promising application domain for our techniques would be the class of non-tabular stochastic games where SOS policies have been studied extensively - cf. the SICON paper by Zhang et al (2020) that we cited above. More generally, our results essentially apply to all continuous games with a "gradient dominance" property guaranteeing that stationary policies are also Nash; this extension lies beyond the scope of the current paper, but, again, it is a very fruitful direction for future research.
>
>
> 1. **On extensions to ergodic stochastic games.**
> This is a very interesting - and challenging - research question. A key difference here is that the REINFORCE algorithm is no longer meaningful in the ergodic setting, so such an extension would require new tools and techniques (at least as far as the gradient estimation process is concerned). This would also necessitate a different, non-episodic algorithm structure - however, if this difficulty is lifted, we believe that it should be posible to extend our analysis to such models; we mention this as an important and intriguing question for future research.
>
> Thanks again for your insightful questions and the positive evaluation of our work. We hope and trust that our replies have alleviated any remaining concerns, and we look forward to an open-minded discussion if you have any further questions.

---

> > ### Comment · Reviewer_kenS · 2022-08-06
> > **Thanks**
> >
> > I would like to thank to authors for answering my all questions. A last question: can you please compare your results with the one in Mertikopoulos, Zhou Math Prog 2019, at least at the technical level?  thanks.

---

> > > ### Author Response · Authors · 2022-08-08
> > > **Replies**
> > >
> > > Thanks for the follow-up! Mertikopoulos and Zhou [40] (referenced as [37] in our original submission) introduced the notion of variational stability for continuous *concave* games - that is, games with continuous action sets and individually concave payoff functions. They did not consider second-order stationary points and/or the rate of convergence to such points, so the only point of contact with [40] would be Theorem 1 (asymptotic convergence to stable equilibria).
> > >
> > > Still, even on this point, the analysis of [40] is drastically different for the following reasons:
> > >
> > > 1. The authors of [40] consider a version of Nesterov's (2009) "dual averaging" method which can be well-approximated by a system of ordinary differential equations (ODEs) in continuous time. This ODE allowed the authors of [40] to derive their local convergence result in the context of continuous concave games by means of stochastic approximation arguments in the spirit of Kushner and Yin (1997) and Benaïm (1999). By contrast, because of the projection step, the policy gradient algorithm **cannot be expressed** as the stochastic approximation of an ODE, so the technical analysis of [40] breaks down at the very first step.
> > >
> > > 1. In addition to the above, [40] only considers learning with unbiased stochastic gradients with finite variance (akin to Model 2 of our paper). However, the REINFORCE algorithm (Model 3) **does not adhere to these assumptions**: in particular, the log-trick estimator (11) is either unbiased with infinite variance, or it has finite variance but nonzero bias (if an explicit mixing step is included). These features of REINFORCE led us to the introduction of an additional smooth exploration mechanism which cannot be handled by the analysis of [40], even if the first obstacle mentioned above was somehow overcome.
> > >
> > > Instead, our analysis focuses directly on the discrete-time algorithm (i.e., without going through a continuous-time proxy), and we leverage a series of convergence results for almost supermartingales to control the evolution of the iterates of the process (namely the Robbins-Siegmund theorem). All this makes the analysis and techniques of [40] radically different from our own - and, as we stated above, [40] does not consider second-order stationary points and/or rates of convergence (which, in turn, require different toolboxes altogether).
> > >
> > > Please let us know if you have any further questions - and thanks again for your detailed input and positive evaluation!

---

> > > > ### Comment · Reviewer_kenS · 2022-08-08
> > > > **Comments**
> > > >
> > > > Dear Authors
> > > >
> > > > Thanks for the detailed and precise reply which clarifies for me the main technical differences with Mertikopoulos and Zhou. This confront me in the score I gave and on the paper's quality. I hope the best for your paper.

---

### Official Review · Reviewer_uYSS · 2022-07-11

**Rating:** 6
**Confidence:** 3
**Soundness:** 2 fair
**Presentation:** 3 good
**Contribution:** 2 fair

**Summary:**

This paper studies the convergence of the policy gradient method for a class of MDP problems. The equilibrium point investigated is the second-order stationary point, which is a popular notion in the optimization literature.


**Questions:**

(1) Most real-world applications do not have a random stopping time $T$. The authors should better motivate and justify this setting. In addition, it is well known in the stochastic control literature that a control problem with a random stopping time sampled from an exponential distribution with rate lambda corresponds to an infinite-time horizon problem with discount factor lambda. The authors should make the linkage between these two classes of problems if there are any.

(2) Lemma 2 is listed without proof. This is an important building block that later results rely on. I doubt whether the PL condition could hold (as stated in Lemma 2) without additional assumptions. It is well-known that the PL condition does not hold for the general two-player linear-quadratic zero-sum game case and some carefully constructed conditions are required [1]. General-sum game [2] and MDP games [3] are much harder problems and I am surprised to see that PL condition holds almost for free in Lemma 2.

(3) The convexity of the projection set is never discussed. When $\Pi$ does not enjoy good properties, finding the projected point can be NP-hard.

(4) In Theorem 1, the authors assume that $\pi^*$ is a stable Nash policy. However, the authors never verified under what verifiable conditions the original problem has a Nash equilibrium policy and under what conditions it is unique. Similar issue also appears in Theorem 2. The authors never discussed the sufficient conditions for $\pi^*$ to be SOS. This leads to the danger that the main results (Theorems 1 and 2) may never be applicable.

Minor comments:

(a) The $a_t$ in Eqn (1) should be replaced by $\alpha_t$ based on the definition.

(b) Stopping time $T$ is not mathematically rigorously defined (i.e., the adaptiveness).

(c) Please be precise with what you mean by ``realized values’’ in line 217. I understand you refer to the instantaneous rewards but it is a bit vague in the statement.

[1] Zhang, Kaiqing, Zhuoran Yang, and Tamer Basar. "Policy optimization provably converges to Nash equilibria in zero-sum linear quadratic games." Advances in Neural Information Processing Systems 32 (2019)

[2] Mazumdar, Eric, et al. "Policy-gradient algorithms have no guarantees of convergence in linear-quadratic games." arXiv preprint arXiv:1907.03712 (2019)

[3] Xie, Qiaomin, et al. "Learning zero-sum simultaneous-move Markov games using function approximation and correlated equilibrium." Conference on learning theory. PMLR, 2020
=========
==========
I have read the reviews from other reviewers and the responses from the authors. The authors have addressed my concerns and I raised my rating to 6.

**Limitations:**

Yes.

**Strengths And Weaknesses:**

Pros: The topic studied in this paper is no doubt interesting and meanwhile challenging. Instead of adopting the first-order condition (KKT) used in most of the game papers in literature, this paper borrows the idea of a second-order stationary point from the optimization community and established the corresponding framework to understand the well-definedness and learning behavior in the stochastic game regime. The authors establish asymptotic convergence (in high probability) for a class of MDP games.

Cons:  However, I have some concerns regarding (1) the generality of the set-up (especially on the stopping time), (2) verifiability of the assumptions, and (3) key steps with missing proofs.

Detailed questions that require clarifications are listed below.

---

> ### Author Response · Authors · 2022-08-02
> **Replies 2/2**
>
> ---
> **[This post is a continuation of "Replies 1/2"]**
>
> 1. **On the existence of stable Nash policies.**
> The question of existence of a stable policy is akin to the existence of evolutionarily stable policies in population games, cf. the classical textbook of Sandholm (2010). As in that case, many interesting classes of stochastic games possess such policies, among them potential stochastic games, games with strict equilibria and a cooperative transition structure, etc. For example, in single-state repeated games with payoffs uniformly distributed in $[0,1]$, strict (and hence stable) Nash policies exist with probability that converges to $1-1/e \approx 64\%$ in the large $N$ limit, so this covers a very large class of games (Dresher, 1970).
>
>     As for verifying the SOS condition, it is easy to see that $\pi^\ast$ is SOS when the Jacobian $Jv(\pi^\ast)$ of $v$ at $\pi^\ast$ is positive-definite (cf. the proof of Proposition 1 in Appendix G), and this can be verified within $\mathcal{O}(NSA^{\log_2 7})$ arithmetic operations \textendash\ and, again, all potential games, single-state and cooperative games with strict equilibria admit such a policy.
>
>     To make sure there are no doubts for the reader, we have included a version of this discussion in our revision, as well as the relevant literature pointers.
>
>
> 1. **On Eq. (1).**
> Yes, this was a typo, thanks for the catch!
>
> 1. **On the definition of the stopping time $T(\tau)$.**
> Due to space restrictions, we avoided the formal definition of the stopping time. This random-stopping model is equivalent to (i) having a 'terminal' state $s_f$ with zero value that is reachable from every state-action pair $(s,\alpha)$ with probability $\zeta_{s,\alpha}$, and (ii) once reaching state $s_f$, the game terminates. Hence, we can define it formally as $T(\tau):= \inf\{t \in \mathbb{N}:s_t = s_f\}$. Since $\zeta_{s,\alpha} \geq \zeta >0$ for all $(s,\alpha) \in S \times \mathcal{A}$, we readily obtain that $\mathbb{E}[T(\tau)] < \infty$, and, subsequently, $T(\tau) < \infty$ on almost every trajectory. The reason is that $\mathbb{E}[T(\tau)]$ is upper-bounded by the expected value of a *geometrically* distributed random variable with parameter $\zeta$, which is finite.
>
>
> 1. **On the term "realized values".**
> Yes, we meant "instantaneous rewards", thanks for the catch!
>
>
> Thanks again for your detailed reading and your remarks! We hope and trust that our replies have alleviated your concerns regarding the merits of our submission, and we look forward to an open-minded discussion if any such concerns remain.

---

> ### Author Response · Authors · 2022-08-02
> **Replies 1/2**
>
> Thank you for your time and your input. We reply to your main questions point-by-point below, and we have colored all relevant revisions in our paper in $\color{purple}{\textrm{purple}}$. All numbering used below corresponds to the updated version of our paper if not explicitly mentioned:
>
>
> 1. **On the random stopping-time model.**
> In his seminal work, Shapley defined the problem for two-player zero-sum stochastic games using a random stopping time $T$. Motivated by this work, we used the random-stopping attribute in our model.
> Furthermore, when the ‘terminating’ probability $\zeta_{s,\alpha}$ is equal for every state-action pair, and equal to $\zeta_0$, the value function of the players in the random-stopping model is the same as in the infinite horizon case with discounted factor equal to $\zeta_0$.
> For policy gradient methods, an additional request to be satisfied for both models (random-stopping/infinite horizon with discounted factors) is the finding of an unbiased estimator for the gradient of the value function. While for the random stopping case the proof is direct (See Lemma 4 in Appendix F), for the case of infinite horizon multiple different approaches have been established. Although many sophisticated ones have been proposed (for the single agent case, see [1] and references therein), the easiest gateaway is actually a Monte Carlo simulation via random stopping-time, which leads back to the first formulation.
>
>     A further reason that supports our formulation passes through  the Von Neumann–Morgenstern utility axiomatization. More precisely, from game theory perspective, the players in a game should have a well-defined way to evaluate their utility function in each round. While in MARL with inifinite horizon this is in principle impossible, random stopping-time formulation describes syntactically how the players could evaluate probabilsitically their value.
>
>     For the interested reader, in the camera-ready version we will add a short paragraph to explain the equivalences between the various models.
>
>     [1] *Global Convergence of Policy Gradient Methods to (Almost) Locally Optimal Policies* Kaiqing Zhang, Alec Koppel, Hao Zhu, Tamer Başar.
>
> 1. **On Lemma 2 and the gradient dominance property.**
> We would like to respectfully point out that we ***included a proof of Lemma 2*** in "*Appendix G: Solution concepts*". We would also like to point out that we only said that stochastic games satisfy a ***version*** of the PL inequality: this property only holds for each agent's individual value function keeping all other players' policy variables fixed. Of course, this is a much weaker version of the PL inequality, but it suffices to show that first-order stationarity implies Nash (cf. Lemma 3).
>
>     To make things clearer, we have moved the proof of Lemma 2 to "*Appendix E: Structural properties of policy gradient methods*", and we included a clear pointer to the appendix in our revision, as well as a remark to clarify that the gradient dominance property is only a related version of the PL inequality, not the actual PL condition itself.
>
>
>
> 1. **On projecting to $\Pi$.**
> Thanks for bringing this up. Indeed, projecting to an arbitrary non-convex subset can be at least $\mathsf{NP}$-hard. However, ***$\Pi$ is not an arbitrary set:*** it is a Cartesian product of canonical simplices, so the projection to $\Pi$ immediately boils down to a projection to each factor simplex $\Delta(\mathcal{A}_i)$. This projection can be performed efficiently in $\mathcal{O}(A_i \log A_i)$ operations by sorting the components of the vector to be projected and, subsequently, doing a "water-filling" pass. This is a widely known procedure that dates back at least to Brucker in the 1980's, and an explicit description can be found in https://arxiv.org/pdf/1309.1541.pdf (see also the standard optimization textbook of Boyd and Vandenberghe, 2004, Exercise 4.1).
>
>     If the committee finds this useful, we will include a version of this discussion, as well as the relevant literature pointers.
>
> ---
> **[Please see next post for the continuation of our replies]**

---

> ### Author Response · Authors · 2022-08-02
> **Structure of our replies**
>
> Because of the openreview character limit, our replies were broken up into 2 posts, labeled as "Replies 1/2" and so on. Unfortunately, openreview comments appear in reverse chronological order, so we had to edit our posts in order for our replies to appear in a more natural order on openreview.

---

### Official Review · Reviewer_oi33 · 2022-07-11

**Rating:** 6
**Confidence:** 3
**Soundness:** 3 good
**Presentation:** 2 fair
**Contribution:** 2 fair

**Summary:**

The authors consider multi-player stochastic games. They study the convergence property of policy gradient type algorithms toward Nash equilibrium policies that are second-order stationary, similar to the type of KKT sufficiency conditions. They prove that policy gradient algorithms where the gradient is estimated with the reinforce algorithm, if started close to a second-order stationary equilibrium converge to this equilibrium, in terms of squared distance, with high probability at the rate O(1/\sqrt(n)) where n is the number of steps. If the Nash equilibrium is deterministic, for a lazy version of the policy gradient algorithm, the rate of convergence improves to a constant number of steps.

**Questions:**

Specific comments/questions:

-L30: By "game itself evolves" you mean the strategy of the other player or the rules of the game?

-L80: The notation $a_i$ for the action seems more natural than $\alpha_i$.

-L92: Do you have a concrete example of game that you can model with this framework? In particular, a game where after each step all players fully observe the state of the game.

 -L127: Can you elaborate on this assumption, which at first sight seems very strong in particular for deterministic policy. And do you have a non-trivial example where this assumption is true?

-L138: Can you precise what is the scalar product you consider and I assume that you see \pi as a vector of size S\times A right?

-L153: What do you mean by close?

-L163: Can you detail this point?

-L167: Can you clarify "sufficiently close" and what norm do you consider?

-L198: constructed

-L210-215: 'reverse P'

-L212: Can you precise what you mean by 'game involves training over datasets' and how you estimate the gradient.

-L230, (14): for all s?

-L235: Can you provide for G the dependence in A and S?

-L244: Do all stochastic games admit a stable Nash policy?

-L250: r is already used for the rewards.

-L278: Can you provide any intuition on what is the typical size of \mathcal{U}? Indeed if this set is too small there is no hope to be able to start in it.


-L285: Can you provide an order for the constant hidden by the O in terms of the parameters of the problem?
-L286: This part is not very clear if p=1 how do we know that q = 1/2 since 2p = q ?

-L289: Can you provide lower bounds in order to state that it is effectively the rate of convergence of PG ?

-L292: Can you explain how to 'to adapt the parameters of the learning process according to the complexity and limitations of the environment' in the light of Theorem 2?

-L332: Can you precise how n_0 depends in the parameters of the problem? Because if this constant  n_0 is very large in comparison to the one hidden in O(1/n) the relevant regime will still be the one of Corollary 2.

-L630: R is already used for the cumulative rewards.

-L634: a is not defined, is it any such that a+\sqrt{a} < r, r is the rate of \epsilon-greedy? If so not that it is only defined in Corollary 1 stated after Theorem 1.

-L656: Can you detail why the two limits are zero.

-L667: For (B16) you do not telescope the bound (B2) but just use that \pi_k is in \cB?

-L678: Can you explain why you need the indicator 1_{\cE_n} to get (B.24)?

-L700: + is missing in (B.32). And you should also detail quickly why the different sums are bounded for the choice of the parameters.

**Limitations:**

see Strengths And Weaknesses.

**Strengths And Weaknesses:**

Contributions:
-flexible algorithmic template for the analysis of policy gradient methods,  novelty: low, relevance: low.
-O(1/\sqrt(n)) rate of convergence of policy gradients method with reinforce gradient estimate towards second-order stationary equilibrium Nash equilibrium,  novelty: medium, relevance: medium.
-Finite-time convergence for a lazy version of the policy gradient methods toward a deterministic Nash equilibrium, novelty: medium, relevance: medium.


The paper is well-written even if some notations could be clarified, see specific comments. It could be interesting to ground the considered setting with a concrete example, see specific comments. Similarly, the claims are obtained through a cascade of hypotheses that are not exhaustively discussed and justify. The proofs seem correct at least in the part I read. The related work could be improved. For example, the author may also want to compare the obtained rate with the rate obtained by Jin et al. (V-Learning—A Simple, Efficient, Decentralized Algorithm for Multiagent RL , 2021) and reference therein since they seem to consider the same setting. It would also be interesting to discuss the results to provide lower bounds on the rate of convergence toward a Nash equilibrium.

---

> ### Author Response · Authors · 2022-08-02
> **Replies 3/3**
>
> ---
> **[This post is a continuation of "Replies 2/3"]**
>
> 1. **On the constants in the $\mathcal{O}(\cdot)$ guarantees of Theorem 2.**
> Getting an explicit estimate for the constant in the $\mathcal{O}(\cdot)$ guarantee of Theorem 2 is quite involved but, up to logarithmic and subleading factors, Chung's lemma [14,48] can be used to show that:
>     - For $q<2\mu\gamma$ and $p=1$ the constant is $\frac{C_1+C_2}{(2\mu\gamma -q)(1-\delta)}$, where $C_1,C_2$ is the $\sup_n\gamma_nB_n$ and $\sup_n\gamma_n^2\sigma_n^2$ respectively. We stress that these constants depend on the choice of parameters of the algorithm (i.e., $\gamma ,\varepsilon$ etc.) and the estimator used. This is the case used in the corollaries.
>     - For $q\geq 2\mu\gamma$ and $p=1$ the constant is $\frac{(C_1+C_2)(1+ \max\{(2\mu\gamma)^2,4\mu\gamma\})A}{1-\delta}$, where  $A$ is equal to $\frac{1}{q-2\mu\gamma}$ in the case of an inequality; while in the case where $q=2\mu\gamma$ corresponds to a logarithmic factor which we have omitted.
>     - For $p<1$, the constants are almost identical up to some dependence on $p$.
>
>
> 1. **On the case $p=1$.**
> In L286 of our original submission (the line numbers in the revision have changed because of the added material), we stated that $q=p/2$ (not $q=2p$), so $q=1/2$ if $p=1$. This equality results from the proof of Theorem 2: specifically, we define $q = \min\{\ell_b, p-2\ell_\sigma\}$; in the case of Model 3, $\ell_b = p/2$ and $\ell_\sigma = p/4$, which results  to $q=p/2$.
>
>
> 1. **On the dependence of $n_0$ on the parameters of the game.**
> The convergence time $n_0$ scales proportionally to the number of states and strategies in the game, the minimum payoff difference $c$ between an equilibrium strategy and a non-equilibrium strategy for any given state, and inversely proportionally to the algorithm's step-size (i.e., smaller step-sizes lead to larger values $n_0$) . Specifically, as can be seen from the last part of the proof of Theorem 3, $n_0$ can be upper bounded by the stopping time
> $
> n_0 \leq n^\ast = \inf_{n\geq1} \{ H_p(n) ≥ \frac{4MAS}{c\gamma}\}
> $
> where $M$ is a measure of the initial distance from equilibrium and $H_p(n) = \sum_{k=1}^{n} k^{-p}$ is the $n$-th generalized harmonic number of order $p$ (recall that the algorithm is run with a step-size of the form $\gamma_n = \gamma / n^p$). For $p<1$, $H_p(n)$ scales as $\Theta(n^{1-p})$, so, up to a universal constant, we have
> $
> n_0 = \mathcal{O}\left(\left(\frac{MAS}{c\gamma}\right)^{1/(1-p)}\right)
> $
> The dependence on the various parameters of the algorithm and the game cannot be lifted in the context of Corollary 2, so the finite-time convergence regime is always stronger in this regard.
>
>
> 1. **On the overload of $R$.**
> Good catch, thanks! We changed $R$ to $W$.
>
>
> 1. **On the constants $r$ and $a$ in the proof of Theorem 1.**
> No, $r$ referred here to the definition of $\mathcal{B}$, in L623 of the original submission. We have changed the notation of $r$ to $\varrho$ to avoid any overload.
>
>
> 1. **On the limit of $C(\gamma,m)$ in the proof of Proposition B.1.**
> Since $\gamma_n = \gamma / (n+m)^p$, the terms involving $\gamma_n$ in (B.15) both tend to $0$ as $\gamma\to0$ and $m\to\infty$.
>
>
> 1. **On telescoping (B.2).**
> Yes, for (B.16) we only need the fact $\pi_k$ lies in $\mathcal{B}$, this was a typo.
>
>
> 1. **On the indicator of $E_n$ in (B.24).**
> This was a typo, the indicator is not actually needed – thanks for catching this!
>
>
> 1. **On (B.32).**
> Thanks for catching the missing $+$ sign. The different sums are bounded because of (8) and the step-size conditions $p+\ell_b > 1$ and $p-\ell_\sigma > 1/2$ of Theorem 1.
>
>
> We thank you again for your detailed reading and your constructive input! We hope and trust that our replies have alleviated your concerns regarding the merits of our submission, and we look forward to an open-minded discussion if any such concerns remain.

---

> > ### Comment · Reviewer_oi33 · 2022-08-08
> > **Post rebuttal**
> >
> > Thanks for the detailed rebuttal. I'm globally satisfied with the author's answers and will modify my score accordingly.
> > Some comments on two specific points:
> >
> > -1 I agree that obtaining the last iterate is more convenient than only average convergence, especially since it is usually harder to compute the average policy in practice. But It also seems that you require more assumptions than in the mentioned paper, no?
> >
> > -4 I was rather thinking about imperfect information games for which there exists also a large literature:
> >
> >  *J. V. Romanovsky. Reduction of a game with complete memory to a matricial game. 1962
> >
> >  *Bernhard von Stengel. Efficient Computation of Behavior Strategies. 1996
> >
> >  *Daphne Koller, Nimrod Megiddo, and Bernhard Von Stengel. Efficient Computation of Equilibria
> > for Extensive Two-Person Games. 1996
> >
> >  *Marc Lanctot, Kevin Waugh, Martin Zinkevich, and Michael Bowling. Monte Carlo Sampling for
> > Regret Minimization in Extensive Games. In Advances in Neural Information Processing Systems,
> > 2009
> >
> >  *Samid Hoda, Andrew Gilpin, Javier Peña, and Tuomas Sandholm. Smoothing Techniques for Computing Nash Equilibria of Sequential Games. 2010
> >
> >  *Christian Kroer, Gabriele Farina, and Tuomas Sandholm. Solving Large Sequential Games with the Excessive Gap Technique. 2018.
> >
> >   *Gabriele Farina, Christian Kroer, and Tuomas Sandholm. Stochastic Regret Minimization in Extensive-Form Games. 2020
> >
> >  *....
> >
> > Indeed it is not clear if it is possible with your setting to model, e.g, a simple card game where one player does not know the opponent's hand. In this case, which I think covers a substantial part of the games in practice, the players do not have access to the state of the game after each round.

---

> > > ### Author Response · Authors · 2022-08-09
> > > **Thanks for the follow-up!**
> > >
> > > Thank you for your follow-up comments and your positive re-assessment! We reply to your two remarks point-by-point below:
> > >
> > > 1. ***On the assumptions of Jin et al.***
> > >
> > >     The only Nash equilibrium convergence result of Jin et al. concerns two-player zero-sum stochastic games; by contrast, our paper treats general stochastic games, so the assumptions of Jin et al. are stronger in terms of structure on this point. Other than that, the results of Jin et al. for general stochastic games concern (time-averaged) convergence to *coarse correlated equilibria* (CCE), which is a much weaker solution concept than Nash equilibrium. [For example, as was shown by Viossat and Zapechelnyuk (JET, 2013), there exist CCE that assign positive probability only to strictly dominated strategies, and thus fail even the most basic postulates of rationalizability.] Still, even in that case, Jin et al. also assume an initialization that gives positive weight to all states/actions, so their assumptions are similar to our own concerning the mismatch coefficient.
> > >
> > >     Overall, given that the analysis and results of Jin et al. are fundamentally incomparable to our own (in terms of both the type of convergence and the solution concepts involved), it does not seem possible to perform a finer ablation study between each paper's assumptions and results, so we did not undertake one.
> > >
> > >
> > > 1. ***On the issue of imperfect information.***
> > >
> > >     Thanks for the clarification that you had **extensive form games** in mind! Indeed, in the case of stochastic games *in extensive form*, there is an important distinction between perfect and imperfect information (Chess and Go for the former, versus Poker for the latter). Since we focus throughout on stochastic games *in normal form*, this distinction is not germane to our study, but we will make sure to include a remark and the relevant literature pointers that you brought up to clarify this in a subsequent revision.
> > >
> > > Thank you again for your constructive input and engagement – and please let us know if you have any further questions!

---

> ### Author Response · Authors · 2022-08-02
> **Replies 2/3**
>
> ---
> [This post is a continuation of "Replies 1/3"]
>
> 1. **On the scalar product.**
> The notation $\langle\cdot,\cdot\rangle$ actually refers to the canonical pairing between a primal and a dual vector, i.e., $\langle v,\pi \rangle = \sum_{j} v_j \pi_j$, where the summation over $j$ is taken over an index set of appropriate dimension, depending on the input to $\langle\cdot,\cdot\rangle$. This dimension is $S\times A_i$ for $\pi_i$ and the corresponding sum over all players $i=1,\dotsc,N$ for the policy profile $\pi = (\pi_1,\dotsc,\pi_N)$.
>
>
> 1. **On the meaning of "close" in Definition 2.**
> Formally, we mean here that there exists a neighborhood $\mathcal{U}$ of $\pi^\ast$ in $\Pi$ such that the stated inequality holds for all $\pi\in\mathcal{U}$.
>
>
> 1. **On meager sets.**
> In the sense of Baire's category theorem, a set is "meager" when it is a countable union of nowhere dense sets - and hence, negligible from a topological standpoint.
>
>
> 1. **On the notion of "sufficiently close" in Proposition 1.**
> Again, we mean here that there exists a neighborhood $\mathcal{U}$ of $\pi^\ast$ in $\Pi$ such that the stated inequality holds for all $\pi\in\mathcal{U}$. The norm can be taken to be the standard $L^2$ norm (though the choice of norm does not really matter in this context).
>
>
> 1. **On the meaning of "construe".**
> We actually did mean "construe" in the sense of "interpret".
>
>
> 1. **On the reverse P.**
> This is the typographic pilcrow sign, which we used to mark the end of each example and make it more visible to the reader. We can remove it if the committee finds it distracting.
>
>
> 1. **On training over datasets.**
> We simply referred here to games where the policy can be encoded as a neural network - as in the case of deep reinforcement learning. We removed the remark to avoid confusion. As for ways to estimate a stohastic gradient, Model 3 provides a model-agnostic way to do so; in concrete applications (e.g., in drone-flying), the gradient can be estimated efficiently using deep RL methods as above.
>
>
> 1. **On Eq. (14).**
> Yes, we meant uniform sampling for all $s$.
>
>
> 1. **On the value of $G$ in Eq. (15).**
> Notice that $b_{i,n} = v_i(\hat\pi_{n}) - v_i(\pi_{n})$, from lemma D.7 we know that $\|v_i(hat\pi_{n}) - v_i(\pi_{n})\|\leq \frac{3\mathcal{A}|}{\zeta^3}\sum_j\|\hat\pi_{j,n}-\pi_{j,n}\|$. Moreover, by the definition of $\pi_{i,n}$ and $\hat\pi_{i,n}$, we obtain that $|\pi_{i,n}(\alpha \mid s)-\hat\pi_{i,n}(\alpha \mid s)| \leq \varepsilon_n$ for all $s \in S$ , $\alpha \in A_i$, and therefore, $\|\pi_{i,n} - \hat\pi_{i,n}\| \leq \sqrt{SA_i} \varepsilon_n$. Combining the aformentioned quantities, we get the value of $G=\frac{3N|\mathcal{A}|^{3/2}\sqrt{|\mathcal{S}|}}{\zeta^3}$ in Eq. (15).
>
>
>
> 1. **On games that admit a stable Nash policy.**
> No, not all games admit a stable Nash policy - the question is akin to the existence of evolutionarily stable policies in population games. However, many interesting classes of stochastic games do, among them potential stochastic games, games with strict equilibria and cooperative transitions, etc. In particular, in single-state repeated games with payoffs uniformly distributed in $[0,1]$, strict (and hence stable) Nash policies exist with probability at least $1-1/e \approx 64\%$, so this covers a very large class of games (Dresher, 1970).
>
>
> 1. **On the overload of $r$.**
> Excellent catch, we have changed the exponent $r$ to $\ell_{\epsilon}$.
>
>
>
> 1. **On the size of $\mathcal{U}$.**
> The size of $\mathcal{U}$ scales with the minimum payoff difference between equilibrium and non-equilibrium actions per state, so it is only "small" in games with very small payoff differences between actions in a given state. The precise size involves solving a nonlinear inequality which does not admit a closed form description in general.
>
>
> ---
> **[Please see next post for the continuation of our replies]**

---

> ### Author Response · Authors · 2022-08-02
> **Replies 1/3**
>
> Thank you for your detailed reading and constructive comments. We reply to your main questions point-by-point below, and we have colored all relevant revisions in our paper in $\color{purple}{\textrm{purple}}$. All numbering used below corresponds to the updated version of our paper if not explicitly mentioned:
>
>
> 1. **On the paper of Jin et al. (2021).**
> Thanks for bringing up this paper. To put things in perspective, Jin et al. propose an algorithm (called V-learning) which is based on an adversarial bandit wrapper, and which updates the policy $\pi_n$ of the $n$-th episode based on the observed rewards so far. The paper's main contributions may then be summarized as follows:
>     1. In *two-player zero-sum* stochastic games, the time-averaged state $\bar\pi_n = (1/n) \sum_{k=1}^n \pi_k$ converges to a Nash policy at a rate of $\mathcal{O}(1/\sqrt{n})$ in terms of the game's primal-dual gap (cf. Theorem 5 of the paper, and note that the output policy $\hat\pi$ is obtained by sampling uniformly over $k=1,\dotsc,n$, so the theorem's guarantee concerns $\bar\pi_n$). We stress here that this result relies crucially on the zero-sum nature of the game, and *it does not apply to general stochastic games*.
>     1. In general stochastic games, the authors show that the empirical frequency of play (that is, the average number of action profiles played by a given policy) converges to the set of *coarse correlated equilibria* (a substantial relaxation of the notion of Nash equilibrium) at a rate of $\mathcal{O}(1/\sqrt{n})$ in terms of regret values (cf. Theorem 4).
>
>     By contrast, our paper focuses on **the actual sequence of play** (i.e., $\pi_n$ instead of $\bar\pi_n$) and the rate of convergence to **Nash policies** (not coarse correlated equilibria) in terms of the **distance** to such a policy (not the gap function or the regret). Needless to say, the convergence of time-averages is a much weaker convergence guarantee than the convergence of the actual trajectory of play to a Nash equilibrium: for example, if we want to minimize the loss function $f(x)=x^2$ over $[-1,1]$, the sequence $\pi_n = (-1)^n$ converges in the mean to $0$, even though each individual iterate of the sequence yields the worst possible loss. In this regard, the $\mathcal{O}(1/\sqrt{n})$ convergence rates of Jin et al. are incomparable to our own as they concern a weaker type of convergence (time-averaged instead of the actual sequence), to a coarser solution concept (correlated equilibria instead of Nash equilibria), and with a different merit function. To make all this clear, we have included this discussion in the revised version of our paper.
>
>
> 1. **On the evolution of the game.**
> Yes, we meant that the stage game evolves over time (based on the transition from one state to another).
>
>
> 1. **On the notation $a_i$ versus $\alpha_i$.**
> The choice of notation is, of course, subjective. Nash himself used $\alpha_i$ to denote the actions of a game, while Shapley used $a_i$ for the payoffs of the game (not the actions) in his foundational paper on stochastic games. To avoid any errors during this (very short) revision phase, we maintained our original notation on this point.
>
>
> 1. **On observing the state of the game.**
> To the best of our knowledge, the literature on stochastic games (including the paper by Jin et al. that you cited) is almost exclusively based on this natural assumption: after all, the very notion of a policy is defined as a *map from states to actions*, cf. the original paper of Shapley, the recent review monograph by Solan in PNAS, the standard textboks by Filar and Vrieze, as well as the series of recent NeurIPS/ICML papers by Agarwal et al., Daskalakis et al. and many others. In practice, the "state" of a physical RL system (e.g., an aerial drone) involves the state of the environment (i.e., observable weather conditions and the like), so it is indeed observed before taking an action (e.g., deciding at which altitude to fly).
>
>
> 1. **On the mismatch coefficient.**
> The positivity postulate for the mismatch coefficient holds whenever the initial state distribution is fully mixed, (i.e., $\rho$ assigns nonzero probability to each state in the game), an assumption which is fairly reasonable for most RL systems deployed in practice (e.g., game-playing and self-driving vehicles where all states can be observed initially). For this reason, this requirement is also standard in the literature on episodic stochastic games, see e.g., the references [a,b] below as well as the cited works [1,5,15] in the revised version of our paper and references therein.
>
>     [a] Runyu Zhang, Z. Ren, Na Li, *Gradient play in stochastic games: stationary points, convergence, and sample complexity*
>     [b] Runyu (Cathy) Zhang, J. Mei, Bo Dai, Dale Schuurmans, Na Li, *On the Effect of Log-Barrier Regularization in Decentralized Softmax Gradient Play in Multiagent Systems*
>
> ---
> **[Please see next post for the continuation of our replies]**

---

> ### Author Response · Authors · 2022-08-02
> **Structure of our replies**
>
> Because of the openreview character limit, our replies were broken up into 3 posts, labeled as "Replies 1/3" and so on. Unfortunately, openreview comments appear in reverse chronological order, so we had to edit our posts in order for our replies to appear in a more natural order on openreview.

---

### Official Review · Reviewer_wkBA · 2022-07-13

**Rating:** 6
**Confidence:** 4
**Soundness:** 4 excellent
**Presentation:** 3 good
**Contribution:** 3 good

**Summary:**

This paper studied the convergence of policy gradient (PG) methods with direct parameterization (Algorithm 2) to Nash equilibria in general stochastic games. The PG methods considered here apply to full gradient information, unbiased stochastic gradients with bounded variance, and REINFORCE estimator (Models 1-3). Explicit exploration ("epsilon greedy") is also used to bound the variance of PG estimators.

Theorem 1 showed that a stable Nash policy is locally attractive to PG with high probability.

Theorem 2 then showed that a second order stationary (SOS) Nash policy enjoys stronger results of last iterate convergence rates.

Theorem 3 showed that if a Nash policy is deterministic, then a lazy policy gradient (LPG) enjoys a finite convergence to exact Nash policies.

**Questions:**

1. Below Theorem 1 it was stated that the high probability convergence results cannot be improved further. Could you comment on what could be the possible failure cases (e.g., limiting cycles or chaotic behaviours)?

2. It seems to me SOS Nash policies enjoy convergence rates because of Eq. (C.9) has larger progress than Eq. (B.35). I am wondering for stable Nash policies, is it possible to establish a rate of convergence? If not what would be the difficulty, or any counterexample could be constructed to show the rate can be arbitrarily slow?

3. The finite convergence results in Theorem 3 is interesting. I am wondering if there is a parameterization to maintain the policy as valid probability distributions (e.g., standard softmax policies), then what is the implication of this lazy projection in that scenario and can it still provide useful insights?

**Limitations:**

This work is purely theoretical, and it has no potential negative societal impact as far as I can see.

**Strengths And Weaknesses:**

Strengths:

1. The setting of general stochastic games is general and representative for multiagent learning.

2. The PG methods studied here apply to multiple settings (full gradient, stochastic).

3. The results and techniques look novel and interesting to me. I like the idea of discussing different results conditioning on whether a Nash policy is stable, SOS, and deterministic.

4. The presentation is organized and clear.

Weaknesses:

1. It would be better if some simulation results can be provided to verify some of the results (high probability convergence and rate of convergence).

---

> ### Author Response · Authors · 2022-08-02
> **Reply to Reviewer wkBA**
>
> Thank you for your encouraging comments and positive evaluation! We reply to your main questions point-by-point below, and we have colored all relevant revisions in our paper in $\color{purple}{\textrm{purple}}$. All numbering used below corresponds to the updated version of our paper if not explicitly mentioned:
>
> 1. **On the possible failure cases for Theorem 1.**
> There are two types of unavoidable "convergence failures" that make Theorem 1 essentially tight.
>     + *Locality of attractors.* In general, equilibrium policies are not unique – and, indeed, as the reviewer suggests, gradient-based dynamics may also admit non-equilibrium attractors, such as limit cycles and the like (chaotic behavior is less relevant in the stochastic case, as chaos is an inherenetly deterministic notion). As a result, in the presence of multiple equilibria/attractors, the best one can hope for is a local equilibrium convergence result, conditioned on the basin of attraction of said equilibrium.
>     + *Probabilistic convergence.* The second obstruction has to do with the noise that enters the learning process (e.g., in the estimation of policy gradients via the REINFORCE algorithm). In this case, no matter how close one starts to an equilibrium policy, there is always a finite, non-zero probability that an unlucky realization of the noise can drive the process away from its basin, possibly never to return.
>
>     These issues can only be overcome in games with a sufficiently strong global struture – such as potential or min-max games – *but not otherwise*. We have revised the relevant part of our paper to make this clear.
>
>
> 1. **On rates of convergence for non-SOS stable policies.**
> The reviewer is correct that the extra driving term in (C.9) is what provides the convergence benefit over (B.35). In general, it is possible to derive a rate of convergence as long as the game satisfies a local inequality of the form $\langle v(\pi), \pi-\pi^\ast\rangle \geq (1/\rho)\|\pi - \pi^\ast\|^{\rho}$ for some suitable $\rho>0$; however, this level of generality didn't seem warranted, as such examples would not be generic. Beyond this, since the gradient profile near a solution can be arbitrarily flat (think of optimizing a quartic function like $x^4$ or some even higher power), it does not seem possible to obtain a rate of convergence in terms of distance to equilibrium unless there is some metric (sub)regularity condition linking the growth rate of the gradient around a solution to the distance. However, such an analysis would be beyond the scope of the current paper, so we did not undertake it.
>
>
> 1. **On finite-time convergence results for different projectors.**
> This is a very interesting question. Indeed, the projection step could be changed to some other mirror-like mapping – like a softmax/logit step, as suggested by the reviewer. However, a full-support method (like the one resulting from an exponentiated policy gradient scheme) would mean that the players' policy is fully mixed *for all* iterations of the algorithm, so it wouldn't be possible to achieve convergence to a deterministic policy in a *finite* number of iterations. Nevertheless, extending our analysis to mirror-type policy gradient schemes is a very fruitful direction for future research; we now discuss this as an open question in the conclusions section.
>
>
> 1. **On numerical experiments.**
> The numerical properties of the policy gradient algorithm have been studied quite extensively in the literature - see for example the cited papers by Leonardos et al. or Zhang et al. Given that the focus of our paper is the *analysis* of an existing, extensively tested algorithm (as opposed to proposing a new one), we felt that a raw verification of our theorems via numerical simulations would not offer further insights into the properties and behavior of policy gradient methods. Nonetheless, we would of course be happy to include some simulations in a further revision if the program committee converges that they are needed.
>
>
> We hope that the above addresses your questions and remarks, and we will be happy to clarify any remaining points during the discussion phase.
> Thanks again for your encouraging and constructive input!

---

### Meta-Review · Area_Chair_xmdt · 2022-08-25

**Recommendation:** Accept
**Confidence:** Less certain

**Metareview:**

This paper analyzes the convergence of policy gradient algorithms in "generic" stochastic games. The authors provide local convergence guarantees for projected gradient descent with the REINFORCE gradient estimator. Reviewers were generally positive on this paper--- though I think it needs to be much better contextualized in the literature on gradient-based learning in games (of which this is a special case).

Indeed--- while interesting in the context of MARL --- the results are not very surprising given that they seem very similar with other local analyses of (stochastic) gradient-play in games (see e.g., [1]). Furthermore the equivalence of equilibria follows from well known manipulations of the single-agent RL loss function like those performed in [2], genericity arguments for Nash equilibria [3], as well as work on variational inequality approaches to learning in games [4]. The final version of the paper should really comment on these previous results. Nevertheless, due to the positive reviews and the relevance to MARL, I recommend this paper for acceptance.

[1] Chasnov, Ratliff, Mazumdar, Burden; Convergence Analysis of Gradient-Based Learning in Continuous Games

[2] Zhang, Ren,  Li; Gradient play in stochastic games: stationary points, convergence, and sample complexity

[3] Ratliff, Burden, Sastry; Characterization and computation of local Nash equilibria in continuous games

[4] Mertikopolous and Zhou; Learning in games with continuous action sets and unknown payoff functions

**Award:**

No

---

### Decision · Program_Chairs · 2022-09-14

Accept